# A differentiable, physics-informed ecosystem modeling and learning framework for large-scale inverse problems: Demonstration with photosynthesis simulations

Doaa Aboelyazeed[1], Chonggang Xu[2], Forrest M. Hoffman[3,4], Jiangtao Liu[1], Alex W. Jones[5], Chris Rackauckas[6], Kathryn Lawson[1], Chaopeng Shen[1]

[1] Civil and Environmental Engineering, The Pennsylvania State University, University Park, PA 16802, USA

[2] Earth and Environmental Sciences Division, Los Alamos National Laboratory, Los Alamos, NM, 87544, USA

[3] Computational Sciences & Engineering Division and the Climate Change Science Institute, Oak Ridge National Laboratory, Oak Ridge, Tennessee, USA

[4] Department of Civil and Environmental Engineering, University of Tennessee, Knoxville, Tennessee, USA

[5] SciML. Open Source Software Organization, https://sciml.ai

[6] Computer Science and Artificial Intelligence Laboratory (CSAIL), Massachusetts Institute of Technology, Cambridge, MA, 02139, USA

*Correspondence to*: Chaopeng Shen (cshen@engr.psu.edu)

**Abstract.** Photosynthesis plays an important role in carbon, nitrogen, and water cycles. Ecosystem models for photosynthesis are characterized by many parameters that are obtained from limited in-situ measurements and applied to the same plant types. Previous site-by-site calibration approaches could not leverage big data and faced issues like overfitting or parameter non-uniqueness. Here we developed an end-to-end programmatically differentiable (meaning gradients of outputs to variables used in the model can be obtained efficiently and accurately) version of the photosynthesis process representation within the Functionally Assembled Terrestrial Ecosystem Simulator (FATES) model. As a genre of physics-informed machine learning, differentiable models couple physics-based formulations to neural networks that learn parameterizations (and potentially processes) from observations, here photosynthesis rates. We first demonstrated that the framework was able to recover multiple assumed parameter values concurrently using synthetic training data. Then, using a real-world dataset consisting of many different plant functional types, we learned parameters that performed substantially better and greatly reduced biases compared to literature values. Further, the framework allowed us to gain insights at a large scale. Our results showed that the carboxylation rate at 25°C ($V_{c,max25}$), was more impactful than a factor representing water limitation, although tuning both was helpful in addressing biases with the default values. This framework could potentially enable a substantial improvement in our capability to learn parameters and reduce biases for ecosystem modeling at large scales.

**Short Summary.** Photosynthesis is critical for life and is affected by a changing climate. Many parameters come into play when modeling, but traditional calibration approaches have faced many issues. Our framework trains coupled neural networks to provide parameters to a photosynthesis model. Using big data, we independently found parameter values that were correlated with those in the literature while giving higher correlation and reduced biases in photosynthesis rates.

## 1. Introduction

Plant photosynthesis is critically important for regulating the global carbon and nutrient cycles, and thus the future climate. Understanding future climate trajectories requires the understanding of photosynthetic responses to changes in environmental

factors including atmospheric $CO_2$ concentrations, radiation, temperature, humidity, and nutrient and water availability (Kirschbaum, 2004). Photosynthesis is influenced by many factors such as higher $CO_2$ levels, reduced productivity of vegetation (i.e., nutrient concentration) (Thompson et al., 2017), intensified droughts (Urban et al., 2017; Xu et al., 2019) and rising temperatures (Dusenge et al., 2019) under a changing climate. To comprehensively evaluate the impacts of these changing processes and vegetation feedbacks to the atmosphere, we need accurate representations of photosynthesis in our models.

For global assessments of the carbon cycle, vegetation models were developed to simulate terrestrial ecosystem processes and the distributions of vegetation, both vertically in the soil-plant system and horizontally across the landscape. Substantial efforts over the last few decades have improved the representation of vegetation and its responses and feedbacks to climate change (Fisher et al., 2018). A typical framework structure for a vegetation model is to keep track of changes in carbon and optionally nutrient states, driven by climatic variables and modulated by soil properties, with feedback to the climate, e.g., $CO_2$ releases, radiation, and vegetation composition and structure. A core component of the vegetation module is photosynthesis (Quillet et al., 2010).

Present ecosystem models for photosynthesis are based primarily on mechanistic descriptions of plant photosynthesis pathways, but this theoretically-sound modeling paradigm faces many challenges, with parametric uncertainty being a major one. Photosynthesis models may describe limitations of carboxylation rates, light availability, and plant-specific factors like enzyme efficiencies for $C_3$ and $C_4$ plants differently (Farquhar et al., 1980; Farquhar and Caemmerer, 1982; Meyer, 1983; Von Caemmerer, 2003, 2013; Yin and Struik, 2009). They contain many parameters that quantify these efficiencies and limitations. In the past, these parameters have been estimated from different approaches: 1) obtained from a limited set of *in-situ* sites and scaled based on climate and environmental factors (Verheijen et al., 2013); 2) calibrated on observational data site by site or for a few sites for a specific plant functional type (PFT) (Mäkelä et al., 2019; Wang et al., 2014); or 3) optimized based on environmental conditions (Ali et al., 2016). However, these estimated values may not be optimal at the global scale. Site-by-site calibrations using genetic or similar algorithms are highly expensive and are limited in their spatial coverage and generalizability to different PFTs and species. Furthermore, such calibration faces the issue of nonuniqueness (which some call equifinality (Beven and Freer, 2001)), where different parameter sets produce the same outcome. As a result, calibration can easily lead to poorly-generalizable parameter values. This problem exists for many domains with diverse parameters, including ecosystem modeling (Tang and Zhuang, 2008). It is similarly found in hydrologic modeling and has troubled scientists there for decades (Beven, 2006). More recently, some parameters can be fitted directly from large datasets with directly measured parameter values (Luo et al., 2021), which is highly valuable but is limited to those parameters with extensive observations, e.g., soil water retention and hydraulic properties. An efficient way to permit large-scale inverse modeling is needed.

There has been substantial progress in utilizing modern machine learning (ML) for geosciences. Purely data-driven deep learning models (LeCun et al., 2015; Reichstein et al., 2019; Shen, 2018; Shen et al., 2018) directly learn from data so they tend to be fairly accurate, and many have outperformed traditional models for a large number of applications across domains such as hydrology (Feng et al., 2020, 2021; Rahmani et al., 2020, 2021; Liu et al., 2023; Wunsch et al., 2021; Fang and Shen, 2020), agriculture (ElSaadani et al., 2021; Hossain et al., 2019; Liu et al., 2022a; Saleem et al., 2019), energy balance (Zhu et al., 2021), cryosphere (Leong and Horgan, 2020; Zhang et al., 2019), water quality (Hrnjica et al., 2021; Zhi et al., 2021; Saha et al., 2023; Zhi et al., 2023), and ecosystem modeling (Zhang et al., 2020, 2021). Especially, the loss function is defined on all data points and sites, enforcing a stronger constraints and leading to the data synergy effect, where the model gains better performance as the amount of training data increases (Fang et al., 2022). Unfortunately, deep learning models also lack interpretability and process clarity, and can only output trained variables with extensive observations. This need for data is often not satisfied for ecological processes.

To aid geoscientific models in general, Tsai et al. (2021) presented an efficient framework known as differentiable parameter learning (dPL), in an effort to leverage recent progress in ML to mitigate the issues with parameter inversion discussed above. This framework turns parameter estimation into a large-scale ML problem. It is mainly composed of a parameter estimation module based on a neural network (NN), combined with a process-based model (or its surrogate). The whole framework must be "programmatically differentiable" (Baydin et al., 2018; Innes et al., 2019), which refers to a programming paradigm that can efficiently and accurately obtain the gradients of the outputs with respect to any of the variables used in the model (Shen et al., 2023). Once we have programmatic differentiability, dPL can efficiently learn unknown functions from big data to serve as either a parameterization or process representation. Tsai et al. (2021) found that this framework scales well with more data, produces spatially and temporally well-generalized parameter sets, extends well to uncalibrated variables, and saves orders of magnitude in computational time. Feng et al. (2022a) further showed that a differentiable process-based hydrologic model with dPL could approach the performance of a purely data-driven ML model, and potentially outperform ML in data-sparse regions (Feng et al., 2022b). These successes can be conveniently migrated to the ecosystem modeling domain.

Here, we applied the dPL framework to the leaf scale photosynthesis module of the Functionally Assembled Terrestrial Ecosystem Simulator (FATES) model. FATES is an ecosystem model that describes co-existence and competition in plant functional types (PFTs) (Koven et al., 2020). FATES can be used as an ecosystem module in the Community Land Model (CLM) (Oleson et al., 2013; Lawrence et al., 2019) to represent the ecosystem demography (Fisher et al., 2015). The photosynthesis module is based on the Farquar photosynthesis model. To apply the dPL framework in our study, we first reimplemented the photosynthesis module from FATES so that it became programmatically differentiable. Second, we connected this model to neural networks for parameter estimation. With this tool, we aimed to answer the following questions: *(1) What is the achievable model performance, in terms of predicting photosynthesis rates in space and time, by tuning the parameters for the classical photosynthesis module without changing the model structure? (2) Are parameters like $V_{c,max25}$ and*

105 *soil water limitation factor simultaneously identifiable? (3) Are parameters learned from a large global dataset similar to the values used in current models?* In the following, we first described the photosynthesis model with different parameter estimation experiments and target datasets. We then discussed the parameters chosen to be estimated and their significance. Afterward, we presented the results from synthetic experiments and experiments based on real datasets from sites around the globe. Finally, we compared the learned parameters to values from the literature and provided some suggestions for future
work.

## 2. Methods and datasets

### 2.1. General overview

Our general framework trains connected neural networks to provide parameters (and later process representations) to process-based models (PBM), in this case the photosynthesis module of the FATES ecosystem model, on all the training data points simultaneously (Figure 1a). The neural networks map from some raw inputs to some tuneable physical parameters ($\theta$) (later
extensible to processes) required for the PBM. The predicted physical parameters are then fed into the differentiable PBM along with other required forcing variables (F) and untuned constant attributes ($\theta_c$) to compute the simulated target variable ($y_{sim}$) which is compared with observations to compute a loss function. The forward run starts from the neural network inputs and ends at the loss function (following the blue arrows in Figure 1a); the model's goal is to minimize the output of the loss
function. We then backpropagate the errors (shown by black arrows in Figure 1a) through the PBM equations back to the neural networks so we can train them using gradient descent. To support gradient-based training, the entire framework must be differentiable (Shen et al., 2023) which ensures that neither the neural network nor the process-based model is a black box --- they both allow explicit inspection and modification of the internal structures. Thus, the photosynthesis module of FATES had to be reimplemented on a differentiable platform.

In this case, the process-based model is the photosynthesis module in FATES, which can be written as a nonlinear system of equations, and its solution is implicit. The system can be written as:

$$f(x;\ \theta, \theta_c, F) = 0;\ y = h(x, \theta, \theta_c, F) \tag{1}$$

where *f* represents the physical system constraint, *h* is an observation operator, *x* represents the unknowns of the equations (in this case the internal leaf $CO_2$ partial pressure [Pa]), *y* is an observable variable (in this case net photosynthetic rate [$\mu$mol m$^{-2}$ s$^{-1}$])
that is dependent on *x* via *h*, *F* represents some meteorological forcing variables such as radiation and air temperature, $\theta$ represents a list of tuneable physical parameters, and $\theta_c$ represents untuned constant attributes. Given a set of $\theta$ with known $\theta_c$ and *F*, we need to solve for *x* from *f* and send the solution into *h* to further compute *y*: y = $h(f^{-1}(\theta, \theta_c, F), \theta, \theta_c, F)$. This whole workflow can be lumped into one model:

$$y = \delta_{psn}(\theta, \theta_c, F) \tag{2}$$

where $\delta_{psn}$ represents the overall photosynthesis model. Some of the tuneable parameters are typically formulated as being Plant Functional Type (PFT)-dependent (e.g., the maximum carboxylation rate at 25°C, $V_{c,max25}$) where each PFT includes groups of plant species that share similar physical and phenological characteristics leading to similar interactions with the environment. Other tuneable parameters are related to soil water availability (e.g., the soil water stress parameters). We posit that there exists a parameterization scheme, $\theta = g^W(R)$, which is a mapping relationship from some underlying attributes $R$ (e.g., soil attributes and plant traits) to the physical parameters represented by neural network $g$ with $W$ learnable weights. Thus, we can learn $W$ so that the simulated variable $y$ matches the observations $y^*$:

$$W = \underset{W}{\mathrm{argmin}}(L(\delta_{psn}(g^W(R), \theta_c, F), y^*)) \tag{3}$$

where $L$ is the loss function. For the purpose of solving the inverse problem and training the neural network $g^W$ in an "online" mode using gradient descent (the only practically-employed algorithm for neural network training), we reimplemented the photosynthesis module in FATES onto two differentiable platforms: Julia and PyTorch (discussed in more detail below).

In order to test the learning capability of our framework and the identifiability of the parameters, we first ran synthetic experiments to verify if the model would be able to correctly retrieve assumed values for the physical parameters. Second, using a dataset with thousands of photosynthesis rate measurements, we trained the differentiable model to obtain estimated parameters at the global scale, and compared them to the literature.

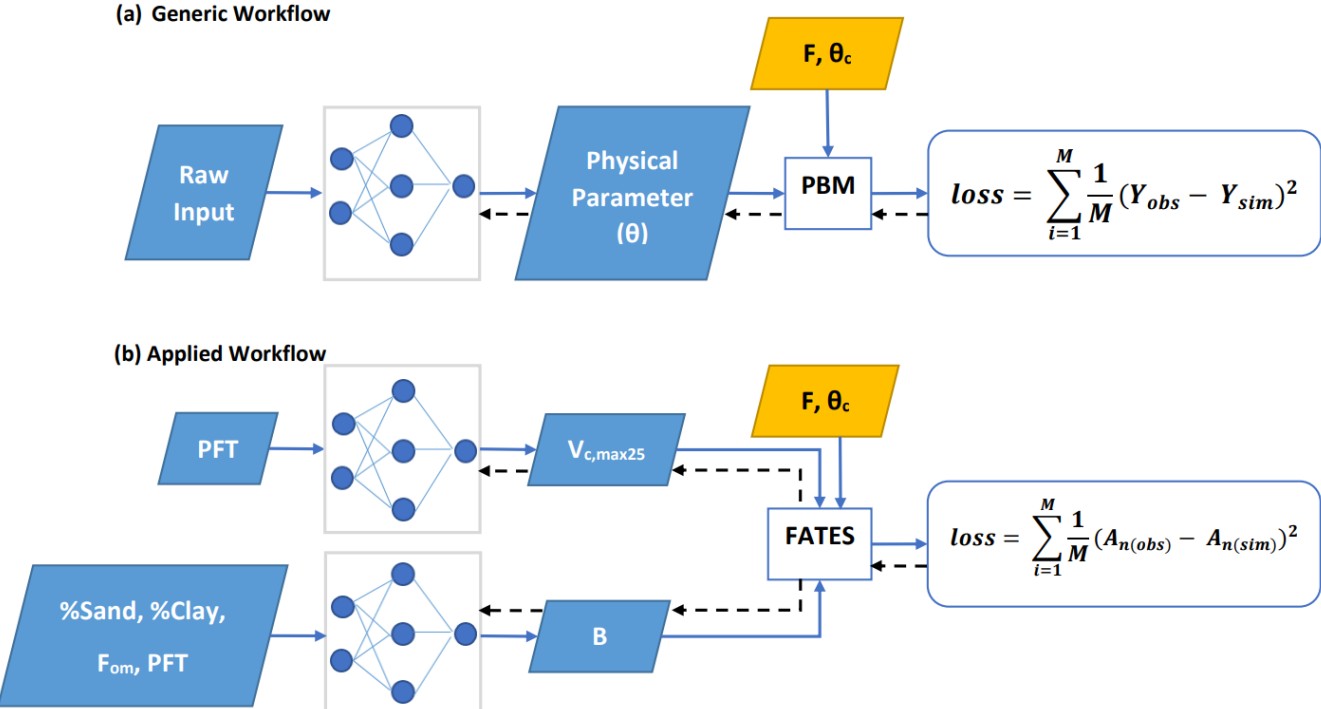

**Figure 1. Diagram showing the differentiable parameter learning (dPL) framework which is a hybrid of neural networks and the photosynthesis module in the FATES ecosystem model written on a differentiable platform. (a) The generic workflow: Some raw information is mapped into physical parameters via a neural network. These parameters are sent into a process-based model (PBM), which then outputs variable Y that is compared with the observations of Y. Direct supervision for the physical parameters themselves is not required -- we do not need ground truth values for these parameters. The loss function is "global" in that it involves all training data points, rather than being computed site-by-site as done in traditional calibration. (b) The workflow for the computational example described in this work. We estimate either $V_{c,max25}$ or the parameter B, or both of them at the same time, using neural networks. The parameters are then fed into the differentiable photosynthesis module in FATES, which then outputs the net photosynthesis rate, $A_{n(sim)}$, that is compared with $A_{n(obs)}$. When not being estimated from data, default values from the literature were used. Blue arrows show running the neural networks with the PBM in a forward mode ("prediction" mode), while black arrows indicate backpropagation from the loss function back through the differentiable model equations to the neural networks to update their weights, which is only done during initial NN training.**

## 2.2. The Farquhar photosynthesis model

The FATES photosynthesis module is based on the classical Farquhar model for $C_3$ plants (Farquhar et al., 1980), which calculates the photosynthetic rate based on carbon fluxes under different limitations. For $C_4$ plants, it uses the Collatz model (Collatz et al., 1992). Both models assume that the gross photosynthetic rate is affected by the maximum rate of carboxylation and is limited by the concentration of RuBP carboxylase (Rubisco) ($A_c$), light and electron transport ($A_j$), and the concentration of PEP carboxylase enzyme in $C_4$ plants ($A_p$). The final gross photosynthetic rate "$A$" is calculated using the empirical curvature parameters ($\theta_{cj}$ and $\theta_{ip}$), while the net photosynthetic rate $A_n$ is the same as the gross rate ($A$) after the plant respiration rate ($R_d$)

is subtracted. The system can be described succinctly as the following, with Equations 4 and 5 playing the roles of $f$ and $h$ in Equation 1, respectively, and the whole set of associated equations detailed in Appendix A.

$$C_i = C_a - A_n P_{atm} \frac{(1.4g_s + 1.6g_b)}{(g_s \times g_b)} \tag{4}$$

$$A_n = A(C_i) - R_d \tag{5}$$

Equation 4 is a single-variable nonlinear equation, with the intercellular leaf $CO_2$ pressure ($C_i$) as the unknown term to be solved (serving as the $x$ term in Equation 1). $C_i$ is influenced by the $CO_2$ partial pressure near the leaf surface ($C_a$), the net photosynthetic rate ($A_n$), the atmospheric pressure ($P_{atm}$), the leaf stomatal conductance ($g_s$), and the leaf boundary layer

conductance ($g_b$). Upon solving for $C_i$, we can further calculate $A_n$, which is the $y$ term in equation 1. In the original implementations of FATES and CLM, the system of nonlinear equations was solved iteratively using fixed-point iteration (Oleson et al., 2013).

In order to train the physical equations and neural networks together using gradient descent, the above equations were

implemented on differentiable platforms to support backpropagation. We developed two alternative implementations: using PyTorch in Python (Paszke et al., 2019), and using Julia (Bezanson et al., 2012). The PyTorch version solves the coupled nonlinear equations using our parallel implementation of Newton iteration with automatic differentiation, while the Julia version uses adjoint-based methods implemented via a symbolic computer algebra system and is compatible with a wide variety of nonlinear solvers (Gowda et al., 2022). In contrast to the previous fixed-point iteration used by the original FATES

model, our PyTorch Newton iteration solver can run on a graphical processing unit (GPU) in parallel for many sites. Newton's iteration features second-order convergence compared to the slower convergence of fixed-point iteration, while GPU parallelism represents orders-of-magnitude savings in computational time compared to the original algorithm in FATES. The photosynthesis problem studied here has only one unknown ($C_i$) even though there are many other supporting equations, but we have successfully tested other larger nonlinear systems. Altogether, we can train this model with the coupled neural

networks for hundreds of data points in under 10 minutes (typically in 600 iterations) and could also train the model on 10,000 data points. For future work where time steps are involved, the adjoint method will likely be employed to reduce GPU memory use during nonlinear iterative solving. For the Julia implementation, the symbolic toolbox ModelingToolKit.jl (Gowda et al., 2022; Ma et al., 2021b) was employed to automatically generate the solution scheme as Julia code, and along with solvers from NonlinearSolve.jl, solve the system of equations in the forward problem. Presently, we have implemented the Julia

version in serial mode only. Results presented in this paper were produced using the PyTorch version, although the computational results were the same with the Julia version. Hence, we think both versions have value for future effort.

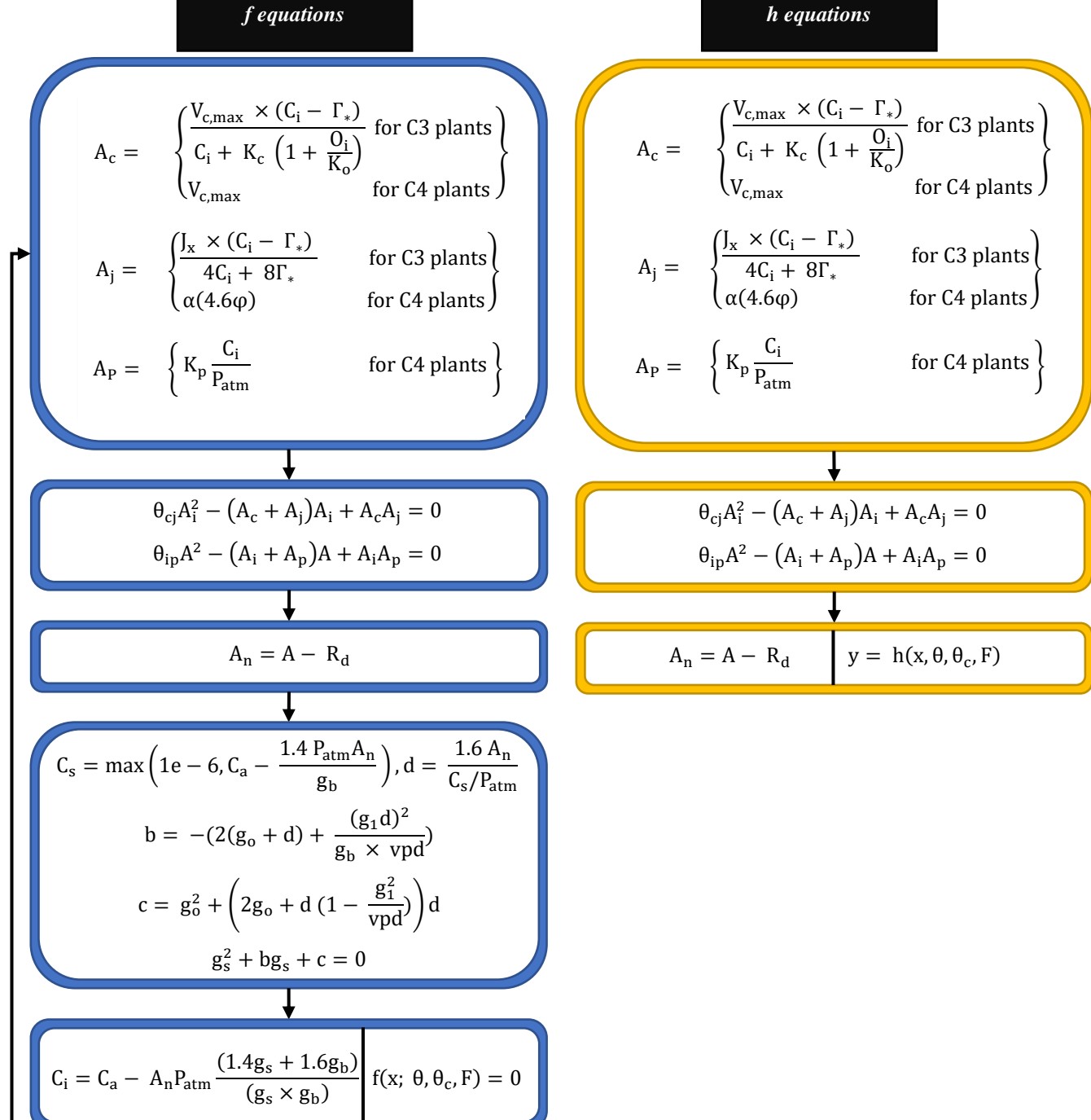

**Figure 2. Model equations corresponding to *f* and *h* in equation 1. Blue boxes indicate equations corresponding to *f*. Yellow boxes indicate equations corresponding to *h*. First, we obtain a solution for $C_i$ (intercellular leaf $CO_2$ pressure) by solving the nonlinear system (*f* equations) as illustrated in the last (bottom) blue box. Then, we run the *h* equations**

forward to compute $A_n$ (net photosynthesis rate) using $A_c$, $A_j$, and $A_p$ as discussed in section 2.2. Details about different variables and parameters included in the $f$ and $h$ equations are provided in Appendix A.

## 2.3. The parameterization pipeline and model changes

We used multilayer perceptron (MLP) neural networks as the parameterization module $g$ in Equation 3. The purpose of the MLPs is to estimate parameters $\theta$, which are then fed into the photosynthesis module to obtain the net photosynthetic rate ($A_n$) (Appendix A). The MLPs were trained based on minimization of the loss function – in brief, the goal is to minimize the difference between the solved and observed values of $A_n$. As described in Equation 2, the whole workflow is hereafter referred to as the $\delta_{psn}$ model ("delta-photosynthesis model") (the Greek letter $\delta$ is selected because the model is programmatically differentiable and $\delta$ is often associated with gradients). There may be multiple MLPs to estimate different parameters in $\theta$, each with different inputs of either continuous or categorical data, and they can all be trained together. Figure 1a shows the general framework for parameter estimation experiments and 1b shows the framework of this particular work.

We chose to estimate one or both of two specific parameters in our experiments. The first one is the plant maximum carboxylation rate at 25°C ($V_{c,max25}$), which is normally formulated as a PFT-dependent parameter. Although $V_{c,max25}$ is hypothesized to be PFT-dependent, recent studies have shown that the parameter can vary in space and time and by different species in the PFT as well (Ali et al., 2015; Chen et al., 2022; Qian et al., 2019). Estimating $V_{c,max25}$ is not a trivial matter due to its high variability and sensitivity to different factors such as drought, leading some studies to suggest a substitute for it. For example, Croft et al. (2017) suggested using the leaf chlorophyll content as a direct proxy for $V_{c,max25}$. Nevertheless, considering this is an initial study applying dPL, we followed the convention and parameterized it based on PFT:

$$V_{c,max25} = NN_V (PFT) \tag{6}$$

where PFT is the plant functional type category (in one-hot encoding format, which translates each category to a binary vector) and the neural network used for parameterization of $V_{c,max25}$ is referred to as $NN_V$ hereafter.

The second parameterization is for parameter $B$ defined by Clapp & Hornberger (1978), which influences the soil water stress function ($\beta_t$, where the subscript $t$ indicates transpiration). $\beta_t$, called "btran" in the CLM code, reflects the impacts of soil wetness on stomatal conductance and ranges from zero (extreme dry conditions causing stomata closure) to one (wet conditions with stomata fully open). In the following, we describe $B$ and $\beta_t$ computations as in CLM4.5 (Oleson et al., 2013). $B$ is purely a function of soil properties and is defined for each soil layer as $B_i$ where $i$ refers to the soil layer (see Appendix B). $B_i$ equations will later be replaced by our NN-based parameterization scheme as explained in section 2.3.1, because they were originally empirical and may not be optimal at the global scale. $B$ comes into play when calculating the soil water potential $\psi_i$ using a power-law formulation:

$$\Psi_i = \Psi_{sat,i} \times S_i^{-B_i} \geq \Psi_c \tag{7}$$

where $\psi_{sat,i}$ is the saturated soil matric potential and $S_i$ is the soil wetness, both defined for a specific soil layer (see Appendix B for detailed calculations). The plant wilting factor ($w_i$) is then calculated using $\psi_i$ and other PFT-dependent parameters (defined in CLM4.5) such as the soil matric potentials for closed stomata $\psi_c$ and open stomata $\psi_o$, which represent the soil water potentials when stomata are fully closed and fully open, respectively, as defined in equation (8). The factor $w_i$ is also dependent on other factors like the temperature of the soil layer ($T_i$) relative to the freezing temperature ($T_f$), the volumetric liquid water ($\theta_{liq}$) and ice ($\theta_{ice}$) contents, and the volumetric water content at saturation ($\theta_{sat}$). In our calculations, $\theta_{ice}$ was ignored since both the leaf and the air temperatures in our dataset were above the freezing temperature (0 °C or 273.15 K) by at least 5 °C.

$$w_i = \begin{cases} \dfrac{\Psi_c - \Psi_i}{\Psi_c - \Psi_o}\left[\dfrac{\theta_{sat,i} - \theta_{ice\,i}}{\theta_{sat,i}}\right] \leq 1 & ; \ T_i > T_f - 2 \text{ and } \theta_{liq,i} > 0 \\ 0 & ; \ T_i \leq T_f - 2 \text{ or } \theta_{liq,i} \leq 0 \end{cases} \tag{8}$$

Finally, $\beta_t$ can be calculated by aggregating the plant wilting factor ($w_i$) and plant root distribution ($r_i$) across different soil layers based on the PFT as:

$$\beta_t = \sum_i w_i r_i \tag{9}$$

### 2.3.1 Model changes

In the original water limitation function in CLM4.5, the stomata response to soil water potential is based on a linear function between the water potential for stomata openness and closeness (see Equation 8). In light of the possibility that plants could respond differently to soil water potential dependent on plant hydraulic traits (Christoffersen et al., 2016), in this study, we modified the soil water limitation for PFTs so that they could have different shapes. Specifically, we defined PFT-dependent soil water stress, $\psi_i(PFT)$ ranging from $\psi_c$ and $\psi_o$, depending on the soil water content, which is calculated as follows:

$$\Psi_i(PFT) = \Psi_o \times S_i^{-B_i(soil,PFT)} \geq \Psi_c \tag{10}$$

$B_i$ is a PFT- and soil-texture-dependent shape parameter (between 0 and 1) estimated as:

$$B_i = NN_{Bi}(\%sand_i, \%clay_i, F_{om,i}, PFT) \tag{11}$$

where *%sand$_i$*, *%clay$_i$*, and *F$_{om,i}$* respectively represent the percentage of sand, the percentage of clay, and the fraction of organic matter in soil layer *i*. The PFT-dependent soil water stress, $\psi_i(PFT)$, is then fed into the plant wilting equation (9) as the following:

$$w_i = \frac{\Psi_c - \Psi_i \, (PFT)}{\Psi_c - \Psi_o} = \frac{\Psi_c - \max\left(\Psi_c, \Psi_o \times S_i^{-B_i(soil,PFT)}\right)}{\Psi_c - \Psi_o} \leq 1 \qquad (12)$$

The new shape parameter $B_i$ in equation 11 has a different range (between 0 and 1) from the original one defined by Clapp & Hornberger (1978) in equation 7 and it varies spatially for different static attributes and for different PFTs as well. The default equations in the Community Land model V4.5 (CLM4.5) for computations of $B_i$ (Appendix B) show that the parameter $B_i$ depends on two attributes, $\%clay_i$ and $F_{om,i}$, which is why they were used in $NN_{Bi}$. To account for the dependence of $\psi_{sat,i}$ on $\%sand_i$ (Appendix B) and its replacement by $\psi_o$ (see equations 7 and 10), $\%sand_i$ was also added to $NN_{Bi}$. We also added PFT to $NN_{Bi}$ inputs because vegetation may interact with the soil moisture constraint and we want to allow relevant factors to be included, rather than restricting the list of inputs to what was previously used in the literature. Since in $NN_{Bi}$ we use quantitative inputs ($\%sand_i$, $\%clay_i$, $F_{om,i}$) along with categorical inputs (PFT), we used the embedding layer in PyTorch, which translates each category to a vector of quantitative variables. This categorical data can then easily be combined with other quantitative inputs we provide to our neural network.

Using the original Equation 7 for computing $\psi_i$ resulted in a plant wilting factor, $w_i$, equal to one for more than 90% of the datapoints across different soil layers. Changing Equation 7 to the form shown in Equation 10 helped to express more variability in $w_i$ and eventually in the computed soil water stress function ($\beta_t$).

**2.4. Input and observation datasets**

**2.4.1 Forcing and Photosynthesis rates:**

For meteorological forcings, we used the ERA5 Reanalysis dataset (Copernicus Climate Change Service (C3S), 2017), which provides hourly estimates of soil moisture at different soil depths. Soil moisture contributes to the computation of $\beta_t$ (see Appendix B), where the soil wetness $S$ depends on both the soil moisture and the saturated soil moisture.

For observations of photosynthesis, we used data from the leaf gas exchange database (Knauer et al., 2018; Lin et al., 2015) which is a global database of stomatal conductance measurements and leaf-level photosynthetic rates. It incorporates data from several sites around the world in Australia, Europe, USA, and Asia (Figure 3). We refer to this dataset as Lin15 throughout the rest of this work, with 43 sites chosen whose dates and times of measurements were available. Lin15 covered nine different PFT categories: rainfed crop "Crop R", Broadleaf Evergreen Tree Tropical "BET Tropical", Broadleaf Evergreen Tree Temperate "BET Temperate", C3 grass, C4 grass, Needleleaf Evergreen Tree Boreal "NET Boreal", Needleleaf Evergreen Tree Temperate "NET Temperate", Broadleaf Deciduous Tree Temperate "BDT Temperate", and Broadleaf Deciduous Shrub Temperate "BDS Temperate". Measurements were taken on a sub-hourly scale but not necessarily on a continuous daily interval. That's why for almost all the sites, data were available on some random days (not necessarily continuous) in one or a few years.

Lin15 also contained forcing variables, including air temperature ($T$), leaf temperature ($T_v$), atmospheric pressure ($P_{atm}$), relative humidity ($RH$), photosynthetic active radiation ($\varphi$) and boundary layer conductance ($g_b$).We used ERA5 to fill in for any missing forcing variables in Lin15. In equation 4, $P_{atm}$ and $g_b$ were used directly from the dataset, while $C_a$ was estimated using observations of the leaf surface $CO_2$ concentrations, and $g_s$ was calculated using the Medlyn conductance model (Medlyn et al., 2011) as explained in Appendix A.

### 2.4.2 Static attributes:

For $\beta_t$ calculations, we used data from Hengl & Wheeler (2018) for the soil organic carbon content at different soil depths, where the conventional Van Bemmelen factor of 1.72 was used to convert to soil organic matter ($F_{om}$). Data for sand and clay percentages (*%sand*, *%clay*) were obtained from Hengl (2018). Both are global datasets available at 250 m resolution at 6 different soil depths (0, 10, 30, 60, 100, and 200 cm) which describe five soil layers.

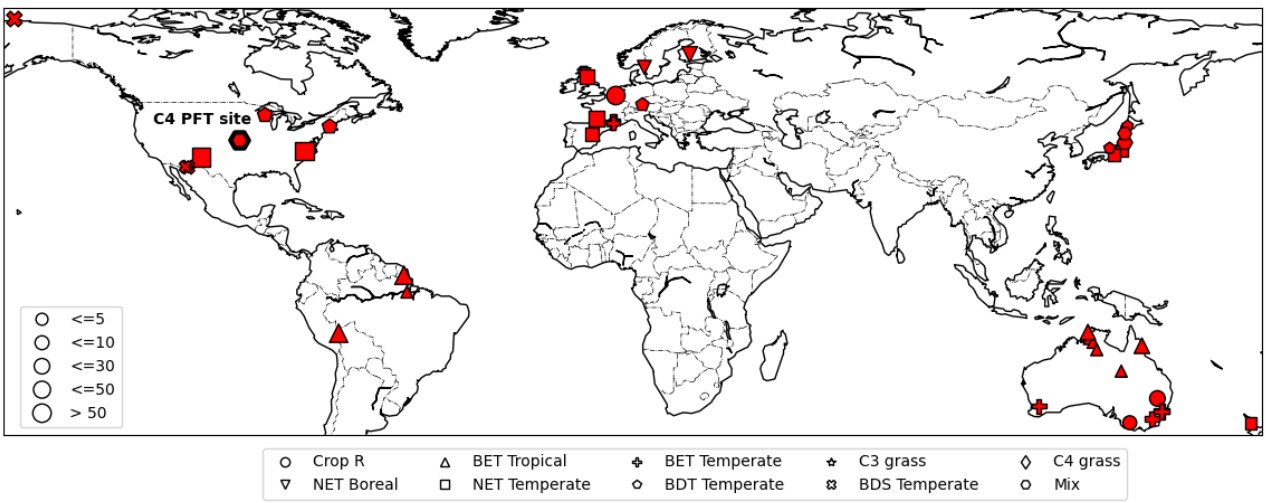

**Figure 3. Map of sites available from the leaf gas exchange database (Lin et al., 2015). Different symbols represent different plant functional types. The C4 site is highlighted by a thick-bordered hexagon. The marker sizes represent the quantity of data available for each site. (Map based on matplotlib basemap (Whitaker, 2013))**

### 2.4.3 CLM4.5 default parameters

CLM4.5 documentation (Oleson et al., 2013) provides reference values for comparison and equations for both target parameters $V_{c,max25}$ and $B$. For $V_{c,max25}$, default values corresponding to each PFT (shown in Table 3) are well documented in CLM4.5 (chapter 8; table 8.1). Similarly, for parameters $B$ and $\beta_t$, their default equations (shown in this work in Appendix B) are provided in the documentation of CLM4.5 as well. We also used other PFT photosynthetic parameters required for $\beta_t$

computations, such as the soil matric potentials for closed stomata, $\psi_c$, and open stomata, $\psi_o$, (see Equations 8,10,12), and the plant root distribution parameters (see Equation 9).

### 2.5 Synthetic data and real data experiments

### 2.5.1 Case 1: Synthetic data

In our synthetic experiments, we assumed values for some parameters to generate synthetic photosynthesis rates which could serve as synthetic training data. Then, we estimated those parameters with NNs while keeping other components unmodified. These experiments were intended to verify the plausibility and efficiency of the differentiable parameter learning framework, and the identifiability of parameters.

In the first synthetic case, "$V_{c,max}$-only", the $\delta_{psn}$ framework was tested for its ability to accurately retrieve a single PFT-dependent parameter, $V_{c,max25}$, using $NN_V$. We used the suggested values for $V_{c,max25}$ from CLM4.5 for different PFTs to calculate the synthetic net photosynthetic rates (synthetic training data). For this case the $\beta_t$ values were kept constant (equal to one) for all datapoints, since we intended to test the retrievability of one parameter.

In the second synthetic case, "$V_{c,max} - B$", we tested simultaneously retrieving both $V_{c,max25}$ and $B$, the latter of which varies spatially for different static attributes. For simplicity, we used only the topsoil layer for this case and excluded the influence of the PFT term; therefore we assumed $B_1 = 0.1 * F_{om,1} + 0.45 * (\%sand_1 + \%clay_1)$ to generate the synthetic data. The plant wilting factor ($w_1$) was then calculated using equation 12 and was fed into equation 9 to compute the soil water stress function ($\beta_t$). Since we were using only the topsoil layer, $\beta_t$ was simplified to ($\beta_t = w_1 r_1$) with a root distribution value for the topsoil 320 layer ($r_1 = 1$). To retrieve $B_1$, we used $NN_{Bi}$ (see Equation 11) but excluded the PFT term since it was not used in synthesizing $B_1$ values.

    For both synthetic runs "$V_{c,max}$-only" and "$V_{c,max} - B$", the MLP models were trained concurrently for all PFTs with several data points for each PFT. Moreover, white noise was added to the synthetic values of $A_n$ with a standard deviation of 5% of 325 the mean value.

### 2.5.2. Case 2: Real data

    Once we confirmed that the model passed the test of the synthetic case (correctly retrieving parameter values which were used to generate the data it was given), it was then applied to a real dataset (Lin et al., 2015) using observation data. This tested whether the model, learning from this dataset for many of the PFTs, could find parameters to better describe photosynthesis 330 data than the literature values. There was no ground truth in this case, so we tested multiple formulations to better understand the impacts of allowing more or less flexibility in the estimation and role of each parameter.

We tested several formulations to estimate either one ($V_{c,max25}$) or two parameters ($V_{c,max25}$ and $B$) at a time. In essence, we compared allowing either one or two of the parameters to be estimated vs. using the default formulation or values from the original model. For $V_{c,max25}$, the default values were those defined in CLM4.5, while for $\beta_t$, the default equations (Appendix B) were used to obtain its values. Altogether, we trained the following models:

**$V_{def}$ + $B_{def}$ :** in this case, $V_{c,max25}$ took the default values from CLM4.5 and $B$ was calculated using the default equations (Appendix B). This was used as a reference case.

**$V_{def}$ + $B$ :** in this formulation, the default $V_{c,max25}$ values from CLM4.5 were used while $B$ was estimated using $NN_{Bi}$.

**$V$ + $B_{def}$ :** in this formulation, $V_{c,max25}$ was estimated using $NN_V$, while $B$ was calculated using the default equations (Appendix B).

**$V$ + $B$ :** in this formulation, we employed both $NN_V$ and $NN_{Bi}$. They were trained concurrently to see if this interfered with parameter retrieval.

Representing a real case, $B_i$ was estimated for the i-th soil layer based on the static attributes for that layer in the four tested model formulations. Thus, $B_i$ varied both horizontally and vertically for each PFT.

Just as in the synthetic case, the MLPs were shared between all sites. All sites were used to calculate one loss function as in typical machine learning tasks, with the hope of ensuring the wide applicability of the MLPs and leveraging the synergy between all sites (Fang et al., 2022). In this way, we also hoped to identify parameters that generalize well in space and are applicable at large scales.

The MLPs employed had three layers: an input layer, one hidden layer, and an output layer. To ensure an output value between 0 and 1 for both $V_{c,max25}$ and $B$ parameterizations, sigmoid activation functions were used for both hidden and output layers. $V_{c,max25}$ was then rescaled to be within a pre-defined range based on literature values of 20 to 150 µmol m$^{-2}$ s$^{-1}$. For the $i$-th soil layer, $B_i$ values were kept between 0 and 1, so with $S_i$ ranging between 0.01 and 1 (see Appendix B), the term $S_i^{-Bi}$ then had a range of 1 to 100. This ensured that the value of $\psi_i$ ranged from $\psi_c$ to $\psi_o$ (see Equation 10).

The quantity of available data posed a limitation and did not permit an extensive hyperparameter tuning experiment with a train/validation/test split. Hence, we employed a "lazy" trial and error process with hyperparameters (learning rates and hidden size) using 70% of the data as training data and 30% as a validation set, just to ensure we had a roughly reasonable-performing

hyperparameter set (see Appendix C). We selected a learning rate of 0.045 and a hidden size equal to the number of inputs (9 for $NN_V$ and 8 for $NN_{Bi}$). We kept these same hyperparameters when we ran 5-fold cross validation with an 80%:20% train:test ratio. In addition, we found that moderately perturbing the hyperparameters resulted in very little change in the performance. This design was necessary considering the practical limits of the available data, even though this study already represents a large-sample study in the domain of ecosystem modeling.

Two different tests were performed with respect to data splitting: temporal holdout and randomized cross-validation --- the former test stresses the models' ability to project into the future, while the latter is the typical experiment run in the literature. Due to the irregularity of measurement dates at each location (as mentioned previously in section 2.4.1), the temporal periods for the training and testing datasets varied by location. In the temporal holdout test, for each PFT in each location, the available dates of measurements were recorded. The oldest 80% of these dates were used for training and the remaining more recent 20% were used for testing. The temporal holdout test was run for both synthetic and real data experiments. For the randomized cross-validation test, as the name implies, the dataset was randomly split into 5 folds (groups) and each time the model was trained on 4 folds (80% of the datapoints) and tested on the 5th fold (20% of the data points). This was done a total of 5 rounds, so that all of the data points were used for testing once. The cross-validation test was run only for the real data experiments.

We then compared the values of $V_{c,max25}$ learned by the V+B model, trained on all data points, against values of $V_{c,max25}$ in other data sources (Kattge et al., 2020; Rogers, 2014), which highlights the variability of these parameters. The TRY database (Kattge et al., 2020) has $V_{c,max25}$ values defined for several species which can be aggregated to get unique values for each PFT (Table 3). Moreover, we compared our $V_{c,max25}$ values to the ones used in different earth system models (Rogers, 2014) for various PFTs, e.g., the Atmosphere-Vegetation Interaction Model "AVIM" (Ji, 1995) and the Biosphere Energy Transfer Hydrology scheme "BETHY" (Knorr and Heimann, 2001). The comparison enabled us to determine whether the inversely determined values were on the same order of magnitude as previously employed in the literature, and were physically plausible. We expected our values for different PFTs to be at least partially correlated with the ones used in the literature, as they were meant to represent the same physical quantity. A complete disagreement or a different order of magnitude would suggest that our values may be not physically representative. Partial discrepancies would highlight any knowledge gaps. We didn't perform a similar comparison between learned and computed $B_i$ values from default equations since the new shape parameter $B_i(soil, PFT)$ (see Equation 11) is different from the original one and has a different range (between 0 and 1).

## 2.6. Statistical metrics

To evaluate different experiments and explore the sensitivity of the results to changing different parameters, we chose four different metrics as shown in table 1, below. The four metrics were root-mean-square error (RMSE), bias, Pearson's correlation (COR), and Nash-Sutcliffe Efficiency (NSE). Both RMSE and bias measure how far the model simulations are from the observations (and thus the ideal value is 0); however, RMSE is the standard deviation of all errors while bias is calculated as

the average. COR measures the linear relationship between both the simulations and the observations, ranging between -1 and
1. NSE measures the relative magnitude of the residual variance relative to the observed data variance (Nash and Sutcliffe, 1970), and has a perfect score of 1. Table 1 below shows the formulations of the four metrics and their possible ranges.

**Table 1. Performance metrics used for evaluation and their possible ranges**

| Metric | Formula | Range |
|--------|---------|-------|
| COR | $$\frac{\sum_{i=1}^{n}(OBS - \overline{OBS})(SIM_i - \overline{SIM})}{\sigma_{OBS}\sigma_{SIM}}$$ | [-1 , 1] |
| RMSE | $$\sqrt{\frac{\sum_{i=1}^{n}(SIM_i - OBS_i)^2}{n}}$$ | [0 , ∞] |
| BIAS | $$\frac{\sum_{i=1}^{n}(SIM_i - OBS_i)^2}{n}$$ | [-∞ , ∞] |
| NSE | $$1 - \frac{\sum_{i=1}^{n}|SIM_i - OBS_i|}{\sum_{i=1}^{n}|OBS_i - \overline{OBS}|}$$ | [-∞ , 1] |

*σ refers to the standard deviation, $\overline{OBS}$ refers to the mean of the observed values, and $\overline{SIM}$ refers to the mean of the simulated values.*

## 3. Results

### 3.1. Results for synthetic data case

The results of the synthetic experiments showed that our workflow successfully recovered the parameters in both the one-parameter case ("$V_{c,max}$-only", Figure 4) and the two-parameter case ("$V_{c,max}$-B", Figure 5). In the one-parameter "$V_{c,max}$-only" case, the recovered parameters agreed with the assumed values almost completely for each PFT (Figure 4a). The model was able to capture the variability in the values of $V_{c,max25}$ for different PFTs, where the values ranged from 100.7 μmol m$^{-1}$ s$^{-1}$ for rainfed crops (defined as Crop R in CLM4.5) to around 50 μmol m$^{-1}$ s$^{-1}$ for C$_4$ grasses (Figure 4a). Moreover, we found nearly complete agreement between the synthetic and recovered net photosynthesis rates ($A_n$) (Figure 4b). This single-parameter case demonstrated that the dPL framework and the posited formulation $V_{c,max25} = NN_V(PFT)$ were functional, but (as intended) did not show the effects of parameter interactions.

With the dual-parameter case, we found a similarly near-complete recovery for $V_{c,max25}$ (Figure 5a) and a near-complete reproduction of simulated photosynthesis (Figure 5d). However, we noticed a negligible amount of scattering with $\beta_t$ (Figure 5c), and to a larger extent, with $B$ (Figure 5b). For all experiments, we verified that the training and test periods were highly

consistent (between green and blue points in the scattered plots). The results indicate that the problem formulation allows for sufficient sensitivity of the net photosynthesis rate with respect to PFT-specific $V_{c,max25}$ and the soil water constraint. In addition, $V_{c,max25}$ and $B$ influence the photosynthesis rate in different ways so that, along with a large dataset with different combinations of moisture conditions and PFTs, they can be identified simultaneously. This forms the basis of the next stage of the work. The soil moisture parameter identifiability was slightly weakened compared to $V_{c,max25}$ because there were more

equations involved between $B$ and $A_n$, and some of them had parameters in the exponential operators, e.g., $\psi_i = \psi_o * S_i^{-Bi}$. Mathematically, such a curve can be flat and the gradients can be small in some ranges of $S$. Mechanistically, $A_n$ can have reduced sensitivity to $B$ under some conditions. Therefore, we do not expect soil properties to be fully identifiable from photosynthesis data, but the general pattern may still be learnable.

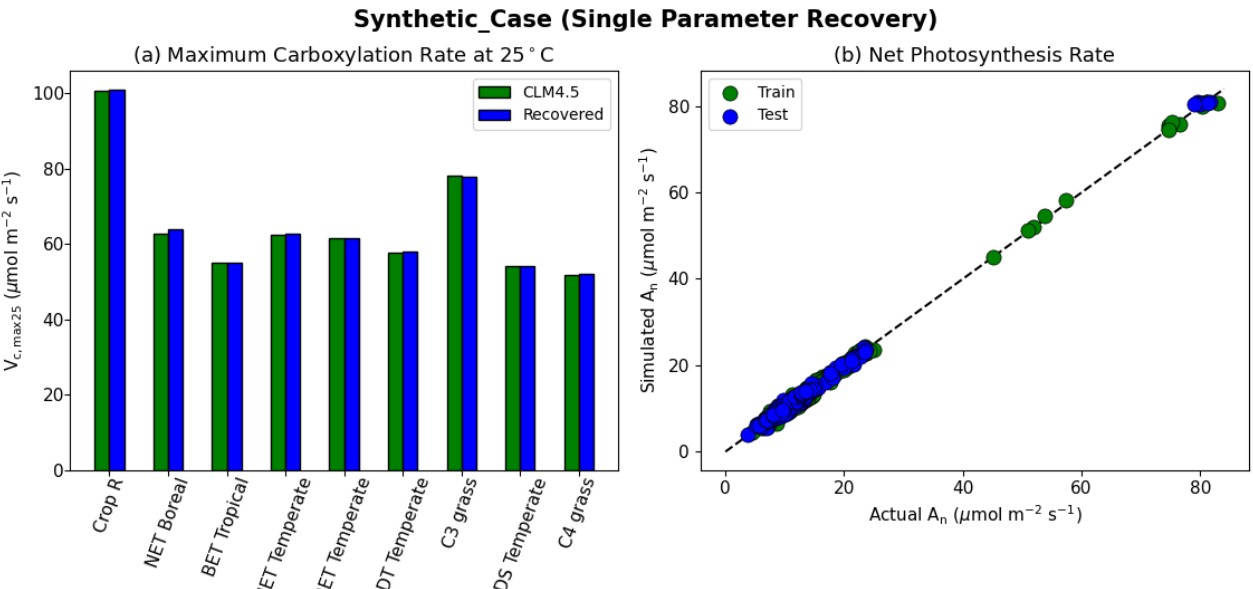

**Figure 4. Single parameter recovery for synthetic data. (a) Comparison of modelled parameter values to literature values by plant functional type (PFT). (b) Actual and modelled net photosynthesis rates for training and testing periods (dashed line indicates the ideal 1:1 relationship).**

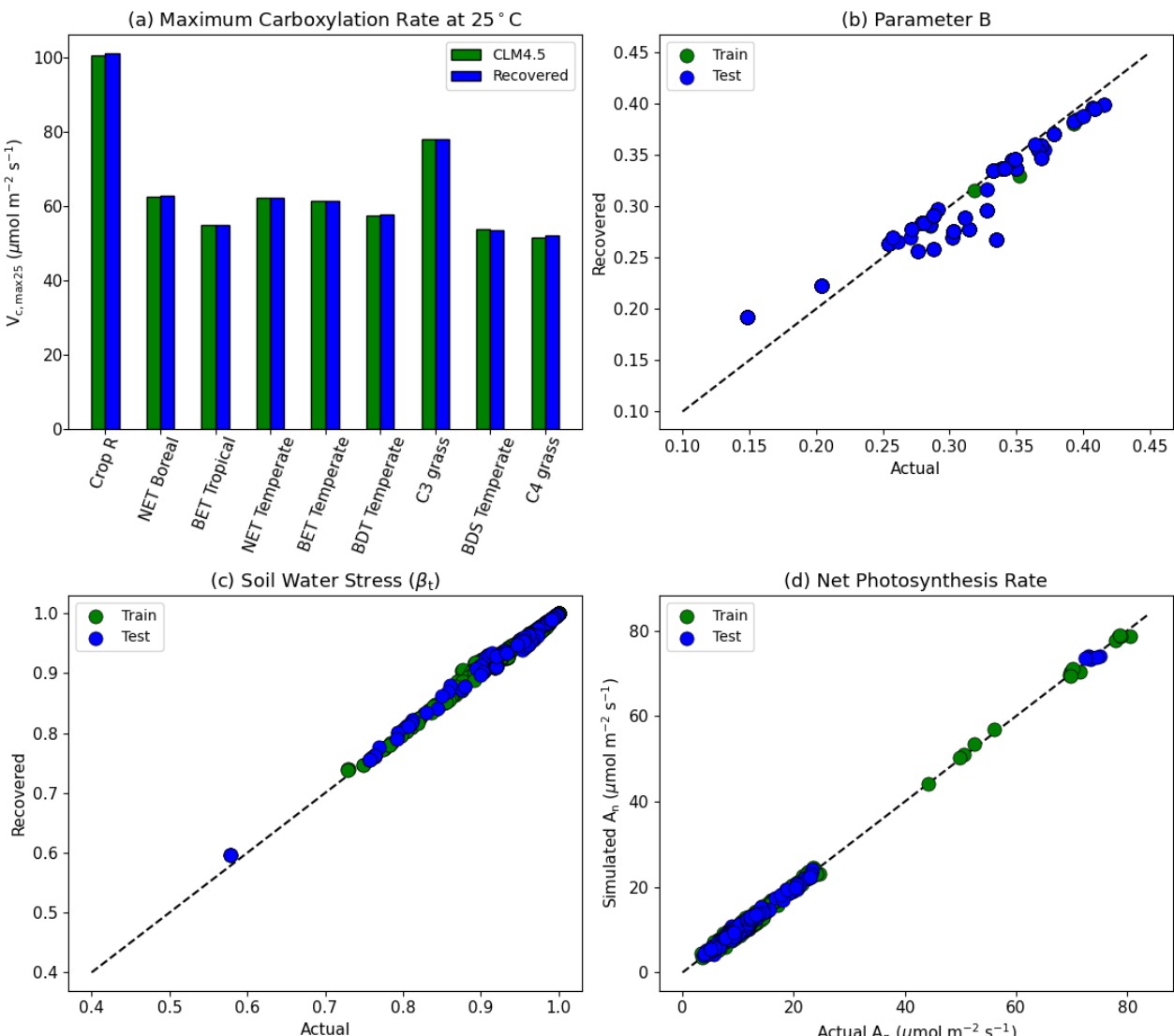

**Figure 5. Dual parameter recovery for synthetic data. (a) Comparison of modeled parameter values to literature values by plant functional type (PFT) estimated using NN$_V$. (b) Actual and modeled parameter values for B, estimated using NN$_{Bi}$. (c) Actual and modeled parameter values for $\beta_t$ for the topsoil layer. (d) Actual and modeled net photosynthesis rates for training and testing periods. Dashed lines in b-d indicate the ideal 1:1 relationship.**

### 3.2. Results for real data case

#### 3.2.1. Comparisons between candidate formulations

In the test cases employing real datasets, the V+B model (employing both NN$_V$ and NN$_{Bi}$) exhibited obvious advantages over the default photosynthesis module in FATES model and the default parameters, as well as the models that learned only one of

the parameters (Table 2). For the temporal holdout test, the default CLM4.5 parameters ($V_{def}$ +$B_{def}$) led to a lower correlation (0.539), a large bias (1.330 μmol m$^{-2}$ s$^{-1}$) and nearly zero NSE (0.001, resulting mainly from the large bias) (Table 2a). In particular, the default values appeared to cause an under-estimation of the net photosynthetic rate ($A_n$) for BET Tropical (Figure 6a-I) and C3 grass (Figure 6a-II) but large over-estimation for the high-photosynthesis data points of $C_4$ plants (Figure 6a-III). After allowing $B$ to be learned ($V_{def}$ +$B$), the correlation for testing slightly increased to (0.551), while the bias remained high (0.724 μmol m$^{-2}$ s$^{-1}$); it seems that the learning of water stress alone did not address the bias. On the other hand, when we only allowed $V_{c,max25}$ to be estimated ($V$+$B_{def}$), the bias greatly increased (-1.653) and the test NSE was slightly decreased to (0.130). Finally, if we allowed both parameters to be learned ($V$+$B$), a decent correlation was obtained (0.757), the bias was the smallest value yet (-0.327 μmol m$^{-2}$ s$^{-1}$), and the test NSE was 0.565, which means the model explained about half of the variance in the observed photosynthesis rate. The remaining error might be attributable to other untuned parameters, processes related to vegetation states which are not considered by the present model. These issues can be potentially further improved in the future using the differentiable modeling paradigm.

A similar behavior in the performance metrics was observed for the five-fold cross-validation test (see Methods; Table 2b). The cross validation test decreased to a great extent the disparity in the metrics' values between training and testing datasets (Table 2b). However, contrary to the temporal holdout test, we found a slight improvement in COR (0.596) and NSE (0.137) when $B$ was learned ($V_{def}$ +$B$), while a much higher boost was found in the metrics when $V_{c,max25}$ was learned ($V$+$B_{def}$). This shows the higher impact of learning $V_{c,max25}$ on the simulation of $A_n$, where the COR and NSE increased to 0.695 and 0.449, respectively, while the bias decreased to -0.478. Similar to the temporal holdout test, the $V$+$B$ model showed the best metrics in comparison to other models, with the lowest RMSE (4.492) and bias (0.022) values, and the highest COR (0.763) and NSE (0.579) values.

**Table 2. Performance metrics for the candidate models for the Lin15 dataset. In all the following, subscript $_{def}$ indicates the default parameter value from CLM4.5 was used, while a parameter lacking $_{def}$ means the parameter was estimated as an output from a neural network (in all cases, V indicates that $V_{c,max25}$ was estimated as a function of PFT using $NN_V$ and B indicates estimation using $NN_{Bi}$). Panel (a) is for the temporal holdout test where the oldest 80% of data points were used for training and the most recent 20% were reserved for testing; panel (b) is for the cross validation (5-fold) test.**

### (a) Temporal holdout test results

| Runs | Corr | | RMSE ($\mu$mol m$^{-2}$ s$^{-1}$) | | Bias ($\mu$mol m$^{-2}$ s$^{-1}$) | | NSE | |
|---|---|---|---|---|---|---|---|---|
| | Train | Test | Train | Test | Train | Test | Train | Test |
| $V_{def}+B_{def}$ | 0.539 | | 6.922 | | 1.330 | | 0.001 | |
| $V_{def}+B$ | 0.607 | 0.551 | 6.469 | 6.220 | 1.357 | 0.724 | 0.142 | 0.140 |
| $V+B_{def}$ | 0.744 | 0.555 | 4.730 | 6.255 | -0.234 | -1.653 | 0.541 | 0.130 |
| $V+B$ | 0.771 | 0.757 | 4.455 | 4.423 | 0.034 | -0.327 | 0.593 | 0.565 |

### (b) Cross Validation (5-fold) test results

| Runs | Corr | | RMSE ($\mu$mol m$^{-2}$ s$^{-1}$) | | Bias ($\mu$mol m$^{-2}$ s$^{-1}$) | | NSE | |
|---|---|---|---|---|---|---|---|---|
| | Train | Test | Train | Test | Train | Test | Train | Test |
| $v_{def}+B_{def}$ | 0.539 | | 6.922 | | 1.330 | | 0.001 | |
| $v_{def}+B$ | 0.597 | 0.596 | 6.419 | 6.434 | 1.242 | 1.242 | 0.140 | 0.137 |
| $v+B_{def}$ | 0.701 | 0.695 | 5.086 | 5.142 | -0.487 | -0.478 | 0.460 | 0.449 |
| $v+B$ | 0.769 | 0.763 | 4.440 | 4.492 | 0.012 | 0.022 | 0.589 | 0.579 |

Consistent with the observations of the synthetic experiments, $V_{c,max25}$ and $B$ impacted $A_n$ in different ways. When $V_{c,max25}$ was not adjusted, the photosynthesis rates simulated for a number of sites in the high-$A_n$ range (most of them $C_4$ plants) had some substantial overestimation, regardless of whether $B$ had learned or default values (Figure 6a). It was only after we also learned $V_{c,max25}$ that these high biases were reduced (Figure 6b). Hence, apparently, the learning reduced the $V_{c,max25}$ for these sites compared to the default values. In contrast, learning $B$ mainly corrected the low bias for low simulated $A_n$ data points

(specifically for BET Tropical, C$_3$ grass, and C$_4$ grass) (Figure 6b). A group of sites with underestimations in $A_n$ has been corrected upward (from yellow to green, bottom points in Figure 6b), due to a correction in the soil parameter $B$. Apparently, the original parameters overestimated the water stress for these sites. Learning both parameters together was also effective in reducing overestimations and underestimations in the simulated $A_n$ for NET Boreal and BDS Temperate respectively. Our results suggest the adjustments to both parameters improved the results, but $V_{c,max25}$ was more impactful, especially in
addressing the bias.

We also notice the different PFTs benefited differently from the parameter learning. For example, BDS Temperate and Crop R did not benefit much (compare red symbols in Figure 6a and green symbols in Figure 6b), BET Tropical and NET Temperate saw moderately improved correlation, while C3 grass and C4 grass saw significant improvements in both correlation and bias.
These observations indicate the parameters (and thus related processes) tuned here ($V_{c,max25}$ and B) have large impacts on C3 and C4 grass while other untuned processes, e.g., vegetation growth and nutrient states, may be contributing to the errors with BDS Temperate and Crop R. C3 plants' improvement is mostly due to learning $B$, as they are more sensitive to drought in the model, while C4 plants' improvement is due to learning $V_{c,max25}$, as they are more resistant to drought but more sensitive to light in the model.

In addition, our test showed that the framework is moderately impacted by long-term nonstationarity, as the temporal test had worse metrics than the cross-validation test (comparing Table 2b with 2a). The absolute value of the bias increased from 0.022 in the cross validation test to 0.327 in the temporal test. This suggests the current model (and perhaps the training data) still has some limitations with representing long-term changes. Possible reasons may include $CO_2$ fertilization and its impact on
water use efficiency or differences in the state of plants, as this factor is not included in our present parameterization. In the future, these issues could be addressed by assembling a more long-term training dataset (the Lin15 dataset has data ranging from 1991 to 2013), as well as improving the parameterization and physics of the model.

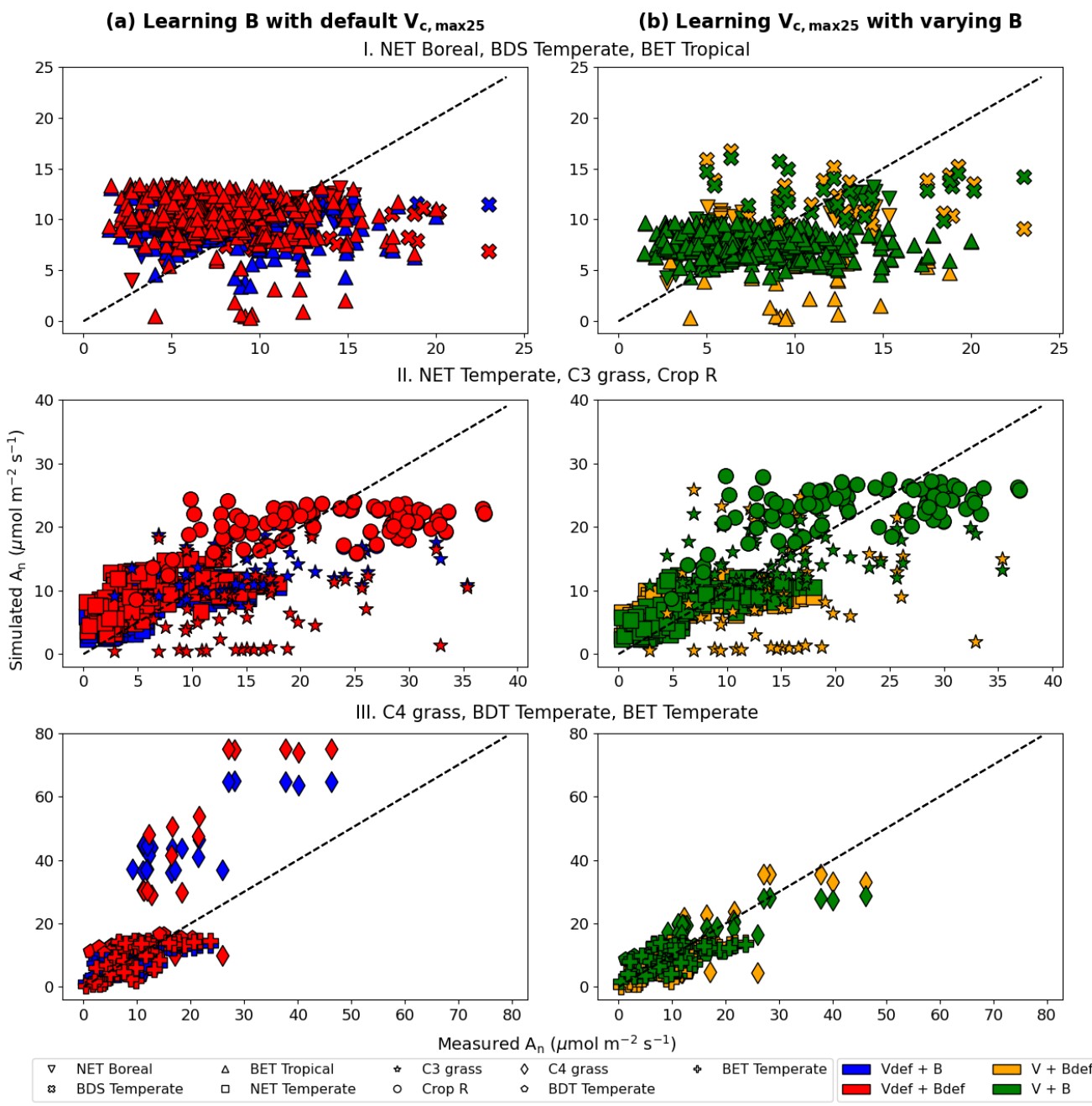

**Figure 6. Comparisons of photosynthesis model calibration. Comparing impacts of default and learned parameters by plotting observed vs. simulated $A_n$ (net photosynthetic rate) values calculated using different candidate models (described by which parameter definitions they use). (a) Impact of learning B with default $V_{c,max25}$. (b) Impact of learning $V_{c,max25}$ with varying B (either learned alongside V in V+B, or defined by the default equations in CLM4.5. The colors represent the results from the four different models, the shapes indicate the plant functional type (PFT)**

**groups, and the dotted line in each panel indicates the ideal 1:1 relationship. Subscript "def" indicates that the variable was calculated using the default definitions in CLM4.5, while lack of this subscript indicates that the parameter was learned using a NN. Scatter plots were created using the test results from each of the 5 folds of the cross-validation test. For better illustration, only 3 PFTs are placed in a panel, as indicated by the panel titles. Comparing symbols in the same panel gives insights about the role of estimating B, while comparing left and right panels gives insights about the role of estimating $V_{c,max25}$.**

### 3.2.2. Recovered parameters

Even though we did not prescribe the values of $V_{c,max25}$, the training on the dataset converged to parameter values that were partially correlated with, yet still substantially different from, the literature values, and were on the same order of magnitude (Figure 7). The default $V_{c,max25}$ values came from in-situ measurements at a limited number of sites, while our values came from learning from a moderately large dataset (essentially an inversion process limited to the model structure). The fact that they agreed with each other in the main pattern suggests both have merit, and that the learning process captured fundamental physics. The upper half of Figure 7b saw a high correlation, but $V_{c,max25}$ values for the V+B model were uniformly higher than the CLM4.5 defaults, especially for the NET Boreal PFT. The correlation was lower toward the lower half of Figure 7b (where $V_{c,max25}$ from CLM4.5 was lower than 65 $\mu mol\ m^{-2}\ s^{-1}$) – the learned $V_{c,max25}$ had a larger variability. In particular, the learned $V_{c,max25}$ (V+B) for $C_4$ grass is much lower than the default, which could be attributed to species-level variability and the fact that the dataset contains very limited sites with $C_4$ plants. Hence, we do not argue that the values learned here would be applicable globally to other $C_4$ grasses. It seems that the inter-PFT variability in $V_{c,max25}$ was previously under-represented by the CLM4.5 default parameter values (BET Tropical, BET Temperate, BDS Temperate, $C_4$ grass), and the learning process used here enhanced the variability. Moreover, we note that for either Crop R, BET Tropical, BET/BDT Temperate or $C_4$ grass, the influence of learning B on $V_{c,max25}$ was mostly small ($V_{c,max25}$ from V+B and V+B$_{def}$ models were mostly similar) and thus the interactions between these two parameters were not significant. The overall results showcase the ability of the algorithm to adapt to data and reveal parameter interactions.

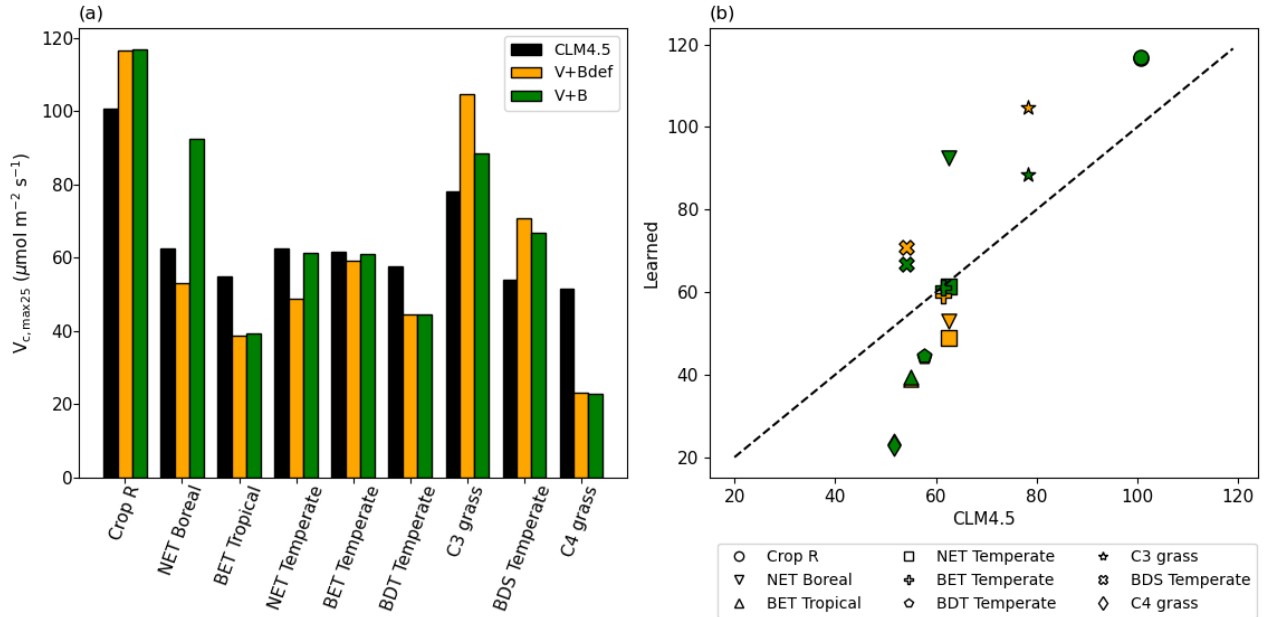

**Figure 7. Parameter recovery for real data. (a) Comparison of modelled parameter values to literature values of $V_{c,max25}$ by plant functional type (PFT). (b) Actual and modelled $V_{c,max25}$ values plotted by PFT (dashed line indicates the 1:1 ideal relationship). In this figure, both "V+Bdef" and "V+B" models were trained using the whole dataset.**

In our interpretation, the learned values represent a more "fine-tuned" version of the literature $V_{c,max25}$ values, with the interference from soil water stress disentangled. The magnitude and ranking for PFTs remained similar to the literature values, but the results were improved in different ways for different PFTs. The V+B model obtained lower $V_{c,max25}$ for $C_4$ grasses, addressing the significant overestimation bias for these sites, which we noted in Figure 6a-III. Due to their different photosynthesis pathway, $C_4$ plants have the lowest learned $V_{c,max25}$, but overall the highest net photosynthesis rates, which were not heavily influenced by the choice of the $B$ parameter. For $C_3$ grasses, V+B only slightly increased $V_{c,max25}$ compared to the default CLM4.5 values, which addressed the low bias noticeable in Figure 6b-II. The default soil parameterization for $C_3$ grass sites seemed somewhat deficient as soil water stress accounted for the other parts of variance in net photosynthesis, as demonstrated by the comparison between V+B and V+B$_{def}$ models in Figures 6b-II and 7b for $C_3$ grass.

We compared our learned $V_{c,max25}$ values (Table 3 and Figure 8) with values from other earth system models (ESMs) and with some observatory values in the TRY database (Kattge et al., 2020; Rogers, 2014). The learned $V_{c,max25}$ values are higher than those of the TRY database for most PFT classes except for BDT Temperate and BDS Temperate; however, they are within the range of values used in other ESMs except for relatively higher estimations for Crop R, NET Boreal, and $C_3$ grasses. On the scale of ESMs, several values for $V_{c,max25}$ are adopted by those models. We computed the correlation coefficient between our learned $V_{c,max25}$ values and reference values from other ESMs and the TRY database, finding high correlations (except for the

AVIM model) between the learned and reference $V_{c,max25}$ values for CLM4.5 (0.844), BETHY (0.900), and the TRY database (0.699). For instance, $V_{c,max25}$ for $C_4$ grasses is 25 and 20 ($\mu$mol m$^{-2}$ s$^{-1}$) in the AVIM and BETHY models, respectively (Table 3). These values agree with the learned $V_{c,max25}$ by the V+B model of 22.90 ($\mu$mol m$^{-2}$ s$^{-1}$), whereas much higher values were found to be adopted for $C_4$ grasses with 60 ($\mu$mol m$^{-2}$ s$^{-1}$) being used in the Biogeochemical cycles model "BiomeBGC'' as reported in Rogers (2014), and 51.6 ($\mu$mol m$^{-2}$ s$^{-1}$) in CLM4.5. $V_{c,max25}$ from the V+B model and TRY database are similar for BET Tropical and BDT Temperate. For BDS Temperate, the learned $V_{c,max25}$ was lower than that in TRY by ~20 ($\mu$mol m$^{-2}$ s$^{-1}$), but similar values were used by BETHY and lower values were used by AVIM. For Crop R, NET Boreal, NET Temperate, and BET Temperate, the learned $V_{c,max25}$ values were all ~20 – 30 ($\mu$mol m$^{-2}$ s$^{-1}$) higher than those of the TRY database, but (except for NET Boreal) similar values have been used by AVIM or BETHY. Both the learned (V+B) and the observed (TRY database) $V_{c,max25}$ values show a similar pattern with the lowest $V_{c,max25}$ for BET Tropical and a high value assigned for Crop-R.

Table 3. $V_{c,max25}$ simulated by the V+B model versus observed values from the TRY database (with partial overlap in species with the Lin15 dataset – the percentage of overlap is provided in the table), and used in different earth system models such as CLM4.5, Atmosphere-Vegetation Interaction Model "AVIM", and the Biosphere Energy Transfer Hydrology scheme "BETHY".

| PFT | CLM4.5 | AVIM | BETHY | V+B (ours) | TRY (mean / % species overlap) | TRY (std) |
|---|---|---|---|---|---|---|
| **Crop R** | 100.7 | 55 | 90 | 116.83 | 84.20 / 60.0% | 2.19 |
| **NET Boreal** | 62.6 | 58 | 58 | 92.58 | 62.90 / 100.0% | 22.53 |
| **BET Tropical** | 55 | 64 | 28/36 | 39.37 | 33.14 / 86.5% | 14.09 |
| **NET Temperate** | 62.5 | 60 | 58 | 61.27 | 44.33 / 50.0% | 7.13 |
| **BET Temperate** | 61.5 | 68 | 58 | 61.10 | 37.73 / 26.7% | 0.27 |
| **BDT Temperate** | 57.7 | 60 | 54 | 44.68 | 50.27 / 50.0% | 21.62 |
| **C$_3$ grass** | 78.2 | 55/40 | 71 | 88.58 | - | - |
| **BDS Temperate** | 54 | 52 | 65 | 66.70 | 87.61/ 58.3% | 11.77 |
| **C$_4$ grass** | 51.6 | 25 | 20 | 22.90 (limited data points) | - | - |

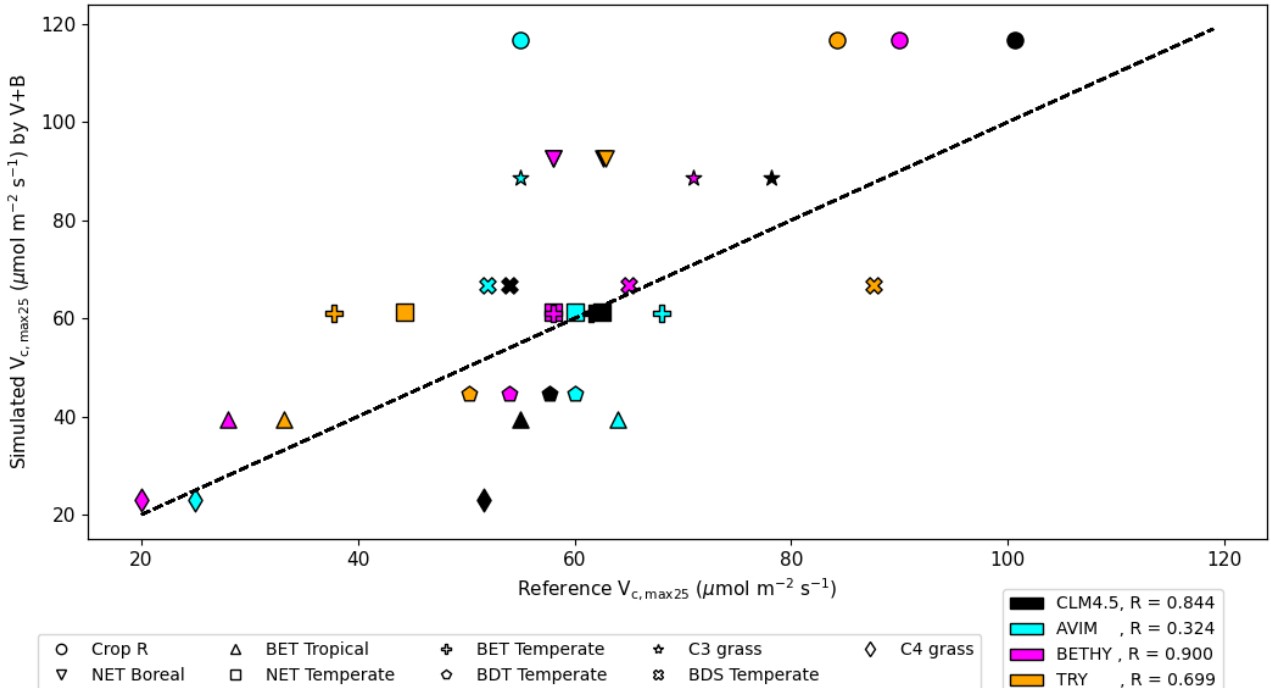

**Figure 8. The correlation between the $V_{c,max25}$ values estimated by the V+B model on** the **y-axis versus $V_{c,max25}$ values from CLM4.5 (black markers), AVIM (cyan markers), BETHY (magenta markers), and the TRY database (orange markers). Different marker shapes represent different PFTs, while different colours represent different reference sources for $V_{c,max25}$ per PFT. For the TRY database, we don't have values for C3 grass and C4 grass due to the lack of overlap in species between** the **TRY database and our dataset for those two PFTs. The dashed line indicates a 1:1 relationship.**

## 4. Discussion

As an initial exploration of the potential of the emerging differentiable computing paradigm (a genre of physics-informed machine learning) (Shen et al., 2023) for application to ecosystem modeling, our work showed promise but also had many limitations, as the goal was not to produce the best-performing photosynthetic model. We restricted our parameter sets to be dependent on PFT, whereas it is known that within-PFT variation can be significant, and parameters could also be determined on the trait level as well as by multiple environmental factors. Our model did not consider the effects of memory, e.g., rainfall

in previous days, and the states of the vegetation, e.g., carbon stored in the canopy or carbon: nitrogen ratios in the canopy. The soil moisture data comes from the ERA5 dataset, which, based on comparisons to in-situ data, would be outperformed by ML-based soil moisture predictions (Fang et al., 2017; Liu et al., 2022a, b), but we used it due to its global seamless coverage and availability for multiple soil depths. This work also only modified the parameterization scheme and did not learn model structures. Recently, development in differentiable hydrologic models allows learning parts of the model using neural networks

(Feng et al., 2022a, b). In summary, we believe there is still lots of room for improving model quality, but at some point we will likely run into the limits of measurements (aleatoric uncertainty) or data availability (epistemic uncertainty) (Hüllermeier and Waegeman, 2021). Future effort can harness deep networks to establish reference levels as a measure of the data uncertainty (Feng et al., 2022a).

This work appears to be the first evaluation of the Lin15 dataset, and, as such, it establishes a reference level to which future studies can compare. The current dataset may still have limitations in that the number of sites for $C_4$ plants is small and does not allow ample testing. Some geoscientific domains have well-known benchmark datasets, e.g., the CAMELS dataset in hydrology (Feng et al., 2020). Having such a common (and hopefully large) benchmark dataset allows better model structures to be rapidly discovered and is highly beneficial to the growth of the community (Shen et al., 2018). Related to the limits of

measurement errors discussed above, multiple deep-learning-based studies have explored the approximate limit of data error (or best achievable model) of CAMELS and that knowledge has been appreciated by the community (Feng et al., 2021). Moreover, deep learning methods benefit from data synergy effects (Fang et al., 2022), where more sites and more diverse data lead to a more robust model and better performance for each site.

Although applying the dPL framework improved the parameters to an extent, the model still has similar structural limitations as other Farquhar-type models. We didn't test the model's ability to capture the seasonality of the net photosynthetic rate due the limited site-level temporal data. The seasonal behavior of the model is expected to be similar to other Farquhar models as here we only learned static parameters. Further improvement likely will need to consider vegetation growth. Also, this study didn't cover spatial generalization since we don't present results for spatial tests or based on site-level comparison. To improve

spatial generalization may require further changes in the model, dynamical parameters, or using other error mitigation

approaches (Feng et al., 2021, 2022b; Ma et al., 2021a). This was not within the scope for this study; however, it will be considered for future work.

We would like to highlight that such parameterizations are suitable to the target and forcing datasets used in training (which is still the most representative accessible dataset) and are related to the process-based model employed. The dataset may have limitations related to the consistency in the measurement approach, and there may be errors in the forcing data, or imperfections in model structure. The model performance may also vary based on different forcing data and inputs used.

## 5. Conclusions

In this study, we proposed a novel differentiable ecosystem modeling framework that uses neural networks as a parameterization scheme to support a process-based ecosystem model (FATES). Training coupled neural networks was not previously possible without differentiable programming, and it allows us to approximate complex, *a priori* unknown mapping relationships between plant functional types, landscape characteristics, and physical parameters. The photosynthesis module was treated as a system of nonlinear equations, and, like other such systems, could be solved efficiently and in a massively parallel fashion on graphical processing units (GPUs) by our differentiable framework. $V_{c,max25}$ and a soil water parameter ($B$) could be simultaneously identified in our synthetic experiments, because they played different roles in the model.

Compared to purely data-driven machine learning approaches, the differentiable programming framework provides physically meaningful variables and can be used to learn relationships from big data. Via training on a global dataset, we found $V_{c,max25}$ values for global sites that correlate with the values in the literature, but produce more accurate net photosynthesis rates. It is noteworthy that these values were identified without any supervision from experts other than the preparation of the training dataset and the model. We conclude that $V_{c,max25}$ has a larger impact on photosynthesis than the soil water stress parameter, but both can be useful in tuning model responses, with varied impacts on different plants, and their default values were not optimal. Not only is differentiable modeling able to improve simulation quality and provide model parameterization, it also allows us to modify model structure and ask questions regarding unclear parts of the model in the future. There is significant room for this framework to improve and expand to other ecosystem modeling applications.

## 6. Appendices

### Appendix A

### The System of nonlinear equations

The FATES photosynthesis module is based on the classical Farquhar model for $C_3$ plants (Farquhar et al., 1980), which
calculates the photosynthetic rate based on carbon fluxes under different limitations. For $C_4$ plants, it uses the Collatz model (Collatz et al., 1992). Both models assume that the gross photosynthetic rate is affected by the maximum rate of carboxylation and is limited by the concentration of RuBP carboxylase (Rubisco) ($A_c$, see Equation A1), light and electron transport ($A_j$, see Equation A2), and the concentration of PEP carboxylase enzyme in $C_4$ plants ($A_p$, see Equation A3). $A_c$, $A_j$, and $A_p$ are calculated as:

$$A_c = \begin{cases} \dfrac{V_{c,max} \times (C_i - \Gamma_*)}{C_i + K_c \left(1 + \dfrac{O_i}{K_o}\right)} & \text{for C3 plants} \\ V_{cmax} & \text{for C4 plants} \end{cases} \tag{A1}$$

$$A_j = \begin{cases} \dfrac{J_x \times (C_i - \Gamma_*)}{4C_i + 8\Gamma_*} & \text{for C3 plants} \\ \alpha(4.6\varphi) & \text{for C4 plants} \end{cases} \tag{A2}$$

$$A_P = \begin{cases} K_p \dfrac{C_i}{P_{atm}} & \text{for C4 plants} \end{cases} \tag{A3}$$

where $V_{c,max}$ is the maximum carboxylation rate, $C_i$ is the intercellular leaf $CO_2$ pressure (nonlinear system output), $\Gamma_*$ is the $CO_2$ compensation point, $K_c$ and $K_o$ are the Michaelis-Menten constants, $O_i$ is the $O_2$ partial pressure (calculated as 20% of the atmospheric pressure), $J_x$ is the electron transport rate (see Equations A4 and A5), $\alpha$ is the quantum efficiency (0.05 mol $CO_2$ mol$^{-1}$ photon), $\varphi$ is the photosynthetically active radiation (available in Lin15), $K_p$ is the initial slope of $C_4$ $CO_2$ response curve, and $P_{atm}$ is the atmospheric pressure (available in Lin15).

$\Gamma_*$, $K_c$, and $K_o$ are the scaled parameters based on leaf temperature ($T_v$) calculated using their standardized values at 25°C which are $\Gamma_{*25} = 42.75 \times 10^{-6} P_{atm}$, $K_{c,25} = 404.9 \times 10^{-6} P_{atm}$, and $K_{o,25} = 278.4 \times 10^{-3} P_{atm}$, multiplied by the temperature response functions defined in chapter 9.0 in CLM5.0 (Lawrence et al., 2019).

$J_x$ is given by the minimum root of the following quadratic equation:

$$\theta_{PSII} J_x^2 - (I_{PSII} + J_{max}) J_x + I_{PSII} J_{max} = 0 \tag{A4}$$

where $J_{max}$ is the maximum electron transport rate, $\theta_{PSII}$ is an empirical curvature for the electron transport rate (0.7) and $I_{PSII}$ is the light utilized in electron transport calculated using a quantum yield parameter ($\Phi_{PSII} = 0.85$) as:

$$I_{PSII} = 0.5\Phi_{PSII}(4.6\,\varphi) \tag{A5}$$

The three biophysical rates $V_{c,max}$, $J_{max}$, and $K_p$ along with the plant respiration rate ($R_d$), adjusted for $T_v$ are calculated using their standardized values at 25°C multiplied by temperature response functions defined in chapter 9.0 in CLM5.0 (Lawrence et al., 2019). $V_{c,max}$ is also adjusted for the soil water availability by multiplying it with the soil water stress function($\beta_t$). In our case, $V_{c,max25}$ is either the default value provided in CLM4.5 or is learned by a neural network, which then is used to calculate other standardized biophysical rates as:

$$J_{max25} = 1.67\,V_{c,max25} \tag{A6}$$

$$R_{d25} = \begin{cases} 0.015\,V_{c,max25} & \text{for C3 plants} \\ 0.025\,V_{c,max25} & \text{for C4 plants} \end{cases} \tag{A7}$$

$$K_{p25} = \{20000\,V_{c,max25} \text{ for C4 plants}\} \tag{A8}$$

The gross photosynthetic rate ($A$) is then calculated by solving for the minimum root of the quadratic equations:

$$\theta_{cj}A_i^2 - (A_c + A_j)A_i + A_cA_j = 0 \tag{A9}$$

$$\theta_{ip}A^2 - (A_i + A_p)A + A_iA_p = 0 \tag{A10}$$

where $A_i$ is an intermediate co-limited photosynthetic rate calculated using the empirical curvature parameter ($\theta_{cj} = 0.999$). Using $A_i$ and $A_p$, and the empirical curvature parameter ($\theta_{ip} = 0.999$), the gross rate ($A$) is given by the smaller root of equation A10. The net photosynthetic rate ($A_n$) is:

$$A_n = A - R_d \tag{A11}$$

Then using $A_n$, the $CO_2$ partial pressure at the leaf surface ($C_s$) is calculated as:

$$C_s = C_a - \frac{1.4\,P_{atm}A_n}{g_b} \geq 1.0e - 6 \tag{A12}$$

where $C_a$ is $CO_2$ partial pressure near the leaf surface (estimated using observations of the leaf surface $CO_2$ concentrations in Lin15) and $g_b$ is the leaf boundary layer conductance, which was available in Lin15 for some sites and the missing values were filled using the mean $g_b$ of the whole dataset. The stomatal conductance ($g_s$) is then given by the maximum root of the quadratic equation:

$$g_s^2 + bg_s + c = 0 \tag{A13}$$

where $b$, $c$, and $d$ are functions in some PFT-dependent parameters:

$$b = -\left(2(g_o + d) + \frac{(g_1 d)^2}{g_b \times vpd}\right) \tag{A14}$$

$$c = g_o^2 + \left(2g_o + d\left(1 - \frac{g_1^2}{vpd}\right)\right)d \tag{A15}$$

$$d = \frac{1.6\, A_n}{C_s / P_{atm}} \tag{A16}$$

$g_o$, the water stressed minimum stomatal conductance, is calculated as the multiplication of $\beta_t$ and the unstressed minimum

stomatal conductance (10000 µmol m$^{-2}$ s$^{-1}$ for C$_3$, 40000 µmol m$^{-2}$ s$^{-1}$ for C$_4$). $g_1$, the slope of the Medlyn stomatal conductance model (Medlyn et al., 2011) is a PFT specific parameter defined in CLM5.0 (Lawrence et al., 2019). *vpd*, the vapor pressure deficit, was available in Lin15. Finally, $C_i$, is related to $A_n$ using $C_a$, $P_{atm}$, $g_s$, and $g_b$ as the following:

$$C_i = C_a - A_n P_{atm} \frac{(1.4 g_s + 1.6 g_b)}{(g_s \times g_b)} \tag{A17}$$

## Appendix B

**Computations of btran ($\beta_t$) in CLM4.5**

$\beta_t$ is calculated by aggregating the plant wilting factor ($w_i$) and plant root distribution ($r_i$) across different soil different layers as:

$$\beta_t = \sum_i w_i r_i \tag{B1}$$

The plant wilting factor ($w_i$) for soil layer i is mainly dependent on the soil water potential $\psi_i$ and other PFT-dependent parameters such as the soil matric potentials for closed stomata $\psi_c$ and open stomata $\psi_o$, which represent the soil water potentials when stomata are fully closed and fully open, respectively. The factor $w_i$ is also dependent on other factors like the temperature of the soil layer ($T_i$) relative to the freezing temperature ($T_f$), the volumetric liquid water ($\theta_{liq,i}$) and ice ($\theta_{ice,i}$) contents, and the volumetric water content at saturation ($\theta_{sat,i}$).

$$w_i = \begin{cases} \dfrac{\Psi_c - \Psi_i}{\Psi_c - \Psi_o} \left[ \dfrac{\theta_{sat,i} - \theta_{ice\,i}}{\theta_{sat,i}} \right] \leq 1 & ; T_i > T_f - 2 \text{ and } \theta_{liq,i} > 0 \\ 0 & ; T_i \leq T_f - 2 \text{ or } \theta_{liq,i} \leq 0 \end{cases} \tag{B2}$$

The soil matric potential $\psi_i$ is calculated using a power-law formulation:

$$\Psi_i = \Psi_{sat,i} \times S_i^{-B_i} \geq \Psi_c \tag{B3}$$

where $\psi_{sat,i}$ is the saturated soil matric potential, $S_i$ is the soil wetness, and $B_i$ is the Clapp and Hornberger parameter, all defined for a specific soil layer ($i$). Different soil attributes such as percentages of sand (%$sand_i$) and clay (%$clay_i$), fraction of organic matter ($F_{om,i}$), and soil moisture ($\theta_{liq,i}$) are used in computing $\psi_{sat,i}$, $S_i$, and $B_i$. $\psi_{sat,i}$ is calculated as:

$$\Psi_{sat,i} = (1 - F_{om,i}) \times \Psi_{sat,min,i} + F_{om,i} \times \Psi_{sat,om} \tag{B4}$$

where $\psi_{sat,om}$ is the saturated organic matter matric potential (-10.3 mm (Letts et al., 2000)) and $\psi_{sat,min,i}$ is the saturated mineral soil matric potential calculated using %$sand_i$ as:

$$\Psi_{sat,min,i} = -10.0 \times 10^{1.88 - 0.0131 \times (\%sand)_i} \tag{B5}$$

The soil wetness ($S_i$) is calculated using the volumetric contents $\theta_{liq,i}$, $\theta_{ice,i}$, and $\theta_{sat,i}$ as:

$$S_i = \dfrac{\theta_{liq,i}}{\theta_{sat,i} - \theta_{ice,i}}, 0.01 \leq S \leq 1 \tag{B6}$$

where $\theta_{sat,i}$ for a soil layer is:

$$\theta_{sat,i} = (1 - F_{om,i}) \times \theta_{sat,min,i} + F_{om,i} \times \theta_{sat,om} \tag{B7}$$

$\theta_{sat,om}$ is the porosity of the organic matter (0.9 (Letts et al., 2000; Farouki, 1981)), while the porosity of the mineral soil ($\theta_{sat,min}$) using *%sand* is:

$$\theta_{sat,min,i} = 0.489 - 0.00126 \times (\%sand)_i \tag{B8}$$

Similar to $\psi_{sat,i}$ and $\theta_{sat,i}$ (see Equations B4 and B7), the parameter $B_i$ is calculated as:

$$B_i = (1 - F_{om,i}) \times B_{min,i} + F_{om,i} \times B_{om} \tag{B9}$$

where $B_{om}$ is the parameter for organic matter (2.7 (Letts et al., 2000)) while $B_{min,i}$ the parameter for mineral soil is:

$$B_{min,i} = 2.91 + 0.159 \times (\%clay)_i \tag{B10}$$

**Appendix C**

**Table C1. V+B model formulation performance for different sizes of NN$_{Bi}$ with 80%:20% train: test split ratio**

| | Corr | | RMSE (μmol m-2 s-1) | | Bias (μmol m-2 s-1) | | NSE | | |
|---|---|---|---|---|---|---|---|---|---|
| | Train | Test | Train | Test | Train | Test | Train | Test | |
| **V+B** | 0.7713 | 0.7570 | 4.4561 | 4.4226 | 0.0090 | -0.3530 | 0.5928 | 0.5651 | NN$_{Bi}$[8,6,1] |
| | 0.7716 | 0.7560 | 4.4544 | 4.4306 | 0.0356 | -0.3210 | 0.5931 | 0.5635 | NN$_{Bi}$ [8,7,1] |
| | 0.7714 | 0.7567 | 4.4553 | 4.4228 | 0.0345 | -0.3273 | 0.5929 | 0.5650 | NN$_{Bi}$ [8,8,1] |
| | 0.7715 | 0.7558 | 4.4542 | 4.4314 | 0.0245 | -0.3317 | 0.5931 | 0.5633 | NN$_{Bi}$ [8,9,1] |
| | 0.7703 | 0.7591 | 4.4703 | 4.4079 | 0.0259 | -0.3427 | 0.5902 | 0.5679 | NN$_{Bi}$ [8,8,8,1] |

## 7. Code Availability

Code example to demonstrate the differentiable model and its training process is available at

705 (https://github.com/hydroPKDN/diffEcosys/ ) and citable via (https://doi.org/10.5281/zenodo.7975564).

## 8. Data Availability

Datasets used in the model are publicly available from the sources cited in this paper.

## 9. Author Contribution

DA implemented the numerical models, ran experiments, and produced the figures. DA and CS completed the initial

manuscript. CS and CX conceived the study. CX, FH, KL and CS edited the manuscript. JL contributed to data downloading and processing. CS implemented the parallel Newton solver for nonlinear system on PyTorch, while DA implemented the Julia version with assistance from AJ and CR.

## 10. Competing Interests

KL and CS have financial interests in HydroSapient, Inc., a company which could potentially benefit from the results of this

research. This interest has been reviewed by the University in accordance with its Individual Conflict of Interest policy, for the purpose of maintaining the objectivity and the integrity of research at The Pennsylvania State University.

## 11. Acknowledgements

This work was supported by the U.S. Department of Energy, Office of Science under award DE-SC0021979. CX was supported by the Department of Energy, Office of Science project Next Generation Ecosystem Experiment-Tropics (NGEE-Tropics).

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
