# Peer review of "A differentiable, physics-informed ecosystem modeling and learning framework for large-scale inverse problems: Demonstration with photosynthesis simulations"

_Biogeosciences, 2022_

## Author Comment (AC1)

The authors of the manuscript 'A differentiable ecosystem modeling framework for large-scale inverse problems: demonstration with photosynthesis simulations' describe the application of the 'differentiable parameter learning'(dPL) framework to the photosynthesis module of FATES model. The framework, and concept, overcomes extrapolation limitations from site-by-site calibration approaches and allows leveraging information content in large-scale datasets towards a global parameterization of photosynthesis models. Neither the concept (Tsai et al., Nature Communications, https://www.nature.com/articles/s41467-021-26107-z, 2021; Bao et al., Authorea, https://www.authorea.com/doi/full/10.1002/essoar.10512186.3, 2022) nor the dPL framework (Tsai et al., 2021; Feng et al., 2022ab) are new. However, the framework is used in the FATES model for the first time and the results would be of interest for further model development, but also to the scientific community at large.

At this point, the experiment focuses on inverting two parameters, $V_{cmax25}$ and B, resulting in that the accuracy of the simulated net photosynthesis rate being slightly improved. The main concerns at this stage relate to apparently incorrect formulations of some key equations, to issues about the validation strategy, to the fact that the forcing data and the experiments are not described sufficiently, challenging the acceptance of the study, while hampering any reproducibility efforts. Please see below for details.

Thank you for your comments! We are preparing a fuller response to your comment, but here are some rapid response to two of your comments.

We were indeed following our previous differentiable parameter learning paradigm which were applied in hydrology (Tsai et al., 2021; Feng et al., 2022), as noted in the manuscript, but this is a novel use in ecosystem modeling. We could not have noticed Bao et al., 2022 as it went online after ours did and seems to be undergoing review. Upon some examination, we believe the basic modules are very different. They are using a light-use-efficiency approach and predicted GPP, while our paper focused on photosynthesis using a Farquhar-type model. Hence we don't think there is much overlap between the two.

Major comments:
1. Two key equations are incorrect in the paper:
1) line 140: equation 5, $C_i = C_a - A_n * P_{atm} * (1.4 g_s + 1.6 g_b)/(g_s + g_b)$;
2) line 505: equation A1, $A_c = V_{cmax} * (C_i - \Gamma_*)/(C_i + K_c * (1 + K_o/O_i))$.
According to the user guide of the FATES model (https://fates-users-guide.readthedocs.io/projects/tech-doc/en/latest/fates_tech_note.html#fundamental photosynthetic-physiology-theory), the equations should be:
1) $C_i = C_a - A_n * P_{atm} * (1.4 g_s + 1.6 g_b)/(g_s * g_b)$;
2) $A_c = V_{cmax} * (C_i - \Gamma_*)/(C_i + K_c * (1 + O_i/K_o))$.
Since the FATES model is reimplemented in Julia and PyTorch by the authors, the codes might be also wrong. If so, the unit of $C_i$ will be incorrect, leading to errors in the inversion of

$V_{cmax25}$ and B. The wrong computation of the effective Michaelis-Menten coefficient (=$K_c*(1+O_i/K_o)$) might only have a slight effect if the temperature is close to 25°C, but should be concerned if the temperature is too low or high (and I do see some points with low leaf temperature in the 'Lin15' database). Thus, I have doubts about the current results and relevant analysis.

Regarding the equations --- we were cautious to adhere to the original FATES equations before implementing it on PyTorch or Julia. Unfortunately, we **realized there were some typos in the manuscript** in line 140 and line 505 in the paper which will then be modified. However, we used the correct equations in our differentiable model as the following:

$C_i=C_a-A_n*P_{atm}*(1.4g_s+1.6g_b)/(g_s*g_b)$;

$A_c=V_{cmax}*(C_i-\Gamma_*)/(C_i+K_c*(1+O_i/K_o))$.

No results need to be changed. The code was correct as we compared carefully against the Fortran code in these subroutines as we developed the differentiable versions of the code. We will be publishing the code as the paper gets closer to acceptance so this can be examined in the code. Again, we apologize for the errors in the manuscript.

As all the results are validated only once using the temporal holdout data or the random holdout data, the generalizability of the dPL (or $NN_B+NN_v$) is not clear. If the N-fold or leave-one-out cross-validation can be adopted, the statistical metrics can be more justifiable to reflect the model performance.

Thanks for being rigorous. We believe the randomly-selected points were representative but we will conduct a cross validation (CV) and show the results. However we would also like to note that a temporal test is actually more stringent than a random-holdout cross validation. The temporal test examines if the model can run forward correctly for a period of time not included in the training whereas a random selection will always include some points from most recent times. The metrics also show that random holdout is better than temporal holdout. We noticed this issue through careful benchmarking in previous papers [Feng et al., 2021], but the uncertainty associated with temporal projection was often not paid attention to. We expect the full CV results to be better than temporal test results. We believe a spatial test, though, would belong to a different paper as the paper is already getting long. There are many techniques to improve spatial generalization and larger dataset from remote sensing which, if combined with the present content, would just be too much for a first paper. We plan to clarify this point in the paper.

The forcing variables and parameters are not clearly differentiated in the paper. For example, is the leaf layer boundary conductance, $g_b$, a constant parameter across sites or a temporally changing variable? If it is a forcing variable for FATES, where is $g_b$ from? is $\theta_{ice}$ a forcing variable or a parameter correlated with temperature and $\theta_{liq}$? Is the $C_a$ a constant value or variable? The model

would be different if the spatial and temporal variability of all these factors are considered. If all these are parameters (i.e., scalars), what are the values?

The Lin15 dataset included different forcing variables that we used in our model including:

| | |
|---|---|
| RH | Relative humidity |
| $T_{air}$ | Air temperature |
| $T_{leaf}$ | Leaf temperature |
| $P_{atm}$ | Atmospheric pressure |
| PAR ($\varphi$) | Photosynthetic active radiation |
| $g_b$ | Boundary layer conductance |

Concerning ($g_b$, $\theta_{ice}$ and $C_a$), here are details about how they were considered in the model:

- $g_b$, the boundary layer conductance values were already available in Lin15 dataset. However, it has some missing values which were computed using the (GetCanopyGasParameters) subroutine available in Fates model. https://github.com/NGEET/fates/blob/main/biogeophys/FatesPlantRespPhotosynthMod.F90

- $\theta_{ice}$, the volumetric ice content values were ignored (considered as zero) since both the air and leaf temperatures in our dataset were above the freezing temperature (0 °C or 273.15 K) by at least 5 degrees.

- $C_a$, the $CO_2$ partial pressure near the leaf surface values were variable spatially and temporally and they were taken as 0.039% of the atmospheric pressure

Line 216-218: the reason for replacing saturated soil matric potential ($\Psi$sat) with soil matric potential for closed stomata ($\Psi$c) is not explained. Equation 10 shows that the $\Psi$sat is replaced with soil matric potential for open stomata($\Psi$o), not $\Psi$c. Furthermore, the $\Psi$i was still calculated using $\Psi$sat in Appendix A (equations A16-A18). I'm confused about which variable was used to calculate $\Psi$i.

Line (216 – 218), we stated the actual equations that we used in for computing $\psi_i$ (in which $\psi_{sat}$ was replaced with $\psi_o$).

In Appendix A, we kept all the original equations the same whether those related to FATES or to computing the soil water stress function ($\beta_t$).

| Actual equation used in this study (Line 216 – 218) | Original equation (Appendix A) |
|---|---|
| $\Psi_i = \Psi_o \times S_i^{-B_i} \geq \Psi_c$ | $\Psi_i = \Psi_{sat,i} \times S_i^{-B_i} \geq \Psi_c$ |

Reasons for this replacement:

In the original CLM4.5 equations, $\psi_{sat}$ is based on empirical functions, percentage of sand *(%sand)*, and fraction of organic matter ($F_{om}$) (Equations A17 – A18). Using the original Equation 7 for computing $\psi_i$ results in a plant wilting factor $w_i$ equals to one for more than 90% of the data points across different soil layers.

To give the model more flexibility in the computation of $\psi_i$ and thus allow more variability in $w_i$ values, $\psi_{sat}$ was replaced with $\psi_o$. However, to ensure having $w_i$ values less than or equal 1 as in the original equation 9, we tried to create equation 10 in a way that satisfies this condition using $\psi_o$. For parameter B (outputted from $NN_B$), it was restricted to be within the range 0 and 1 to satisfy the same condition as well. Applying those changes, we were able to get $\psi_i$ values within the range of $\psi_o$ and $\psi_c$ while showing more variability in the computed $w_i$.

Also, we plan to add this paragraph to the **(Model changes)** section for clarification:

"*These changes were implemented to give more flexibility in the computation of the soil matric potential $\psi_i$. Using the original Equation 7 for computing $\psi_i$ results in a plant wilting factor $w_i$ equals to one for more than 90% of the datapoints across different soil layers. Thus, changing Equation 7 to the form shown in Equation 10 helped to express more variability in $w_i$ and eventually in the computed soil water stress function ($\beta_t$).*"

Here, the point is to calculate photosynthesis. We can see clearly the modified model works very well for photosynthesis. The differentiable modeling approach was specifically designed to enable inspection of various modules and assumptions in the model to update the formula.

As indicated, we are preparing a fuller response to you and will soon post it here. The description above only represents a quick reply. Thank you for your comments again!

References.
Shen, CP. et al., Differentiable modeling in Geosciences to unify machine learning and physical models. https://arxiv.org/abs/2301.04027

Wen-Ping Tsai, Dapeng Feng, Ming Pan, Hylke Beck, Yuan Yang, Kathryn Lawson, Jiangtao Liu, and Chaopeng Shen. From calibration to parameter learning: Harnessing the scaling effects of big data in geoscientific modeling. Nature Communications, (2021). doi: 10.1038/s41467-021-26107-z.

Feng, DP., K. Lawson and CP. Shen, Mitigating prediction error of deep learning streamflow models in large data-sparse regions with ensemble modeling and soft data, Geophysical Research Letters, doi: 10.1029/2021GL092999 (2021)

---

## Author Comment (AC2)

This paper presents a nice example of combining theory based models and machine learning to efficiently identify parameters of an ecosystem model, exploiting observation data recorded at multiple sites. The approach is valid and the results are interesting. However, the documentation of data and methods is currently deficient on a level that makes it hard to grasp the main messages and interpret the results. Section 2 of the paper does in my yes require a thorough revision, including new explanatory figures, restructuring and replacement of text blocks. For this reason I recommend a major revision or rejection with an invitation to resubmit.

Thank you for your evaluation!

1. Major comments

1. I assume a key point of the developed framework is that it enables to directly backpropagate from the outputs through the model equations to the neural networks. This is not clear from the paper at all. Much of the framework description seems like you feed NN predictions of parameters through a black box physics-based model, which is a standard approach. I suggest a dedicated subsection, possibly including a figure, to clarify this detail.

Yes, the differentiability which supports gradient-based optimization is the soul of the proposed work. We have discussed this in the paper (Abstract: "*programmatically differentiable (meaning gradients of outputs to variables used in the model can be obtained efficiently and accurately)…*", lines 146 "*In order to train the physical equations and neural networks together using gradient descent, the above equations were implemented on differentiable platforms to support backpropagation*"). To further emphasize it, we will add a paragraph at the beginning of section 2.1 (General overview) which explains Figure 1 to emphasize it. Also Figure 1 will be modified to represent both the forward run (blue arrows) and the backpropagation (black arrows) and thus better represent the framework (shown below).

"*Our general framework trains connected neural networks to provide parameters (and later process representations) to process-based models (PBM), in this case the photosynthesis module in FATES, on all the training data points simultaneously (Figure 1a). The neural networks maps from some raw inputs to some tuneable physical parameters ($\theta$) (later extensible to processes) required for the PBM. The predicted physical parameters are then fed into the differentiable PBM along with other required forcing variables (F) and untuned constant attributes ($\theta_c$) to compute the simulated target variable ($y_{sim}$) which is compared with observations to compute a loss function. The forward run starts from the neural networks and ends at the loss function (blue arrows in Figure 1a). We then backpropagate the errors (shown by black arrows in Figure 1a) through the PBM equations back to the neural networks to train them. To support gradient-based training, the entire framework must be differentiable [Shen et al., 2023] and neither the neural network nor the process-based model is a black box --- they both allow explicit inspection and modification of the internal structures. We had to reimplement the photosynthesis module of FACETS on differentiable platforms.*"

[Figure]

*Figure 1. Diagram showing the differentiable parameter learning (dPL) framework which is a hybrid of neural networks and the photosynthesis module in the FATES ecosystem model written on a differentiable platform. (a) The generic workflow: Some raw information is mapped into physical parameters via a neural network. These parameters are sent into a process-based model (PBM), which then outputs variable Y that is compared with observations. Direct supervision for the physical parameters is not required -- we do not need ground truth for these parameters. The loss function is "global" in that it involves all training data points, rather than being computed site-by-site as done in traditional calibration. (b) The workflow for the computational example described in this work. We estimate either $V_{c,max25}$ or the parameter B using neural networks, or both of them at the same time. When they were not estimated from data, default values from the literature were used. Blue arrows show running the neural networks with the PBM in a forward mode, while black arrows indicate backpropagation from the loss function back through the differentiable model equations to the neural networks to update their weights.*

2. The datasets used for training and testing are not properly documented. We don't know how many datapoints are included over which time periods. The random holdout suddenly appears in the results, and in general we don't know how training/validation/testing splits are defined.

This paragraph will be added to section **(Synthetic data and real data experiments)** to explain more about the temporal and the random holdout tests as well as data splitting.

"*For training and testing our candidate models, two different tests were performed with respect to data splitting: random holdout test and temporal holdout test, the latter of which stresses the models' ability to project into the future. In the temporal holdout test, for each PFT in each location, the available dates of measurements were counted where data points measured at the older 80% of these dates were used for training and the other more recent 20% were used for testing. Due to the randomness of dates of measurements available at each location (as mentioned previously in section 2.4.1), the temporal periods for the training and testing datasets vary by location. The temporal holdout test was used for both synthetic and real data experiments. For the random holdout test, as the name implies, 80% of the datapoints were randomly selected for training from the available PFT measurements in each location while the rest were used for testing. This test was run only for the real case experiments.*"
--- we will in fact change the train:test ratio to 80:20 and run a cross validation (actually--- this is really easy and we already did it, see results later).

CLM4.5 standard parameters play a central role in the results, but we know nothing about where they come from / how they are defined and if, for example, all or a subset of values are used for comparison.

https://opensky.ucar.edu/islandora/object/technotes%3A515/datastream/PDF/view
CM4.5 documentation presents the standard values of the parameters and the equations that we used in this study as a benchmark. In fact we compared to many other models in Table 3 and provided references (previously in the text and will be in the table itself). We will also add subsection (shown below) to section **(Input and observation datasets)** to better clarify.

*"**CLM4.5 default parameters**"*
*CLM4.5 documentation provided reference values and equations for both parameters $V_{c,max25}$ and B. For $V_{c,max25}$, the values corresponding to each PFT are well documented in CLM4.5 (section 8; table 8.1) (Oleson et al., 2013) and are shown in Table 3 in the manuscript. The same applies for parameter B with the default equations shown in Appendix A. CLM4.5 was also used to provide other photosynthetic parameters such as the soil matric potentials for closed stomata $\psi_c$ and open stomata $\psi_o$ (see Equation 9), and the plant root distribution parameters (see Equation 8) required for $\beta_t$ computations where all these parameters are considered as PFT-dependent."*

3. The explanation of the ecosystem model suffers from a clear struggle between trying not to include the entire set of equations in the paper, while providing sufficient detail. For me the level of detail provided in the paper was actually confusing, because it required constant looking up in the appendix to understand the context, distracting from the main messages. I think a way out could be to include a figure that summarizes the main blocks of the model (including what parts correspond to f1 and f2), include only the changed equations in the paper, and otherwise keep the full model description in the appendix. On a sidenote: is f2 not the same as an observation equation, that is commonly used in state space models?

We plan to add the figure below which show the block equations corresponding to f1 and f2 equations respectively. Yes, $f_2$ is the observation equation. $f_1$ and $f_2$ may share common components but they are mathematically different: f1 is a system constraint while f2 is a "observation equation". In this example, f1 is solved for the unknown $c_i$ while f2 connects $c_i$ to the observation $A_n$.

[Figure]

**Figure 2.** *showing the model block of equations corresponding to f1 and f2. Blue boxes refer to equations corresponding to f1. Orange boxes refer to equations corresponding to f2. Further details about the variables and parameters in these equations will be given in a separate table. Once we get the solution for Ci (intercellular leaf CO2 pressure) from f1 equations (nonlinear system), we can run f2 equations to get An (net photosynthesis rate)*

4. Details on hyperparameters (neural network # of layers, activation functions, learning rates etc.) are not provided at all. Some key information should be provided in the paper, and a reference to supporting information or the code should be provided for details.

This paragraph will be added to section **(Synthetic data and real data experiments)** which states some details about the hyperparameters

*"The used MLPs were very simple with only three layers; input layer, one hidden layer, and an output layer. To ensure an output value between 0 to 1 for both $V_{c,max25}$ and B parameterizations, sigmoid activation functions were used for both hidden and output layers. The quantity of available data posed a limitation and did not permit an extensive hyperparameter tuning experiment with a train/validation/test split. Hence, we employed a lazy trial and error with hyperparameters (learning rates and hidden size) using 70% of the randomly selected data as training data and 30% as a validation set, just to ensure we had a roughly performing hyperparameter set. We selected a learning rate of 0.01 and a hidden size that is equal to the number of inputs (9 for the $NN_v$ and 8 for the $NN_B$). We kept the same hyperparameters in the reporting, where we ran 5-fold cross validation. In addition, we found that moderately perturbing the hyperparameters resulted in very little change in the performance. This design considered the practical limits of available data, even this study already represents a large-sample study in the domain of ecosystem modeling."*

You can see, from the table below, moderate changes to the hiddensize does not matter too much. Thus, due to data limitation, we did not tune hyperparameter extensively. We simply use a hiddensize that is equal to the number of inputs. Should there be more data, we can certainly use a train/validation/test split and run more hyperparameter tuning.

| | Corr | | RMSE (μmol m-2 s-1) | | Bias (μmol m-2 s-1) | | NSE | | |
|---|---|---|---|---|---|---|---|---|---|
| | Train | Test | Train | Test | Train | Test | Train | Test | |
| V+B | 0.7994 | 0.7478 | 4.3002 | 4.4255 | 0.0476 | 0.3618 | 0.6379 | 0.5313 | $NN_B[8,6,1]$ |
| V+B | 0.7984 | 0.7473 | 4.3105 | 4.4281 | 0.0416 | 0.3569 | 0.6362 | 0.5308 | $NN_B [8,7,1]$ |
| V+B | 0.7994 | 0.7479 | 4.3003 | 4.4232 | 0.0376 | 0.3467 | 0.6379 | 0.5318 | $NN_B [8,8,1]$ |
| V+B | 0.7972 | 0.7445 | 4.3211 | 4.4358 | 0.0251 | 0.3001 | 0.6344 | 0.5291 | $NN_B [8,9,1]$ |
| V+B | 0.7989 | 0.7474 | 4.3053 | 4.4320 | 0.0420 | 0.3601 | 0.6371 | 0.5299 | $NN_B [8,8,8,1]$ |

Detailed comments

line 61: nonuniqueness is also going to be a problem if we employ newer frameworks like PINNs or dPL

Agree that nonuniqueness will still remain an issue and will need to be tested/controlled, but it should be better with dPL than with previous site-by-site calibration approach, because one neural network is constrained by all data points. There is an implicit spatial constraint. This effect was demonstrated in fine details in Tsai et al., 2021. As shown in that paper, as we turn parameter calibration into parameter learning, the framework can generalize better in space and in uncalibrated variables. It's obviously a tricky issue between the available data we have, the amount of structure we specify, and the tradeoff between variance and bias. What we hope to achieve is to maximally leverage the available information.

line 110: it might be worthwhile to start with a reference to figure 1 and a down to earth explanation of the objective of your work, i.e. to calibrate model parameters across many sites, to capture the variation of parameters using neural networks, and to employ differentiable programming to speed up the identification process

This paragraph will be added to section **(General overview)**

As replied earlier, we added the new first paragraph in Section 2.1, General overview, about the overall framework and citing Figure 1.
The original first paragraph will be modified:
"*In this case, the process-based model is related to the photosynthesis module in FATES, which can be written as a nonlinear system of equations and its solution is implicit. The system can be written as:*"

line 118: please explain PFT again in this section

PFT will be replaced with the full description plant functional type and the whole text will be modified to:
"*Some of the tunable parameters are typically formulated as being Plant Functional Type (PFT)-dependent (e.g., the maximum carboxylation rate) where each PFT include group of plant species that share similar physical and phenological characteristics leading to similar interaction with the environment*"

line 140: If you preserve eq. 4 and 5 in the paper, I think they should be presented in reverse order (f1 first, f2 second)

Both equations will be reversed

line 146-164: please include only methodological descriptions that are relevant for the results. of the julia implementation was not used, then it should not be described and discussed

Thanks for the point and we do understand where the reviewer is coming from. While Julia was not the main tool for production here, we thought it might be useful to mention it because the SciML toolset, co-developed by two of the coauthors, may be valuable to ecosystem modelers. Moreover, it is formulated very differently in a novel symbolic format which is in fact quite interesting and could potentially lead to a different path, and the package is evolving rapidly. Hence we think preserving it has some value. Removing it will also mean removing some coauthors, which we do not want to do.

line 183: you don't describe anywhere in your data how many PFTs you consider. it is therefore here also not clear how many dummy variables this model receives as input.

Our dataset included 9 different PFTs categories, a paragraph with more details about Lin15 dataset will be added to subsection **(Forcing and Photosynthesis rates)** stating the number of PFTs considered plus the name of each PFT.

*"We refer to this dataset as Lin15 throughout the rest of this work with 43 sites chosen whose dates and times of measurements were available. Lin15 covered nine different PFT categories including the following: rainfed crop "Crop R", Broadleaf Evergreen Tree Tropical "BET Tropical", Broadleaf Evergreen Tree Temperate "BET Temperate", C3 grass, C4 grass, Needleleaf Evergreen Tree Boreal "NET Boreal", Needleleaf Evergreen Tree Temperate "NET Temperate", Broadleaf Deciduous Tree Temperate "BDT Temperate", and Broadleaf Deciduous Shrub Temperate "BDS Temperate". Measurements were taken on sub-hourly scale but not necessarily on a continuous daily interval. That's why for almost all the sites, data were available on some random days (not necessarily continuous) in one or a few years. Lin15 also contained meteorological forcing variables, including air temperature, atmospheric pressure, relative humidity, and radiation. Moreover, we used ERA5 to fill in for any missing forcing variables in Lin15."*

line 190-205: I think this information is not needed to understand the main message

We think this information is important because we refer to it in different parts of the paper and they show brief description on how the soil water stress function ($\beta_t$) is calculated:

Line 190 – 195: show equation 7 which we later refer to as the equation to be replaced with equation 10 in the model changes section. Thus, we need to mention the old and the proposed equations.

Line 195 – 205: show the two final equations for calculating the soil water stress function ($\beta_t$) and the plant wilting factor ($w_i$), which we later refer to as part of the equations used in the synthetic and real data experiments after retrieving or estimating the parameter B

eq. 10: why is psi_max replaced by psi_0? (missing explanation)

Line (216 – 218), we stated the actual equations that we used in for computing $\psi_i$ (in which $\psi_{sat}$ was replaced with $\psi_o$).

In Appendix A, we kept all the original equations the same whether those related to FATES or to computing the soil water stress function ($\beta_t$).

| Actual equation used in this study (Line 216 – 218) | Original equation (Appendix A) |
|---|---|
| $\Psi_i = \Psi_o \times S_i^{-B_i} \geq \Psi_c$ | $\Psi_i = \Psi_{sat,i} \times S_i^{-B_i} \geq \Psi_c$ |

Reasons for this replacement:

In the original CLM4.5 equations, $\psi_{sat}$ is based on empirical functions, percentage of sand *(%sand)*, and fraction of organic matter ($F_{om}$) (Equations A17 – A18). Using the original Equation 7 for computing $\psi_i$ results in a plant wilting factor $w_i$ equals to one for more than 90% of the data points across different soil layers.

To give the model more flexibility in the computation of $\psi_i$ and thus allow more variability in $w_i$ values, $\psi_{sat}$ was replaced with $\psi_o$. However, to ensure having $w_i$ values less than or equal 1 as in

the original equation 9, we tried to create equation 10 in a way that satisfies this condition using $\psi_o$. For parameter B (outputted from $NN_B$), it was restricted to be within the range 0 and 1 to satisfy the same condition as well. Applying those changes, we were able to get $\psi_i$ values within the range of $\psi_o$ and $\psi_c$ while showing more variability in the computed $w_i$.

Also, we plan to add this paragraph to the **(Model changes)** section for clarification:

"*These changes were implemented to give more flexibility in the computation of the soil matric potential $\psi_i$. Using the original Equation 7 for computing $\psi_i$ results in a plant wilting factor $w_i$ equals to one for more than 90% of the datapoints across different soil layers. Thus, changing Equation 7 to the form shown in Equation 10 helped to express more variability in $w_i$ and eventually in the computed soil water stress function ($\beta_t$).*"

Here, the point is to calculate photosynthesis. We can see clearly the modified model works very well for photosynthesis. The differentiable modeling approach was specifically designed to enable inspection of various modules and assumptions in the model to improve model performance. It is possible that alternative formulations can also perform well and we do not preclude that here, as this is not a main point of concern for this paper.
eq. 11: what is F_om?

F_om is the fraction of organic matter and this is mentioned here after equation 7 "*where $\psi_{sat}$ is the saturated soil matric potential and S is the soil wetness, both defined for a specific soil layer. Different soil attributes such as percentages of sand (%sand) and clay (%clay), fraction of organic matter ($F_{om}$), and soil moisture ($\theta_{liq}$) are used in computing $\psi_{sat}$, S, and B (Appendix A).*"

line 232: the CLM4.5 data points should be documented in a dedicated data section. In general, I suggest they you separate the description of data and experiments

https://opensky.ucar.edu/islandora/object/technotes%3A515/datastream/PDF/view
CM4.5 documentation clearly presents the standard values of the parameters and the equations that we used in this study. A subsection (shown below) will also be added to section **(Input and observation datasets)** to better clarify

"***CLM4.5 default parameters***"
*CLM4.5 documentation played an important role in the results of this study by providing reference values and equations for both target parameters $V_{c,max25}$ and B. For $V_{c,max25}$, the values corresponding to each PFT are well documented in CLM4.5 (Oleson et al., 2013) and are shown in Table 3. The same applies for parameter B with the default equations shown in Appendix A. CLM4.5 was also used to provide other photosynthetic parameters such as the soil matric potentials for closed stomata $\psi_c$ and open stomata $\psi_o$ (see Equation 9), and the plant root distribution parameters (see Equation 8) required for $\beta t$ computations where all these parameters are considered as PFT-dependent.*"

The description of data and experiments are already located in different subsections. In the new submission, we can move the data subsection prior to the experiments' description subsection for better clarification.

line 239: were all calculations performed only for the topsoil layer in all experiments?

This is valid for the synthetic case only whose purpose was just to test the whole framework, while for the real case all the five soil layers (mentioned in Static attributes subsection) were used to estimate the parameter B.

Table 1: missing symbol explanations for means and standard deviations

For clarification, this line will be added to the bottom of table 1
 "$\sigma$ refers to the standard deviation, $\underline{OBS}$ refers to the mean of observations, $\underline{SIM}$ refers to the mean of simulations"

line 383: please include time series for observations and model predictions

We did not provide time series for several reasons. Measurements in Lin15 dataset were taken on sub-hourly scale but not necessarily on a continuous daily interval. for almost all the sites, data were available on some random days (not necessarily continuous) in one or a few years. In fact, many of the measurement days are far from each other and we can barely find consecutive days for producing sensible time series. Second, this model was not posed as a time-continuous problem. In other words, there is no accumulated memory between different dates. Hence, we think time series plot may even be somewhat misleading.

fig. 5: symbols in legend cannot be distinguished. are results shown for the test dataset?

These points belong to both training and testing datasets. We previously have a version that distinguish train and test, as pasted below. As you can see, there are no visual differences between two types of points and such symbology does not really bring in new information. Later, we wanted to use symbols to indicate PFTs, which seems more informative. So, to avoid overcomplicating the figure, we removed the train/test differences. We also remind the reviewer that we will provide cross validation results in the revised manuscript, which shows similar statistics as the random holdout.

We already ran the requested cross validation (5-fold). The figures below show train/test points from some of the random holdout folds:

Fold 1:

[Figure]

Fold 2:

[Figure]

Fold 3:

[Figure]

We skipped the other folds.

The one used in the paper:

[Figure]

It seems the one with different PFT type, used in the original paper, delivers more useful information. We can mention in the revision that the train test figures show that the test points were correctly captured.

line 426: i would add that you have identified parameter values that are optimized for the considered set of model equations and forcings. both of these have limitations. Equations may be wrong, ERA5 is rather uncertain, and measurement principles can vary between stations. This is both a limitation and a strength of your framework. Parameter values will not be transferable to other inputs. On the other hand you can obtain optimized predictions for the given set of forcings.

Good point. Just like any other model, the performance may be impacted when you change the forcing datasets because these datasets may have certain biases. If the model is trained on a global scale, we hope the various different kinds of forcings to be encountered can serve to limit overfitting. We will add the following sentences.

*"Such parameterizations are suitable to the target and forcing dataset used in training (still the most representative dataset we have access to) and are related to the process-based model employed. The dataset may have limitations related to the consistency in the measurement approach, the forcing data contain errors, while the model structure can be improved. The model performance may vary based on different forcing data, too."*

---

## Author Comment (AC3)

The authors of the manuscript 'A differentiable ecosystem modeling framework for large-scale inverse problems: demonstration with photosynthesis simulations' describe the application of the 'differentiable parameter learning'(dPL) framework to the photosynthesis module of FATES model. The framework, and concept, overcomes extrapolation limitations from site-by-site calibration approaches and allows leveraging information content in large-scale datasets towards a global parameterization of photosynthesis models. Neither the concept (Tsai et al., Nature Communications, https://www.nature.com/articles/s41467-021-26107-z, 2021; Bao et al., Authorea, https://www.authorea.com/doi/full/10.1002/essoar.10512186.3, 2022) nor the dPL framework (Tsai et al., 2021; Feng et al., 2022ab) are new. However, the framework is used in the FATES model for the first time and the results would be of interest for further model development, but also to the scientific community at large.

At this point, the experiment focuses on inverting two parameters,  $V_{cmax25}$  and B, resulting in that the accuracy of the simulated net photosynthesis rate being slightly improved. The main concerns at this stage relate to apparently incorrect formulations of some key equations, to issues about the validation strategy, to the fact that the forcing data and the experiments are not described sufficiently, challenging the acceptance of the study, while hampering any reproducibility efforts. Please see below for details.

We thank your detailed comments! Wow, this ends up being a 24-page response. As a summary, it seems most of the questions seek clarifications and details about the model. Thank you, and these comments should help us elucidate the model better. We did not find major comments that require computational experiments or major reorganization. There is a question about cross validation, which we have already run. It shows expected and essentially similar results. Moreover, some metrics were requested and we calculated them and reported them in the responses.

We indeed followed our previous differentiable parameter learning paradigm which was first applied in hydrology (Tsai et al., 2021; Feng et al., 2022), as noted in the manuscript, but this is a novel use in the large domain of ecosystem modeling, which is a very large field of study. The system is also different as here we have a nonlinear system of equations while in hydrologic cases we have ordinary differential equations. The mathematical treatment was different. The Julia software solves the system using adjoint solvers, although it is a relatively minor point as we mainly used the PyTorch version for its high parallel efficiency.

We could not have noticed Bao et al., 2022 as it went online after our manuscript did and seems to be undergoing review. Upon some examination, we believe the basic modules are very different. They are using a light-use-efficiency approach and predicted GPP, while our paper focused on photosynthesis using a Farquhar-type model. Hence we don't think there is much overlap between the two.

**Major comments:**

1. Two key equations are incorrect in the paper:

1) line 140: equation 5,  $C_i=C_a-A_n*P_{atm}*(1.4g_s+1.6g_b)/(g_s+g_b);$

2) line 505: equation A1,  $A_c = V_{cmax} (C_i - \Gamma_*)/(C_i + K_c * (1 + K_o/O_i))$ .

According to the user guide of the FATES model (https://fates-users-guide.readthedocs.io/projects/tech-doc/en/latest/fates\_tech\_note.html#fundamental

photosynthetic-physiology-theory), the equations should be:

1)  $C_i = C_a - A_n * P_{atm} * (1.4g_s + 1.6g_b)/(g_s * g_b);$

2)  $A_c = V_{cmax} * (C_i - \Gamma_*) / (C_i + K_c * (1 + O_i / K_o)).$

Since the FATES model is reimplemented in Julia and PyTorch by the authors, the codes might be also wrong. If so, the unit of  $C_i$  will be incorrect, leading to errors in the inversion of  $V_{cmax25}$  and B. The wrong computation of the effective Michaelis-Menten coefficient (= $K_c*(1+O_i/K_o)$ ) might only have a slight effect if the temperature is close to 25°C, but should be concerned if the temperature is too low or high (and I do see some points with low leaf temperature in the 'Lin15' database). Thus, I have doubts about the current results and relevant analysis.

Regarding the equations --- we were cautious to adhere to the original FATES equations before implementing it on PyTorch or Julia. Unfortunately, we **realized there were some typos in the manuscript** in line 140 and line 505 in the paper which will then be modified. However, we used the correct equations in our differentiable model as the following:

 $\begin{array}{l} C_i = C_a - A_n * P_{atm} * (1.4 g_s + 1.6 g_b) / (g_s * g_b); \\ A_c = V_{cmax} * (C_i - \Gamma_*) / (C_i + K_c * (1 + O_i / K_o)). \end{array}$

No results need to be changed. The code was correct as we compared carefully against the Fortran code in these subroutines as we developed the differentiable versions of the code. We will be publishing the code as the paper gets closer to acceptance so this can be examined in the code. Again, we apologize for the errors in the manuscript.

1. As all the results are validated only once using the temporal holdout data or the random holdout data, the generalizability of the dPL (or  $NN_B+NN_v$ ) is not clear. If the N-fold or leave-one-out cross-validation can be adopted, the statistical metrics can be more justifiable to reflect the model performance.

Thanks for being rigorous. We believe the randomly selected points were representative, but we already conducted a cross validation (CV) and show the results, as this is trivial. The results are as follows:

| Deres     | Corr  |       | RMSE                   |       | Bias                   |        | NSE   |       |
|-----------|-------|-------|------------------------|-------|------------------------|--------|-------|-------|
| Kuns      | Train | Test  | Train                  | Test  | Train                  | Test   | Train | Test  |
|           | Corr  |       | RMSE
(μmol m-2 s-1) |       | Bias
(µmol m-2 s-1) |        | NSE   |       |
|           | Train | Test  | Train                  | Test  | Train                  | Test   | Train | Test  |
| Vdef+Bdef | 0.565 |       | 6.780                  |       | 1.476                  |        | 0.041 |       |
| Vdef+B    | 0.632 | 0.581 | 6.315                  | 6.088 | 1.488                  | 0.890  | 0.182 | 0.177 |
| V+Bdef    | 0.758 | 0.567 | 4.599                  | 6.148 | -0.166                 | -1.630 | 0.566 | 0.161 |

(a) Temporal holdout test for the following system (80% train: 20% test)

| V+B | 0.788 | 0.766 | 4.302 | 4.343 | 0.104 | -0.247 | 0.62 | 0.581 |
|-----|-------|-------|-------|-------|-------|--------|------|-------|
|-----|-------|-------|-------|-------|-------|--------|------|-------|

|           | Co    | orr   | RMSE                   |       | Bias                   |        | NSE   |       |
|-----------|-------|-------|------------------------|-------|------------------------|--------|-------|-------|
| Runs      | Train | Test  | Train                  | Test  | Train                  | Test   | Train | Test  |
|           | Corr  |       | RMSE
(µmol m-2 s-1) |       | Bias
(µmol m-2 s-1) |        | NSE   |       |
|           | Train | Test  | Train                  | Test  | Train                  | Test   | Train | Test  |
| Vdef+Bdef | 0.565 |       | 6.780                  |       | 1.476                  |        | 0.041 |       |
| Vdef+B    | 0.620 | 0.618 | 6.283                  | 6.305 | 1.456                  | 1.447  | 0.175 | 0.171 |
| V+Bdef    | 0.714 | 0.707 | 4.963                  | 5.018 | -0.416                 | -0.407 | 0.485 | 0.475 |
| V+B       | 0.783 | 0.772 | 4.308                  | 4.409 | 0.083                  | 0.094  | 0.612 | 0.595 |

(b) Cross Validation (5-fold) test for the following system

We also provide the metrics for each fold:

| Folds | COR_test | RMS_test | BIAS_test | NSE_test |       |
|-------|----------|----------|-----------|----------|-------|
| 1     | 0.726    | 4.835    | 0.959     | 0.495    |       |
| 2     | 0.834    | 3.962    | -0.228    | 0.683    | V±D   |
| 3     | 0.787    | 4.512    | -0.355    | 0.617    | V T D |
| 4     | 0.804    | 4.335    | 0.027     | 0.646    |       |
| 5     | 0.729    | 4.318    | -0.025    | 0.509    |       |

This is exactly as we expected in our initial reply posted a few days earlier --- the 5-fold CV results are similar to the previous random results and better than the temporal test results. In addition, we show the train/test  $A_n$  values for some random folds:

**FOLD1 (solid circle indicate test):**

---

## Author Comment (AC4)

**FOLD1 (solid circle indicate test):**

[Figure]

**Fold 2:**

[Figure]

**Fold 3:**

[Figure]

**Fold 4:**

[Figure]

**Fold 5:**

---

## Author Comment (AC6)

Dear Biogeosciences editor and reviewers,

Thank you for handling and thanks to the reviewers for their constructive suggestions. It seems most comments were asking for clarifications and further exploration of data. Reviewers have commented that they would like to see a revised version. While responding to reviewers, we have already run cross-validation experiments and have provided additional visualizations including plots for different PFTs as requested by the reviewers. We would also stress that as a first paper with a differentiable model for a Farquhar formulation, this paper needs to have a strong focus. We did not claim the model can well simulate seasonality at a site, and will further clarify it in the revised manuscript that this point is to be studied in future work. Because the backbone of the model is Farquhar, its seasonal behavior should be comparable to what we expect out of the other Farquhar models, because in this paper we only estimated static parameters.

In the following, we add more complete answers to a few detailed questions that were asked before the interactive system closes so we didn't have the time to address.

1. Line 218-220: is $NN_B$ used to predict $B_i$ or $\Psi_i$? B depends on only %clay and $F_{om}$ according to equations A22-A23, while the authors add %sand, which is related to $\Psi_{sat}$ and, therefore, $\Psi_i$. I didn't find a direct relationship between $B_i$ and %sand according to the original equations in the FATES model. If $NN_B$ is used to predict $\Psi_i$, I think the equation can be $\Psi_i=\theta_{liq}*NN_B(\%sand,\%clay,PFT,F_{om},T)$, where T represents the factors controlling $\theta_{ice}$, e.g., temperature.

It was suggested in this comment to include the temperature in $NN_B$ to represent $\theta_{ice}$ as the following:
$\Psi_i=\theta_{liq}*NN_B(\%sand,\%clay,PFT,F_{om},T)$, need to rescale between $\Psi_c$ and $\Psi_o$

However, we claim that including T in $NN_B$ to represent $\theta_{ice}$ might not very effective as justified afterwards.

The histogram of air temperature (shown in the figure below) indicates that we don't have in our dataset air temperatures below 5 °C and that clarifies why $\theta_{ice}$ was ignored in our calculations. Thus, there is low probability that the temperature would have a great effect being included in $NN_B$.

[Figure]

2. Line 431-432: I cannot identify the C3 grass at the lower left corner of figure 5b. Maybe a violin plot per PFT can be helpful to show the difference between optimizing B or not for a specific plant type. The figures in the paper only show the net photosynthesis rate across all sites. However, the site-level comparison might be more meaningful to assess the four parameterization strategies: Vdef+B, Vdef+Bdef, V+Bdef, and V+B.

The violin plots showed the net photosynthesis per PFT, but I think readers would be more interested in how different is the simulated net photosynthesis from the measured net photosynthesis. Maybe the fourth violin plot (measured values) can be added on the right side and the NSE can be displayed at the top. Moreover, I think only the test dataset (or better cross-validated dataset) should be compared with the measured values (e.g., Fig. 5) and used to make the violin plots.

For better clarification of different PFTs, we split figure 5 into 3 rows representing 3 PFTs (each of 2 subplots) as shown below. We used cross validated dataset for making the plot to avoid confusion concerning which dataset was used for the plot (train or test). Splitting figure 5 this way helps present the same information as the violin plot (which we can then exclude in the new version to avoid redundancy).

Regarding computing the NSE per PFT, as expected, this leads to lowering down the performance (lower NSE values) as shown in the attached table for each PFT, especially BET Tropical and BDT temperate, and C3 grass. We will comment on it in the revised manuscript.

[Figure]

| PFT | NSE_train | NSE_test | No. of datapoints per PFT | No. of species per PFT | No. of locations per PFT |
|---|---|---|---|---|---|
| Crop R | 0.3687 | 0.3648 | 85 | 5 | 4 |
| NET Boreal | 0.6604 | 0.6178 | 32 | 3 | 3 |
| BET Tropical | -0.1400 | -0.1419 | 367 | 185 | 11 |
| NET Temperate | 0.5859 | 0.5804 | 153 | 10 | 9 |
| BET Temperate | 0.3880 | 0.3401 | 69 | 16 | 9 |
| BDT Temperate | -0.0193 | -0.0329 | 61 | 12 | 7 |
| C3 grass | -0.0180 | -0.0564 | 58 | 17 | 2 |

| | | | | | |
|---|---|---|---|---|---|
| BDS Temperate | -0.0246 | -0.2257 | 28 | 12 | 4 |
| C4 grass | 0.5299 | 0.5021 | 21 | 6 | 1 |

3. There are limited site-level temporal data, thus the seasonality of net photosynthesis cannot be assessed.

We did not claim the model can simulate seasonality very well at a site. Currently our differentiable model follows the same structure of the photosynthesis module in the process-based model "FATES". We didn't make significant changes to the model. Because the backbone of the model is Farquhar, its seasonal behavior should be comparable to what we expect out of the other Farquhar models, because in this paper we only estimated static parameters. On the other hand, the nature of our dataset doesn't enable us to test the seasonality and we didn't mention this in the manuscript. This might be our scope in the future work.

To avoid confusion, we would add limitations section in the new version of the manuscript including the following paragraph:

"*Although applying the dPL framework improved the parameters to an extent, the model still has similar structural limitations as other Farquhar-type models. We didn't test the model's ability to capture the seasonality of the net photosynthetic rate due the limited site level temporal data. The seasonal behavior of the model is expected to be similar to other Farquhar models.*"

4. The authors clarified that the Vcmax25 is predicted per PFT but did not mention Bi. Is Bi predicted per PFT or per site? How is the predicted Bi compared with values from CLM?

$B_i$ differs between different sites and for one site it differs for different PFTs; $B_i = NN_B$(%sand, %clay, $F_{om}$, PFT). Contrary to $V_{c,max25}$, there are no default values for B because of two reasons:

    a. B in the default CLM4.5 equations come from empirical equations based on %clay and $F_{om}$
    b. We changed equation 7 to equation 10 (as shown below). Thus, parameter B in equation 7 has a completely different range from the one in equation 10 which ranges between 0 and

**Equations 7 and 10**

$$\Psi_i = \Psi_{sat,i} \times S_i^{-B_i} \geq \Psi_c \qquad\qquad\qquad (7) \text{ default}$$

$$\Psi_i = \Psi_o \times S_i^{-B_i} \geq \Psi_c \qquad\qquad\qquad (10) \text{ New}$$

**Default B equations in CLM4.5**

$$B_i = (1 - F_{om,i}) \times B_{min,i} + F_{om,i} \times B_{om} \qquad\qquad (A22)$$

$$B_{min,i} = 2.91 + 0.159 \times (\%clay)_i \qquad\qquad (A23)$$

For better clarification, this text can be modified in the new version in which we state that we used the two symbols $\Psi_i$ (PFT) and $C_i$ (in place of $\Psi_i$ and $B_i$ respectively) to avoid confusion with the terms in the original equations.

[*In the original water limitation function in CLM4.5, the stomata response to soil water potential is based on a linear function between the water potential for stomata openness and closeness. In view that plant could respond to soil water potential differently dependent on plant hydraulic traits (Christoffersen et al. 2016), in this study, we modified the soil water limitation for PFTs so that they could have different shapes. Specifically, we designed a PFT-dependent soil water stress [$\Psi_i$ (PFT), ranging from $\Psi_c$ and $\Psi_o$] depending on the soil water content, which is calculated as follows:*

$$\Psi_i \text{ (PFT)} = \Psi_o \times S_i^{-C_i(soil,PFT)} \geq \Psi_c \qquad (10)$$

*Where $C_i$ is a PFT- and soil-texture-dependent shape parameter (between 0 and 1) estimated as:*

$$C_i = NN_c(\%sand, \%clay, F_{om}, PFT) \qquad (11)$$

*The PFT-dependent soil water stress $\Psi_i$ (PFT), is then feed into the plant wiling equation (equation 9) as the following:*

$$w_i = \frac{\Psi_c - \Psi_i \text{ (PFT)}}{\Psi_c - \Psi_o} = \frac{\Psi_c - \max(\Psi_c, \Psi_o \times S_i^{-C_i(soil,PFT)})}{\Psi_c - \Psi_o} \qquad (12)$$

]

5. Since the site-level comparison and the site-average An comparison are not possible, the generalizability cannot be evaluated. However, the model performance across sites can be compared to other papers using the Farquhar model (e.g., Fig 1B of Chen at al., PNAS, https://doi.org/10.1073/pnas.2115627119, 2022).

Concerning, the spatial generalization or the site-level comparison, as mentioned in the previous responses, spatial test is not the scope of this paper. we are working on further improving the spatial generalization with some error mitigation approaches. This will add lots of content and should be for the scope of another paper. So, we can add the following paragraph to the limitations section and the future work:

*"This study doesn't cover the spatial generalization of the dPL model since we don't present results for spatial tests or based on site-level comparison. To improve spatial generalization may require further changes in the model, dynamical parameters, or using other error mitigation approaches. This is not our scope for this study; however, it would be conserved for future work."*

Things being asked for and will be added or modified in the new manuscript version:

1. Explanation for temporal and random holdout tests >> (will be clarified in the new version through paragraphs added)

2. f1 and f2 equations clear explanation in the manuscript body >> (proposed figure 2 will be added with explanation for the terms in the equations)

3. Details on NNs hyperparameters and hyperparameters tuning >> (will be clarified in the new version through paragraphs added)

4. Inquiries about Lin15 dataset >> (Number and full name of PFTs, forcing variables, atmospheric $CO_2$ ($C_a$), leaf layer boundary conductance ($g_b$)) >> (will be clarified in the new version through paragraphs added)

5. Reasons of replacing $\Psi_{sat}$ by $\Psi_o$ in equations 7 and 10 >> (will be clarified in the new version through paragraphs added)

6. CLM4.5 contribution to the study >> (will be clarified in the new version through paragraphs added)

7. Inquiries about B calculations across soil layers >> (will be clarified in the new version through paragraphs added for synthetic and real case experiments)

8. Cross validation tests >> (were performed and results will be added in the new version)

9. Model performance is impacted by certain set of model equations and forcings >> (will be clarified in the new version through paragraphs added)

10. Modify typos in model equations >> (will be modified in the new version)

11. $NN_B$ and $NN_v$ constraints on outputs and output range >> (will be clarified in the new version through paragraphs added)

12. More complex NN for the real case than synthetic case >> (already done for $NN_B$ but not applicable for $NN_v$)

13. Loss function clarification >> (will be better clarified in the new version + Figure 1)

14. Timeseries of observations >> (can't be provided due to the site limited temporal data)

15. Spatial variability of the parameters not fully captured by dPL >> (spatial test is not the scope of this paper, we are working on further improving the spatial generalization with some error mitigation approaches. This will add lots of content and should be for the scope of another paper.

16. Soil organic carbon content unit conversion >> (will be clarified in the new version)

17. $V_{c,max25}$ correlation literature values >> (the proposed figure 8 shown in previous response will be added showing the correlation between learnt and reference $V_{c,max25}$ values)

18. Split plots per PFT >> (Figure 5 will be spitted into 3 rows each with only 3 PFTs)

19. Figure 5b plotted using both training and test datasets >> (will be replotted using the cross-validation test results in the new version)

References

Christoffersen, B. O., M. Gloor, S. Fauset, N. M. Fyllas, D. R. Galbraith, T. R. Baker, B. Kruijt, L. Rowland, R. A. Fisher, O. J. Binks, S. Sevanto, C. Xu, S. Jansen, B. Choat, M. Mencuccini, N. G. McDowell, and P. Meir. 2016. "Linking Hydraulic Traits to Tropical Forest Function in a Size-Structured and Trait-Driven Model (TFS v.1-Hydro)." Geosci. Model Dev. 9(11):4227–55. doi: 10.5194/gmd-9-4227-2016.

---

## Author Response (AR1)

Dear Biogeosciences editor,

Thank you for handling and thanks to the reviewers for their constructive suggestions. We have completed a round of major revision.

Most comments were asking for clarifications and further exploration of data. We have made effort to improve clarity in the manuscript. We have run cross-validation experiments and have provided additional visualizations including plots for different PFTs as requested by the reviewers. Regarding the seasonality of the model, while it is not the focus of this paper, we can comment that the model seasonal behavior should be comparable to what we expect out of the other Farquhar models (photosynthesis only), because here we only estimated static parameters. For a few R2's questions, there is already content in the original manuscript discussing those topics so we pointed out where the content is and added more emphasis if needed.

There are many questions. To facilitate your processing, we pasted the modifications in the manuscript to the respective response to questions. We also provided in this document, the line numbers from the clean manuscript version (uploaded as "diffecosys_paperV3.0_revised_clean") that correspond to text modified or added. In the following, brown text is from reviewers while black text is our response.
* * *
Anonymous Referee #1
This paper presents a nice example of combining theory based models and machine learning to efficiently identify parameters of an ecosystem model, exploiting observation data recorded at multiple sites. The approach is valid and the results are interesting. However, the documentation of data and methods is currently deficient on a level that makes it hard to grasp the main messages and interpret the results. Section 2 of the paper does in my yes require a thorough revision, including new explanatory figures, restructuring and replacement of text blocks. For this reason I recommend a major revision or rejection with an invitation to resubmit.

Thank you for your evaluation!

Major comments

1. I assume a key point of the developed framework is that it enables to directly backpropagate from the outputs through the model equations to the neural networks. This is not clear from the paper at all. Much of the framework description seems like you feed NN predictions of parameters through a black box physics-based model, which is a standard approach. I suggest a dedicated subsection, possibly including a figure, to clarify this detail.

Yes, the differentiability which supports gradient-based optimization is the soul of the proposed work. We have discussed this in the paper (Abstract: "*programmatically differentiable (meaning gradients of outputs to variables used in the model can be obtained efficiently and accurately)…*", lines 146 (in the previous manuscript version) "*In order to train the physical equations and neural networks together using gradient descent, the above equations were implemented on differentiable*

*platforms to support backpropagation*"). To further emphasize it, we added a paragraph {lines 112: 123} at the beginning of section 2.1 **(General overview)** to emphasize it. Also Figure 1 was modified to represent both the forward run (blue arrows) and the backpropagation (black arrows) and thus better represent the framework (shown below).

"*Our general framework trains connected neural networks to provide parameters (and later process representations) to process-based models (PBM), in this case the photosynthesis module of the FATES ecosystem model, on all the training data points simultaneously (Figure 1a). The neural networks map from some raw inputs to some tuneable physical parameters (θ) (later extensible to processes) required for the PBM. The predicted physical parameters are then fed into the differentiable PBM along with other required forcing variables (F) and untuned constant attributes (θ$_c$) to compute the simulated target variable (y$_{sim}$) which is compared with observations to compute a loss function. The forward run starts from the neural network inputs and ends at the loss function (following the blue arrows in Figure 1a). We then backpropagate the errors (shown by black arrows in Figure 1a) through the PBM equations back to the neural networks so we can train them using gradient descent. To support gradient-based training, the entire framework must be differentiable (Shen et al., 2023) and neither the neural network nor the process-based model is a black box --- they both allow explicit inspection and modification of the internal structures. Thus, the photosynthesis module of FATES had to be reimplemented on differentiable platforms.*"

[Figure]

***Figure 1.** Diagram showing the differentiable parameter learning (dPL) framework which is a hybrid of neural networks and the photosynthesis module in the FATES ecosystem model written on a differentiable platform. (a) The generic workflow: Some raw information is mapped into physical parameters via a neural network. These parameters are sent into a process-based model (PBM), which then outputs variable Y that is compared with observations. Direct supervision for the physical parameters is not required -- we do not need ground truth for these parameters. The loss function is "global" in that it involves all training data points, rather than being computed site-by-site as done in traditional calibration. (b) The workflow for the computational example described in this work. We estimate either V$_{c,max25}$ or the parameter B, or both of them at the same time, using neural networks. The parameters are then fed into the differentiable photosynthesis module in FATES, which then outputs the net photosynthesis rate, A$_{n(sim)}$, that is compared with A$_{n(obs)}$. When they were not estimated from data, default values from the*

*literature were used. Blue arrows show running the neural networks with the PBM in a forward mode ("prediction" mode), while black arrows indicate backpropagation from the loss function back through the differentiable model equations to the neural networks to update their weights, which is only done during initial NN training.*

2. The datasets used for training and testing are not properly documented. We don't know how many datapoints are included over which time periods. The random holdout suddenly appears in the results, and in general we don't know how training/validation/testing splits are defined.

We apologize for this oversight. The following paragraph {lines 371: 380} was added to section 2.5 (**Synthetic data and real data experiments**) to explain more about the tests held as well as how the data were split.

"*Two different tests were performed with respect to data splitting: temporal holdout and randomized cross-validation --- the former test stresses the models' ability to project into the future while the latter is the typical experiment run in the literature. Due to the irregularity of measurement dates at each location (as mentioned previously in section 2.4.1), the temporal periods for the training and testing datasets varied by location. In the temporal holdout test, for each PFT in each location, the available dates of measurements were recorded. The oldest 80% of these dates were used for training and the remaining more recent 20% were used for testing. The temporal holdout test was run for both synthetic and real data experiments. For the randomized cross-validation test, as the name implies, the dataset was randomly split into 5 folds (groups) and each time the model was trained on 4 folds (80% of the datapoints) and tested on the $5^{th}$ fold (20% of the data points). This was done a total of 5 rounds, so that all of the data points were used for testing once. The cross-validation test was run only for the real data experiments..*"

3. CLM4.5 standard parameters play a central role in the results, but we know nothing about where they come from / how they are defined and if, for example, all or a subset of values are used for comparison.

Reference for CLM4.5:
https://opensky.ucar.edu/islandora/object/technotes%3A515/datastream/PDF/view
CLM4.5 documentation presents the standard values of the parameters and the equations that we used in this study as a benchmark and a detailed discussion of these choices is outside the scope of this work. We already did provide some of the basic parameter values (model values for $V_{c,max25}$) from CLM4.5 and other similar models in Table 3 and provided references (in the text). We also added a subsection {lines 295: 301} to section 2.4 (**Input and observation datasets**) to better clarify as shown below.

"2.4.3 *CLM4.5 default parameters*
*CLM4.5 documentation (Oleson et al., 2013) provide reference values for comparison and equations for both target parameters $V_{c,max25}$ and B. For $V_{c,max25}$, default values corresponding to each PFT (shown in Table 3) are well documented in CLM4.5 (chapter 8; table 8.1). Similarly, for parameters B and $\beta_t$, their default equations (shown in this work in Appendix B) are provided in the documentation of CLM4.5 as well. We also used other PFT photosynthetic parameters required for $\beta_t$ computations, such as the soil matric potentials for closed stomata, $\psi_c$, and open stomata, $\psi_o$, (see Equations 8,10,12), and the plant root distribution parameters (see Equation 9).*"

4. The explanation of the ecosystem model suffers from a clear struggle between trying not to include the entire set of equations in the paper, while providing sufficient detail. For me the level of detail provided in the paper was actually confusing, because it required constant looking up in the appendix to understand the context, distracting from the main messages. I think a way out could be to include a figure that summarizes the main blocks of the model (including what parts correspond to f1 and f2), include only the changed equations in the paper, and otherwise keep the full model description in the appendix. On a sidenote: is f2 not the same as an observation equation, that is commonly used in state space models?

We added the figure below which shows the block equations corresponding to equations f1 and f2 which we have renamed f and h, respectively. Yes, $f_2$ (h) is the observation equation. f and h may share common components, but they are mathematically different: f is a system constraint while h is a "observation equation". In this example, f is solved for the unknown $C_i$ while h connects $C_i$ to the observation $A_n$.

[Figure]

**Figure 2. Model equations corresponding to f and h in equation 1. Blue boxes indicate equations corresponding to f. Yellow boxes indicate equations corresponding to h. First, we obtain a solution for $C_i$ (intercellular leaf CO₂ pressure) by solving the nonlinear system (f equations) as illustrated in the last blue box. Then, we forward h equations to compute $A_n$ (net photosynthesis rate) using $A_c$, $A_j$, and $A_p$ as discussed in section 2.2. Details about different variables and parameters included in f and h equations are provided in Appendix A.**

5. Details on hyperparameters (neural network # of layers, activation functions, learning rates etc.) are not provided at all. Some key information should be provided in the paper, and a reference to supporting information or the code should be provided for details.

We apologize for this oversight. We added a paragraph {lines 356: 369} to section 2.5 (**Synthetic data and real data experiments**) which states some details about the hyperparameters:

"*The MLPs employed had three layers: an input layer, one hidden layer, and an output layer. To ensure an output value between 0 and 1 for both $V_{c,max25}$ and B parameterizations, sigmoid activation functions were used for both hidden and output layers. $V_{c,max25}$ was then rescaled to be within a pre-defined range based on literature values of 20 to 150 µmol m$^{-2}$ s$^{-1}$. For the i-th soil layer, $B_i$ values were kept between 0 and 1, so with $S_i$ ranging between 0.01 and 1 (see Appendix B), the term $S_i^{-Bi}$ then had a range of 1 to 100. This ensured that the value of $\psi_i$ ranges from $\psi_c$ to $\psi_o$ (see Equation 10).*

*The quantity of available data posed a limitation and did not permit an extensive hyperparameter tuning experiment with a train/validation/test split. Hence, we employed a "lazy" trial and error process with hyperparameters (learning rates and hidden size) using 70% of the data as training data and 30% as a validation set, just to ensure we had a roughly performing hyperparameter set (see Appendix C). We selected a learning rate of 0.045 and a hidden size equal to the number of inputs (9 for the $NN_V$ and 8 for the $NN_{Bi}$). We kept these same hyperparameters when we ran 5-fold cross validation with an 80%:20% train:test ratio. In addition, we found that moderately perturbing the hyperparameters resulted in very little change in the performance. This design was necessary considering the practical limits of the available data, even though this study already represents a large-sample study in the domain of ecosystem modeling.*"

The table below (added in Appendix C in the revised manuscript) shows that moderate changes to the hidden size do not matter too much. Thus, due to data limitation, we did not tune hyperparameter extensively. We simply used a hidden size equal to the number of inputs. Should there be more data available in the future, we could certainly use a train/validation/test split and run more hyperparameter tuning.

**Table C1 V+B model formulation performance for different sizes of NN$_{Bi}$ with 80%:20% train: test split ratio**

| | Corr | | RMSE (µmol m-2 s-1) | | Bias (µmol m-2 s-1) | | NSE | | |
|---|---|---|---|---|---|---|---|---|---|
| | **Train** | **Test** | **Train** | **Test** | **Train** | **Test** | **Train** | **Test** | |
| | 0.7862 | 0.7712 | 4.3188 | 4.2920 | 0.0898 | -0.2339 | 0.6175 | 0.5904 | NN$_{Bi}$[8,6,1] |
| | 0.7863 | 0.7713 | 4.3178 | 4.2912 | 0.0866 | -0.2395 | 0.6177 | 0.5905 | NN$_{Bi}$ [8,7,1] |
| **V+B** | 0.7862 | 0.7706 | 4.3190 | 4.2957 | 0.1023 | -0.2261 | 0.6174 | 0.5897 | NN$_{Bi}$ [8,8,1] |
| | 0.7858 | 0.7700 | 4.3222 | 4.3018 | 0.0711 | -0.2653 | 0.6169 | 0.5885 | NN$_{Bi}$ [8,9,1] |
| | 0.7855 | 0.7720 | 4.3275 | 4.2864 | 0.1049 | -0.2182 | 0.6159 | 0.5914 | NN$_{Bi}$ [8,8,8,1] |

1. line 61: nonuniqueness is also going to be a problem if we employ newer frameworks like PINNs or dPL

We agree that non-uniqueness will still remain an issue and will need to be tested/controlled, but it should be better with dPL than with previous site-by-site calibration approach, because one neural network is constrained by all data points. There is an implicit spatial constraint. This effect was demonstrated in fine detail in Tsai et al., 2021. As shown in that paper, as we turn parameter calibration into parameter learning, the framework can generalize better in space and in uncalibrated variables. It's obviously a tricky issue between the available data we have, the amount of structure we specify, and the tradeoff between variance and bias. What we hope to achieve is to maximally leverage the available information.

2. line 110: it might be worthwhile to start with a reference to figure 1 and a down to earth explanation of the objective of your work, i.e. to calibrate model parameters across many sites, to capture the variation of parameters using neural networks, and to employ differentiable programming to speed up the identification process

As mentioned above in our response to major question 1, we added a new first paragraph {lines 125: 127} in Section 2.1, General overview, about the overall framework and cited Figure 1. That paragraph is followed by:

"*In this case, the process-based model is the photosynthesis module in FATES, which can be written as a nonlinear system of equations, and its solution is implicit. The system can be written as:…*"

3. line 118: please explain PFT again in this section

PFT was replaced with the full description of plant functional type and the whole text was modified {lines 133: 135} to:
 "*Some of the tuneable parameters are typically formulated as being Plant Functional Type (PFT)-dependent (e.g., the maximum carboxylation rate at 25°C, $V_{c,max25}$) where each PFT includes groups of plant species that share similar physical and phenological characteristics leading to similar interactions with the environment.…*"

4. line 140: If you preserve eq. 4 and 5 in the paper, I think they should be presented in reverse order (f1 first, f2 second)

These equations have been reversed to show f (formerly f1) before h (formerly f2).

5. line 146-164: please include only methodological descriptions that are relevant for the results. of the julia implementation was not used, then it should not be described and discussed

Thanks for the point and we do understand where the reviewer is coming from. While Julia was not the main tool for production here, we mention it because the SciML toolset, co-developed by two of the coauthors, may be valuable to ecosystem modelers. Moreover, it is formulated very

differently in a novel symbolic format which is in fact quite interesting and could potentially lead to an alternative path, and the package is evolving rapidly. Hence, we think preserving it has some value. Removing it would also mean removing some coauthors, which we do not want to do because their input was valuable for the development of this work and thus should be credited.

6.  line 183: you don't describe anywhere in your data how many PFTs you consider. it is therefore here also not clear how many dummy variables this model receives as input.

Our dataset included 9 different PFT categories. A paragraph {lines 273: 285} with more details about the Lin15 dataset was added to subsection 2.4.1 (**Forcing and Photosynthesis rates**) stating the number of PFTs considered plus the name of each PFT.

*"We refer to this dataset as Lin15 throughout the rest of this work with 43 sites chosen whose dates and times of measurements were available. Lin15 covered nine different PFT categories: rainfed crop "Crop R", Broadleaf Evergreen Tree Tropical "BET Tropical", Broadleaf Evergreen Tree Temperate "BET Temperate", C3 grass, C4 grass, Needleleaf Evergreen Tree Boreal "NET Boreal", Needleleaf Evergreen Tree Temperate "NET Temperate", Broadleaf Deciduous Tree Temperate "BDT Temperate", and Broadleaf Deciduous Shrub Temperate "BDS Temperate". Measurements were taken on a sub-hourly scale but not necessarily on a continuous daily interval. That's why for almost all the sites, data were available on some random days (not necessarily continuous) in one or a few years.*

*Lin15 also contained forcing variables, including air temperature (T), leaf temperature ($T_v$), atmospheric pressure ($P_{atm}$), relative humidity (RH), photosynthetic active radiation (φ) and boundary layer conductance ($g_b$). Moreover, we used ERA5 to fill in for any missing forcing variables in Lin15. In equation 4, $P_{atm}$ and $g_b$ were used directly from the dataset, while $C_a$ was computed as 0.039% of $P_{atm}$, and $g_s$ was calculated using the Medlyn conductance model (Medlyn et al., 2011) as explained in Appendix A."*

7.  line 190-205: I think this information is not needed to understand the main message

This information is important because it is referred to in different parts of the paper and briefly shows how the soil water stress function ($β_t$) is calculated.

Lines 190 – 195 (in the previous manuscript version) show equation 7 which we later refer to as the equation to be replaced with equation 10 in the model changes section. Thus, we need to mention both the default and the changed equations.

Lines 195 – 205 (in the previous manuscript version) show the equations for calculating the soil water stress function ($β_t$) and the plant wilting factor ($w_i$), which we later refer to as part of the equations used in the synthetic and real data experiments after retrieving or estimating the parameter $B_i$. We also refer to the changed in $w_i$ computations (see Equation 12) in the Model changes section in the revised manuscript.

8.  eq. 10: why is psi_max replaced by psi_0? (missing explanation)

In lines 216 – 218 (in the previous manuscript version), we stated the actual equations that we used in for computing $\psi_i$ (in which $\psi_{sat}$ was replaced with $\psi_o$).

In Appendix A and B (added in the revised manuscript), we kept all the original equations the same whether those related to FATES or to computing the soil water stress function ($\beta_t$).

| Proposed equation in this study (Equation 10) | Original equation (Equation 7) |
|---|---|
| $\Psi_i(PFT) = \Psi_o \times S_i^{-B_i(soil,PFT)} \geq \Psi_c$ | $\Psi_i = \Psi_{sat,i} \times S_i^{-Bi} \geq \Psi_c$ |

Reasons for this replacement:

In the original CLM4.5 equations, $\psi_{sat}$ is based on empirical functions, percentage of sand *(%sand)*, and fraction of organic matter ($F_{om}$) (see Equations B4 – B5 in Appendix B). Using the original Equation 7 for computing $\psi_i$ results in a plant wilting factor $w_i$ equal to one for more than 90% of the data points across different soil layers.

To give the model more flexibility in the computation of $\psi_i$ and thus allow more variability in $w_i$ values, $\psi_{sat}$ was replaced with $\psi_o$. However, to ensure having $w_i$ values less than or equal 1 as in the original $w_i$ (see Equation 8 in the revised manuscript), we tried to create equation 10 in a way that satisfies this condition using $\psi_o$. Parameter $B_i$ (output from $NN_{Bi}$) was restricted to be within the range 0 and 1 to satisfy the same condition as well (see the added paragraph for NN structures in section 2.5). Applying those changes, we were able to get $\psi_i$ values within the range of $\psi_o$ and $\psi_c$ while showing more variability in the computed $w_i$.

Also, we added this paragraph {lines 252: 264} to section 2.3.1 (**Model changes**) for clarification:

"*The default equations in the Community Land model V4.5 (CLM4.5) for computations of $B_i$ (Appendix B) show that the parameter $B_i$ depends on two attributes, %clay$_i$ and $F_{om,i}$, which is why they were used in $NN_{Bi}$. To account for the dependence of $\psi_{sat,i}$ on %sand$_i$ (Appendix B) and its replacement by $\psi_o$ (see equations 7 and 10), %sand$_i$ was also added to $NN_{Bi}$. We also added PFT to $NN_{Bi}$ inputs because vegetation may interact with soil moisture constraint and we want to allow relevant factors to be included, rather than restricting the list of inputs to what was previously used in the literature. Since in $NN_{Bi}$, we use quantitative inputs (%sand$_i$, %clay$_i$, $F_{om,i}$) along with categorical inputs (PFT), we used the embedding layer in PyTorch, which translates each category to a vector of quantitative variables. This categorical data can then easily be combined with other quantitative inputs we provide to our neural network.*

*Moreover, using the original Equation 7 for computing $\psi_i$ resulted in a plant wilting factor, $w_i$, equal to one for more than 90% of the datapoints across different soil layers. Changing Equation 7 to the form shown in Equation 10 helped to express more variability in $w_i$ and eventually in the computed soil water stress function ($\beta_t$).*

Here, the point is to calculate photosynthesis. We can see clearly the modified model works very well for photosynthesis. The differentiable modeling approach was specifically designed to enable inspection of various modules and assumptions in the model to improve model performance. It is possible that alternative formulations can also perform well, and we do not preclude that here.

9. eq. 11: what is F_om?

F_om is the fraction of organic matter. We explained it clearly right after the equation in the revised manuscript {lines 247: 249}as the following:

*"$B_i$ is a PFT- and soil-texture-dependent shape parameter (between 0 and 1) estimated as:*
 *$B_i = NN_{Bi}(\%sand_i, \%clay_i, F_{om,i}, PFT)$                                     (11)*
*where $\%sand_i$, $\%clay_i$, and $F_{Im,i}$ respectively represent the percentage of sand, the percentage of clay, and the fraction of organic matter in soil layer i.."*

10. line 232: the CLM4.5 data points should be documented in a dedicated data section. In general, I suggest they you separate the description of data and experiments

Our data are now described in Section 2.4, preceding the description of our experiments in Section 2.5. CLM4.5 documentation clearly presents the standard values of the parameters and the equations that we used in this study, and a detailed discussion of them is outside the scope of this work. We already did provide some of the basic parameter values (model values for $V_{c,max25}$) from CLM4.5 and other similar models in Table 3 and provided references (in the text). We also added a subsection (shown below) to section 2.4 **(Input and observation datasets)** to better clarify. A subsection (shown in the response to major question 3, above) was also added to Section 2.4 **(Input and observation datasets)**.

11. line 239: were all calculations performed only for the topsoil layer in all experiments?

This is valid for the synthetic case only whose purpose was just to test the whole framework, while for the real case all the five soil layers (mentioned in Static attributes subsection) were used to estimate the parameter $B_i$ for each soil layer. For better clarification we added these paragraphs to section 2.5 **(Synthetic data and real experiments):**

For the synthetic case {lines 314: 320}:

*"In the second synthetic case, "$V_{c,max} - B$", we tested simultaneously retrieving both $V_{c,max25}$ and B , the latter of which varies spatially for different static attributes. For simplicity, we used only the topsoil layer for this case and excluded the influence of the PFT term; therefore we assumed $B_1 = 0.1 * F_{om,1} + 0.45 * (\%sand_1 + \%clay_1)$ to generate the synthetic data. The plant wilting factor ($w_1$) was then calculated using equation 12 and was fed into equation 9 to compute the soil water stress function ($\beta_t$). Since we were using only the topsoil layer, $\beta_t$ was simplified to ($\beta_t = w_1 r_1$) with a root distribution value for the topsoil layer ($r_1 = 1$). To retrieve $B_1$, we used $NN_{Bi}$ (see Equation 11) but excluded the PFT term since it was not used in synthesizing $B_1$ values."*

For the real case {lines 348: 349}:

*"Representing a real case, $B_i$ was estimated for the i-th soil layer based on the static attributes for that layer in the four tested model formulations. Thus, $B_i$ varied both horizontally and vertically for each PFT. "*

**12. Table 1: missing symbol explanations for means and standard deviations**

For clarification, this line was added to the bottom of table 1

"$\sigma$ refers to the standard deviation, $\overline{OBS}$ refers to the mean of the observed values, and $\overline{SIM}$ refers to the mean of the simulated values."

**13. line 383: please include time series for observations and model predictions**

Time series are not the focus for several reasons. Measurements in Lin15 dataset were taken on sub-hourly scale but not necessarily on a continuous daily interval. For almost all the sites, data were available on some random days (not necessarily continuous) in one or a few years. In fact, many of the measurement days are far from each other and we can barely find consecutive days for producing sensible time series. Second, this model was not posed as a time-continuous problem. In other words, there is no accumulated memory between different dates. Hence, we think a time series plot would be somewhat misleading. We believe more effort, including a vegetation growth module, is needed to simulate seasonality nearly optimally.

**14. fig. 5: symbols in legend cannot be distinguished. are results shown for the test dataset?**

These points belong to both training and testing datasets. We previously had a version that distinguished between train and test, as pasted below. As you can see, there are no visual differences between two types of points and such symbology does not really bring in new information. Later, we wanted to use symbols to indicate PFTs, which seems more informative. So, to avoid overcomplicating the figure, we removed the train/test differences. We also remind the reviewer that we provide cross validation results in the revised manuscript, which shows similar statistics to the random holdout.

We already ran the requested cross validation (5-fold). The figures below show train/test points from the five folds. As one can see, there seems no systematic difference between train and test in the cross-validation case.

Fold 1:

[Figure]

Fold 2:

[Figure]

Fold 3:

[Figure]

Fold 4:

[Figure]

Fold 5:

[Figure]

In the revised manuscript, we used Figure 6 shown below since we believe that the figure with different PFT types, delivers more useful information. To avoid confusion between whether training or testing sets were used for plotting, we created this figure using the test points from the 5 folds in the cross-validation test (cross validated dataset).

[Figure]

**Figure 6. Comparisons of photosynthesis model calibration. Comparing impacts of default and learned parameters by plotting observed vs. simulated $A_n$ (net photosynthetic rate) values calculated using different candidate models (described by which parameter definitions they use). (a) Impact of learning B with default $V_{c,max25}$. (b) Impact of learning $V_{c,max25}$ with varying B (either learned alongside V in V+B, or defined by the default equations in CLM4.5. The colors represent the results from the four different models, the shapes indicate the plant functional type (PFT) groups, and the dotted line in each panel indicates the ideal 1:1 relationship. Subscript "def" indicates that the variable was calculated using the default definitions in CLM4.5, while lack of this subscript indicates that the parameter was learned using a NN. Scatter plots were created using the test dataset from the 5 folds of the cross-validation test. For better illustration, only 3 PFTs are placed in a panel, as indicated by the panel titles. Comparing symbols in the same panel gives insights about the role of estimating B, while comparing left and right panels gives insights about the role of estimating $V_{c,max25}$.**

15. line 426: i would add that you have identified parameter values that are optimized for the considered set of model equations and forcings. both of these have limitations. Equations may be wrong, ERA5 is rather uncertain, and measurement principles can vary between stations. This is both a limitation and a strength of your framework. Parameter values will not be

transferable to other inputs. On the other hand you can obtain optimized predictions for the given set of forcings.

Good point. Just like any other model, the performance may be impacted when you change the forcing datasets because these datasets may have certain biases. If the model is trained on a global scale, we hope the various different kinds of forcings to be encountered can serve to limit overfitting. We added the following sentences {lines 617: 620} to section 4 (Discussion).

*"We would like to highlight that such parameterizations are suitable to the target and forcing dataset used in training (which is still the most representative accessible dataset) and are related to the process-based model employed. The dataset may have limitations related to the consistency in the measurement approach, and there may be errors in the forcing data, or imperfections in model structure. The model performance may also vary based on different forcing data and inputs used."*

Anonymous Referee #2
The authors of the manuscript 'A differentiable ecosystem modeling framework for large-scale inverse problems: demonstration with photosynthesis simulations' describe the application of the 'differentiable parameter learning'(dPL) framework to the photosynthesis module of FATES model. The framework, and concept, overcomes extrapolation limitations from site-by-site calibration approaches and allows leveraging information content in large-scale datasets towards a global parameterization of photosynthesis models. Neither the concept (Tsai et al., Nature Communications, https://www.nature.com/articles/s41467-021-26107-z, 2021; Bao et al., Authorea, https://www.authorea.com/doi/full/10.1002/essoar.10512186.3, 2022) nor the dPL framework (Tsai et al., 2021; Feng et al., 2022ab) are new. However, the framework is used in the FATES model for the first time and the results would be of interest for further model development, but also to the scientific community at large.
At this point, the experiment focuses on inverting two parameters, $V_{cmax25}$ and B, resulting in that the accuracy of the simulated net photosynthesis rate being slightly improved. The main concerns at this stage relate to apparently incorrect formulations of some key equations, to issues about the validation strategy, to the fact that the forcing data and the experiments are not described sufficiently, challenging the acceptance of the study, while hampering any reproducibility efforts. Please see below for details.

We appreciate your detailed comments! As a summary, it seems most of the questions seek clarifications and details about the model. Thank you – these comments should help us elucidate the model better. We did not find major comments that require computational experiments or major reorganization. There is a question about cross validation, which we have already run and shows expected and essentially similar results. Moreover, some metrics were requested, and we calculated them and reported them in the responses.

We indeed followed our previous differentiable parameter learning paradigm which was first applied in hydrology (Tsai et al., 2021; Feng et al., 2022), as noted in the manuscript, but this is a novel use in the large domain of ecosystem modeling, which is a very large field of study. The system is also different as here we have a nonlinear system of equations while in hydrologic cases, we have ordinary differential equations; thus the mathematical treatment is different. The Julia software solves the system using adjoint solvers, although it is a relatively minor point as we mainly used the PyTorch version for its high parallel efficiency.

We could not have noticed Bao et al., 2022 as it went online after our manuscript did and seems to be undergoing review. Upon some examination, we believe the basic modules are very different. They are using a light-use-efficiency approach and predicted GPP, while our paper focused on photosynthesis using a Farquhar-type model. Hence, we don't think there is much overlap between the two.

1. Two key equations are incorrect in the paper:
1) line 140: equation 5, $C_i=C_a-A_n*P_{atm}*(1.4g_s+1.6g_b)/(g_s+g_b)$;
2) line 505: equation A1, $A_c=V_{cmax}*(C_i-\Gamma_*)/(C_i+K_c*(1+K_o/O_i))$.
According to the user guide of the FATES model (https://fates-users-guide.readthedocs.io/projects/tech-doc/en/latest/fates_tech_note.html#fundamental photosynthetic-physiology-theory), the equations should be:

1) $C_i=C_a-A_n*P_{atm}*(1.4g_s+1.6g_b)/(g_s*g_b)$;
2) $A_c=V_{cmax}*(C_i-\Gamma_*)/(C_i+K_c*(1+O_i/K_o))$.
Since the FATES model is reimplemented in Julia and PyTorch by the authors, the codes might be also wrong. If so, the unit of $C_i$ will be incorrect, leading to errors in the inversion of $V_{cmax25}$ and B. The wrong computation of the effective Michaelis-Menten coefficient ($=K_c*(1+O_i/K_o)$) might only have a slight effect if the temperature is close to 25°C, but should be concerned if the temperature is too low or high (and I do see some points with low leaf temperature in the 'Lin15' database). Thus, I have doubts about the current results and relevant analysis.

Regarding the equations --- we were cautious to adhere to the original FATES equations before implementing it on PyTorch or Julia. Unfortunately, **there were some typos in the manuscript** in line 140 and line 505 in the paper which have been corrected. We confirmed that we used the correct equations in our differentiable model:
$C_i=C_a-A_n*P_{atm}*(1.4g_s+1.6g_b)/(g_s*g_b)$;
$A_c=V_{cmax}*(C_i-\Gamma_*)/(C_i+K_c*(1+O_i/K_o))$.

No results need to be changed. The code was correct as we compared carefully against the Fortran code in these subroutines as we developed the differentiable versions of the code. We will be publishing the code as the paper gets closer to acceptance so this can be examined in the code. Again, we apologize for the errors in the manuscript.

2. As all the results are validated only once using the temporal holdout data or the random holdout data, the generalizability of the dPL (or $NN_B+NN_v$) is not clear. If the N-fold or leave-one-out cross-validation can be adopted, the statistical metrics can be more justifiable to reflect the model performance.

Thanks for being rigorous. We believe the randomly selected points were representative, but we conducted a cross validation (CV) and show the results below. The results are as follows:

(a) Temporal holdout test for the following system (80% train: 20% test)

| Runs | Corr | | RMSE (µmol m-2 s-1) | | Bias (µmol m-2 s-1) | | NSE | |
|---|---|---|---|---|---|---|---|---|
| | Train | Test | Train | Test | Train | Test | Train | Test |
| Vdef+Bdef | 0.565 | | 6.778 | | 1.475 | | 0.042 | |
| Vdef+B | 0.631 | 0.582 | 6.339 | 6.110 | 1.521 | 0.944 | 0.176 | 0.170 |
| V+Bdef | 0.758 | 0.565 | 4.598 | 6.135 | -0.164 | -1.624 | 0.566 | 0.163 |
| V+B | 0.786 | 0.771 | 4.319 | 4.296 | 0.102 | -0.226 | 0.617 | 0.590 |

(b) Cross Validation (5-fold) test for the following system

| Runs | Corr | | RMSE (μmol m-2 s-1) | | Bias (μmol m-2 s-1) | | NSE | |
|---|---|---|---|---|---|---|---|---|
| | Train | Test | Train | Test | Train | Test | Train | Test |
| Vdef+Bdef | 0.565 | | 6.778 | | 1.475 | | 0.042 | |
| Vdef+B | 0.623 | 0.621 | 6.281 | 6.298 | 1.584 | 1.578 | 0.177 | 0.173 |
| V+Bdef | 0.715 | 0.709 | 4.960 | 5.020 | -0.410 | -0.401 | 0.487 | 0.474 |
| V+B | 0.783 | 0.778 | 4.306 | 4.359 | 0.074 | 0.081 | 0.613 | 0.604 |

We also provide the metrics for each fold:

| Folds | COR_test | RMS_test | BIAS_test | NSE_test | |
|---|---|---|---|---|---|
| 1 | 0.769 | 4.701 | 0.104 | 0.591 | |
| 2 | 0.781 | 3.856 | 0.388 | 0.605 | V+B |
| 3 | 0.789 | 4.108 | -0.146 | 0.622 | |
| 4 | 0.767 | 4.654 | 0.072 | 0.584 | |
| 5 | 0.784 | 4.417 | -0.035 | 0.615 | |

This is exactly as we expected in our initial reply posted online --- the 5-fold CV results are similar to the previous random results and better than the temporal test results. In addition, we showed the train/test $A_n$ scatter plots for the five folds in question 14 above (in the detailed comments) by the first reviewer.

We believe a spatial test, though, would best belong to a different paper as the paper is already getting long. There are many techniques to improve spatial generalization and enlarge datasets using remote sensing which, if combined with the present content, would just be too much for a first paper. We clarify this point in the paper by adding the following sentences {lines 611: 615} to section 4 (Discussion):

"*Also, this study doesn't cover the spatial generalization since we don't present results for spatial tests or based on site-level comparison. To improve spatial generalization may require further changes in the model, dynamical parameters, or using other error mitigation approaches (Feng et al., 2021, 2022b; Ma et al., 2021a). This is not our scope for this study; however, it will be considered for future work.*"

3.  The forcing variables and parameters are not clearly differentiated in the paper. For example, is the leaf layer boundary conductance, $g_b$, a constant parameter across sites or a temporally changing variable? If it is a forcing variable for FATES, where is $g_b$ from? is $\theta_{ice}$ a forcing variable or a parameter correlated with temperature and $\theta_{liq}$? Is the $C_a$ a constant value or variable? The model would be different if the spatial and temporal variability of all these factors are considered. If all these are parameters (i.e., scalars), what are the values?

The Lin15 dataset included different forcing variables that we used in our model including:

| RH | Relative humidity |
|---|---|
| T | Air temperature |
| $T_v$ | Leaf temperature |
| $P_{atm}$ | Atmospheric pressure |
| PAR ($\varphi$) | Photosynthetic active radiation |
| $g_b$ | Boundary layer conductance |

Concerning ($g_b$, $\theta_{ice}$ and $C_a$), here are details about how they were considered in the model:

- $g_b$, the boundary layer conductance values were already available in Lin15 dataset. However, it has some missing values which we then computed using the inverse relationship between $g_b$ and the boundary layer resistance $r_b$. $r_b$ was approximated by the following equation as documented in CLM5.0 (Lawrence et al., 2019) in section 5.1:

$$r_b = \frac{1}{C_v} \sqrt{\frac{d_{leaf}}{U_{av}}}$$

Where $C_v$ and $d_{leaf}$ are both constants (0.01 ms$^{-1/2}$ and 0.04 m respectively), while $U_{av}$ is the wind velocity. We stated this in Appendix A {lines 675: 679} as the following:

"*where $C_a$ is $CO_2$ partial pressure near the leaf surface (calculated as 0.039% of $P_{atm}$) and $g_b$ is the leaf boundary layer conductance, which was available in Lin15 except for some missing values which were computed using the inverse relationship between $g_b$ and the boundary layer resistance ($r_b$). $r_b$ was approximated by the following equation as documented in section 5.1 of CLM5.0 (Lawrence et al., 2019):*

$$r_b = \frac{1}{C_v} * \sqrt{\frac{d_{leaf}}{U_{av}}} \qquad (A13)$$

*where $C_v$ and $d_{leaf}$ are both constants (0.01 ms$^{-1/2}$ and 0.04 m respectively), while $U_{av}$ is the wind velocity.*"

- $\theta_{ice}$, the volumetric ice content values were ignored (considered as zero) since both the air and leaf temperatures in our dataset were above the freezing temperature (0 °C or 273.15 K) by at least 5 degrees. We stated this in section 2.3 {lines 236: 238} as the following:
"*In our calculations, $\theta_{ice}$ was ignored since both the leaf and the air temperatures in our dataset were above the freezing temperature (0 °C or 273.15 K) by at least 5 °C.*"

- $C_a$, the $CO_2$ partial pressure near the leaf surface values were variable spatially and temporally and they were taken as 0.039% of the atmospheric pressure. We stated this in section 2.4 {lines 273: 285} as shown below.

"*We refer to this dataset as Lin15 throughout the rest of this work with 43 sites chosen whose dates and times of measurements were available. Lin15 covered nine different PFT categories: rainfed crop "Crop R", Broadleaf Evergreen Tree Tropical "BET Tropical", Broadleaf Evergreen Tree Temperate "BET Temperate", C3 grass, C4 grass, Needleleaf Evergreen Tree Boreal "NET Boreal", Needleleaf Evergreen Tree Temperate "NET Temperate", Broadleaf Deciduous Tree*

*Temperate "BDT Temperate", and Broadleaf Deciduous Shrub Temperate "BDS Temperate". Measurements were taken on a sub-hourly scale but not necessarily on a continuous daily interval. That's why for almost all the sites, data were available on some random days (not necessarily continuous) in one or a few years.*

*Lin15 also contained forcing variables, including air temperature (T), leaf temperature ($T_v$), atmospheric pressure ($P_{atm}$), relative humidity (RH), photosynthetic active radiation (φ) and boundary layer conductance ($g_b$). Moreover, we used ERA5 to fill in for any missing forcing variables in Lin15. In equation 4, $P_{atm}$ and $g_b$ were used directly from the dataset, while $C_a$ was computed as 0.039% of $P_{atm}$, and $g_s$ was calculated using the Medlyn conductance model (Medlyn et al., 2011) as explained in Appendix A."*

4. Line 216-218: the reason for replacing saturated soil matric potential (Ψsat) with soil matric potential for closed stomata (Ψc) is not explained. Equation 10 shows that the Ψsat is replaced with soil matric potential for open stomata(Ψo), not Ψc. Furthermore, the Ψi was still calculated using Ψsat in Appendix A (equations A16-A18). I'm confused about which variable was used to calculate Ψi.

In lines 216 – 218 (in the previous manuscript version), we stated the actual equations that we used in for computing $\psi_i$ (in which $\psi_{sat}$ was replaced with $\psi_o$).

In Appendix A and B (added in the revised manuscript), we kept all the original equations the same whether those related to FATES or to computing the soil water stress function ($\beta_t$).

| Proposed equation in this study (Equation 10) | Original equation (Equation 7) |
|---|---|
| $\Psi_i(PFT) = \Psi_o \times S_i^{-B_i(soil,PFT)} \geq \Psi_c$ | $\Psi_i = \Psi_{sat,i} \times S_i^{-B_i} \geq \Psi_c$ |

Reasons for this replacement:

In the original CLM4.5 equations, $\psi_{sat}$ is based on empirical functions, percentage of sand *(%sand)*, and fraction of organic matter ($F_{om}$) (see Equations B4 – B5 in Appendix B). Using the original Equation 7 for computing $\psi_i$ results in a plant wilting factor $w_i$ equal to one for more than 90% of the data points across different soil layers.

To give the model more flexibility in the computation of $\psi_i$ and thus allow more variability in $w_i$ values, $\psi_{sat}$ was replaced with $\psi_o$. However, to ensure having $w_i$ values less than or equal 1 as in the original $w_i$ (see Equation 8 in the revised manuscript), we tried to create equation 10 in a way that satisfies this condition using $\psi_o$. Parameter $B_i$ (output from $NN_{Bi}$) was restricted to be within the range 0 and 1 to satisfy the same condition as well (see the added paragraph for NN structures in section 2.5). Applying those changes, we were able to get $\psi_i$ values within the range of $\psi_o$ and $\psi_c$ while showing more variability in the computed $w_i$.

Also, we added this paragraph {lines 252: 264} to section 2.3.1 **(Model changes)** for clarification:

*"The default equations in the Community Land model V4.5 (CLM4.5) for computations of $B_i$ (Appendix B) show that the parameter $B_i$ depends on two attributes, %clay$_i$ and $F_{om,i}$, which is why*

*they were used in $NN_{Bi}$. To account for the dependence of $\psi_{sat,i}$ on $\%sand_i$ (Appendix B) and its replacement by $\psi_o$ (see equations 7 and 10), $\%sand_i$ was also added to $NN_{Bi}$. We also added PFT to $NN_{Bi}$ inputs because vegetation may interact with soil moisture constraint and we want to allow relevant factors to be included, rather than restricting the list of inputs to what was previously used in the literature. Since in $NN_{Bi}$, we use quantitative inputs ($\%sand_i$, $\%clay_i$, $F_{om,i}$) along with categorical inputs (PFT), we used the embedding layer in PyTorch, which translates each category to a vector of quantitative variables. This categorical data can then easily be combined with other quantitative inputs we provide to our neural network.*

*Moreover, using the original Equation 7 for computing $\psi_i$ resulted in a plant wilting factor, $w_i$, equal to one for more than 90% of the datapoints across different soil layers. Changing Equation 7 to the form shown in Equation 10 helped to express more variability in $w_i$ and eventually in the computed soil water stress function ($\beta_t$)."*

Here, the point is to calculate photosynthesis. We can see clearly the modified model works very well for photosynthesis. The differentiable modeling approach was specifically designed to enable inspection of various modules and assumptions in the model to improve model performance. It is possible that alternative formulations can also perform well, and we do not preclude that here, as this is not a main point of concern for this paper.

5.  Line 218-220: is $NN_B$ used to predict $B_i$ or $\Psi_i$? B depends on only %clay and $F_{om}$ according to equations A22-A23, while the authors add %sand, which is related to $\Psi_{sat}$ and, therefore, $\Psi_i$. I didn't find a direct relationship between $B_i$ and %sand according to the original equations in the FATES model. If $NN_B$ is used to predict $\Psi_i$, I think the equation can be $\Psi_i = \theta_{liq} * NN_B(\%sand, \%clay, PFT, F_{om}, T)$, where T represents the factors controlling $\theta_{ice}$, e.g., temperature.

$NN_{Bi}$ is used to predict $B_i$. Indeed, $B_i$ in the original equations depends only on $\%clay_i$ and $F_{om,i}$, however due to the changes we implemented to equation 7 (replacement of $\psi_{sat}$ with $\psi_o$), the $\%sand_i$ was also added to the $NN_{Bi}$. We also added PFT to $NN_{Bi}$ inputs because vegetation may interact with soil moisture constraint and we want to allow relevant factors to be included, rather than restricting the list of inputs to what was previously used in the literature. This is precisely the point of replacing existing equations with NNs --- we can be freed from previous restrictive assumptions and test new ideas rapidly. We discussed the incentives for these changes in the last response above (No.4).

In addition, here, the point is to calculate photosynthesis. We can see clearly the modified model works very well for photosynthesis. The differentiable modeling approach was specifically designed to enable inspection of various modules and assumptions in the model to update the formula, so more modifications will definitely happen in the future.

Concerning this formula, $\Psi_i = \theta_{liq} * NN_B(\%sand, \%clay, PFT, F_{om}, T)$, we would like to thank you for this suggestion. However, we suspect that including T in $NN_{Bi}$ to represent $\theta_{ice}$ might not very effective. The histogram of air temperature (shown in the figure below) indicates that our dataset does not include any points with air temperatures below 5 °C, which clarifies why $\theta_{ice}$ was ignored in our calculations. Thus, there is low probability that the temperature would have a great effect if

included in $NN_{Bi}$. Further investigation would be outside of the scope of this work, but we agree that more data collection and investigation into air temperature and its effects in the future may be worthwhile.

[Figure]

6. I think the neural networks ($NN_B$ and $NN_v$) need constraints on $V_{cmax25}$ and $\Psi_i$. Although the authors declared that the predicted $V_{cmax25}$ without any constraints is within a rational range similar to the literature and measurement, the range of the predicted B is not discussed. If the predicted $B_i$ is very large at some point, $\Psi_i$ can be much higher than $\Psi_o$, leading to $w_i$ being higher than 1 (i.e., exceeding the range defined in equation A15). Besides, the $V_{cmax25}$ is possibly to be inappropriate without any physical constraints at sites not considered in this study.

We actually did impose constraints on both $NN_{Bi}$ and $NN_v$ in predicting $V_{c,max25}$ and B.
For $V_{c,max25}$:
We constrained the output of $NN_V$ to be between 0 and 1 using a sigmoid activation function for the output layer in the NN. We then rescaled the output to be within a pre-defined range based on literature of minimum value of 20 umol m$^{-2}$ s$^{-1}$ to a maximum value of 150 umol m$^{-2}$ s$^{-1}$.

For B:
We constrained the output of $NN_{Bi}$ to be between [0 , 1] using a sigmoid activation function for the output layer in the NN. Given that the soil wetness $S_i$ (in equation 7 and 10) ranges between [0.01,1] as defined in the original CLM4.5 equations, therefore the term $S_i^{-Bi}$ can have a range of [1, 100] which when multiplied by $\psi_o$ ensures having $\psi_i$ values with a maximum limit of $\psi_o$, while the condition of $\psi_i >= \psi_c$ was conserved in equation 10 (same as equation 7) for ensuring a minimum limit of $\psi_c$. Thus, we ensured that $\psi_i$ computed using equation 10 is within the range of $\psi_o$ and $\psi_c$ which resulted in $w_i$ values less than or equal to 1.

The following paragraph {lines 356: 360} was added to section 2.5 (**Synthetic data and real data experiments**) which states some details in this regard:

*"The MLPs employed had three layers: an input layer, one hidden layer, and an output layer. To ensure an output value between 0 and 1 for both $V_{c,max25}$ and B parameterizations, sigmoid activation functions were used for both hidden and output layers. $V_{c,max25}$ was then rescaled to be within a pre-defined range based on literature values of 20 to 150 μmol $m^{-2}$ $s^{-1}$. For the i-th soil layer, $B_i$ values were kept between 0 and 1, so with $S_i$ ranging between 0.01 and 1 (see Appendix B), the term $S_i^{-Bi}$ then had a range of 1 to 100. This ensured that the value of $\psi_i$ ranges from $\psi_c$ to $\psi_o$ (see Equation 10)."*

Furthermore, some other authors may choose to impose additional constraints, which will be a great research topic to pursue and should reduce the uncertainty of the parameters. Yet this seems unnecessary for the model setup here.

7. Line 235-236: 'we tested retrieving both $V_{cmax25}$ and B, the latter of which varies spatially and temporally.' If B varies temporally, it should be clarified how the training data is partitioned and how the 'random holdout test' is done. For example, is B changing per year or every N years? how many years/points per site are used to estimate B? Do the training points have to be in sequence or not?

In the referenced lines, we specifically refer to the synthetic case. For this case, the values for the parameter $B_1$ in the topsoil layer were synthesized using the following equation $B_1 = 0.1 * F_{om,1} + 0.45 * (\%sand_1 + \%clay_1)$, so $B_1$ only varies spatially (different static attributes). We modified this sentence {lines 314: 315} to be "*we tested simultaneously retrieving both $V_{c,max25}$ and B , the latter of which varies spatially for different static attributes*".

Moreover, this paragraph {lines 371: 380} was added in section 2.5 (**Synthetic data and real data experiments**) to explain more about the tests held as well as data splitting.

*"Two different tests were performed with respect to data splitting: temporal holdout and randomized cross-validation --- the former test stresses the models' ability to project into the future while the latter is the typical experiment run in the literature. Due to the irregularity of measurement dates at each location (as mentioned previously in section 2.4.1), the temporal periods for the training and testing datasets varied by location. In the temporal holdout test, for each PFT in each location, the available dates of measurements were recorded. The oldest 80% of these dates were used for training and the remaining more recent 20% were used for testing. The temporal holdout test was run for both synthetic and real data experiments. For the randomized cross-validation test, as the name implies, the dataset was randomly split into 5 folds (groups) and each time the model was trained on 4 folds (80% of the datapoints) and tested on the $5^{th}$ fold (20% of the data points). This was done a total of 5 rounds, so that all of the data points were used for testing once. The cross-validation test was run only for the real data experiments."*

8. Line 238-239: 'For simplicity, the computations of B, $\Psi_i$, $w_i$, $\beta_t$ were performed for the top soil layer only.' In the synthetic experiment, only the top soil layer is considered. However, 'B, $\Psi_i$, $w_i$' for the other layers are not clarified (=zero or default values in CLM?). Are the other soil

layers considered in the real data experiment? If yes, how many 'B' was estimated (i.e., how many soil layers and how many temporally changing $B_i$)? If not, $w_i$ can only represent the water availability at the top layer. The $\beta_t$ is equal to $w_i$ and the root distribution, $r_i$, at the top layer. What is $r_i$ at the top layer (soil depth=0cm according to line 306)?

We added further explanation for the synthetic case {lines 314: 320} in section 2.5 (**Synthetic data and real data experiments**):

*"In the second synthetic case, "$V_{c,max} - B$", we tested simultaneously retrieving both $V_{c,max25}$ and $B$, the latter of which varies spatially for different static attributes. For simplicity, we used only the topsoil layer for this case and excluded the influence of the PFT term; therefore we assumed $B_1 = 0.1 * F_{om,1} + 0.45 * (\%sand_1 + \%clay_1)$ to generate the synthetic data. The plant wilting factor ($w_1$) was then calculated using equation 12 and was fed into equation 9 to compute the soil water stress function ($\beta_t$). Since we were using only the topsoil layer, $\beta_t$ was simplified to ($\beta_t = w_1 r_1$) with a root distribution value for the topsoil layer ($r_1 = 1$). To retrieve $B_1$, we used $NN_{Bi}$ (see Equation 11) but excluded the PFT term since it was not used in synthesizing $B_1$ values."*

For the real experiments:
Five soil layers were considered in these experiments with the exact depths described in subsection 2.4.2 (**Static attributes**). $NN_{Bi}$ used static attributes ($F_{om,i}$, $\%sand_i$, and $\%clay_i$) from all soil layers along with PFT and predicted $B_i$ values for each layer. So according to $B_i = NN_{Bi}(\%clay_i, \%sand_i, F_{om,i}, PFT)$, $B$ varies horizontally (static attributes per location) as well as vertically for each soil layer, and for each PFT. For better clarification, we edited the equation for $B$, which now reads $B_i = NN_{Bi}(\%clay_i, \%sand_i, F_{om,i}, PFT)$ (see Equation 11 in the revised manuscript)

We added further explanation for the real case {lines 348: 349} in section 2.5 (**Synthetic data and real data experiments**):

*"Representing a real case, $B_i$ was estimated for the i-th soil layer based on the static attributes for that layer in the four tested model formulations. Thus, $B_i$ varied both horizontally and vertically for each PFT."*

9. Line 239: 'To retrieve B, we used NNB but exclude the PFT term.'
I think it is not proper if the PFT is excluded from the training but included in the equation. If PFT is excluded, the term should be removed from equation 11. The sentence at line 222 '… along with categorical inputs (PFT), we used…' should be rephrased.

In Line 239 (in the previous manuscript version), we refer to the synthetic case and as mentioned in the last comment (No.8) we synthesized the values for the parameter $B_1$ in the topsoil layer using the following equation $B_1 = 0.1 * F_{om,1} + 0.45 * (\%sand_1 + \%clay_1)$. So, we formulated the $NN_{B1}$ for the synthetic case as $B_1 = NN_{B1}(\%sand_1, \%clay_1, F_{om,1})$.

In equation 11, we show the general equation used for the real case experiments which included the PFT term as well in $NN_{Bi}$ (discussed previously in comment No.4 in the major comments)

10. Line 245: 'The model passing the test of the synthetic case was then applied to a real dataset…'
The same NN was used for synthetic data and real data, but the NN information (layers, neurons activation functions) is not clear. As real data is much more complex, using a different NN structure from the synthetic test might have better performance.

**Concerning the NN formation,** as mentioned above in our response to major question 6, we added a new first paragraph to section 2.5 **(Synthetic data and real data experiments)** which states some details in this regard. That paragraph {lines 362: 369} is followed by:

*"The quantity of available data posed a limitation and did not permit an extensive hyperparameter tuning experiment with a train/validation/test split. Hence, we employed a "lazy" trial and error process with hyperparameters (learning rates and hidden size) using 70% of the data as training data and 30% as a validation set, just to ensure we had a roughly performing hyperparameter set (see Appendix C). We selected a learning rate of 0.045 and a hidden size equal to the number of inputs (9 for the $NN_V$ and 8 for the $NN_{Bi}$). We kept these same hyperparameters when we ran 5-fold cross validation with an 80%:20% train:test ratio. In addition, we found that moderately perturbing the hyperparameters resulted in very little change in the performance. This design was necessary considering the practical limits of the available data, even though this study already represents a large-sample study in the domain of ecosystem modeling."*

What we meant by "The model passing the test of the synthetic case was then applied to a real dataset…" is that we didn't perform significant changes in the general differentiable model structure between running the synthetic and real cases.

Indeed, it is true that the real case should be more complex than the synthetic case. However, for $NN_V$ we kept it the same for both cases since in our reference models (CLM4.5, AVIM, BETHY) $V_{c,max25}$ is a PFT-dependent parameter and for consistency we didn't make any changes to $NN_V$ (in this paper, as a starting point). For $NN_{Bi}$, we indeed made a slight change between the synthetic and the real case regarding the number of soil layers used:

| Synthetic Case (one soil layer) | Real Case (five soil layers) |
|---|---|
| $B_1 = NN_{B1}$ (%sand$_1$, %clay$_1$, $F_{om,1}$) | $B_i = NN_{Bi}$ (%sand$_i$, %clay$_i$, $F_{om,i}$, PFT) |

We discussed the reasons for using these inputs in $NN_{Bi}$ in comment (No.4) in the major comments.

Finally, we would like to mention that this study is one of the first studies in this field, so our purpose is to present the application of the dPL framework without necessarily finding the best NNs for learning our target parameters. We can perform more improvements in the parameterization module in the future.

11. Line 266-267: the loss function is very significant to evaluate the NN, but not explained in the paper. Without the equation of the loss function or the NN information, the dPL framework cannot be assessed by others, in other words, the experiment cannot be repeated. I think this doesn't fulfil the requirement of Biogeosciences: 'Is the description of experiments and calculations sufficiently complete and precise to allow their reproduction by fellow scientists (traceability of results)?'.

**Concerning the loss function,** we already discussed its structure in multiple sections in the manuscript.

$$W = \underset{W}{\mathrm{argmin}}(L(\delta_{psn}(g^W(R), \theta_c, F), y^*)) \tag{3}$$

In equation 3: we stated that the weights are minimized using the loss function between the simulated target variable y (see Equation 2) and the observed target variable $y^*$. We then discussed how f and h equations are reflected on the photosynthesis module in FATES using equations 4 and equation 5 respectively. In section 2.2, we highlighted that the y term is the $A_n$ (the net photosynthesis rate) variable in our problem.

Moreover, figure 1b (new version shown in question 1 above by the first reviewer) shows that the loss function is computed between the simulated and the observed $A_n$. We mentioned that for the dPL framework, we don't need ground truth for the learnt parameters but we do for $A_n$.

**Concerning the NN formation**, the paragraph we added in the last response to comment No.10 in the major comments would further clarify it. Further, our code will be shared upon paper acceptance and the results will be entirely reproducible.

12. Line 268-272: the authors 'hope to identify parameters that can generalize well in space', so I think the readers would wonder if the parameters are estimated per site or per PFT. If parameters are estimated per site, how are they aggregated to parameters per PFT in figure 3a and 4a? If estimated per PFT, I'm afraid the spatial variability of the parameters is not fully captured by dPL.

$V_{c,max25}$ values were estimated per PFT since $NN_V$(PFT) uses just the PFT as input without any static attributes specific to each site. Also, our reference values (used for comparison from CLM4.5, AVIM, BETHY, and TRY) for $V_{c,max25}$ come from models that define $V_{c,max25}$ per PFT not per site. While $B_i$ is considered a spatially variable parameter since it depends on site specific soil attributes. According to this equation $B_i = NN_{Bi}(\%sand_i, \%clay_i, F_{om,i}, PFT)$, (equation 11 in the revised manuscript), $B_i$ differs between different sites and for one site it differs for different PFTs as well.

'hope to identify parameters that can generalize well in space': by this sentence we mean that the dPL, contrary to previous site by site calibration, is able to learn from data from all sites simultaneously. This is due to the structure of the framework enabling it to be trained "globally" so it involves all training data points, rather than being computed site-by-site as done in traditional calibration. In Tsai et al., 2021 we have already established that casting the parameter problem as parameter learning improves spatial generalization.

Further, we have run some preliminary spatial tests which showed only a small decline of performance when tested in an untrained site. While we obtained a temporal test NSE of 0.581 (80%:20%) train: test ratio, the NSE of a spatial test for the current model is already 0.44, suggesting this model is reasonably well-generalized in space. Unfortunately, we could not find spatial tests for benchmarking in the ecosystem modeling literature and would appreciate any suggestions with a comparable dataset. As we mentioned earlier, we are working on further

improving the spatial generalization with some error mitigation approaches. This will add lots of content and should be for the scope of another paper.

13. Line 292-302: the sources of the soil moisture, stomatal conductance, meteorological forcings and the soil properties are mentioned, but the sources of Ca, gb and Patm are not clear.

We mentioned the paragraphs added to clarify this in comment No.3 in the major comments. While these represent simplified treatments, our model's performance suggest that their impacts may be limited. Such simplifications are necessary as we just are getting started with the different model, and the model can be made more sophisticated later.

14. The data source of 'Lin15' was not specified. I found a database at Lin et al., 2015, bud didn't find the dates information on lines 296-300.
In the supplementary information Lin et al., 2015, page 6:

"Supplementary Table 2: List of data source. The whole database is publicly available and can be downloaded from data repository 40 (https://bitbucket.org/gsglobal/leafgasexchange)."

So, they direct the readers to the database (https://bitbucket.org/gsglobal/leafgasexchange) which has the full parameters listed including dates, species, and other variables.

15. Line 304-305: the soil organic carbon content is collected, but the unit is not explained. Does the unit need to be transferred to get the soil organic matter fraction?

Yes, we had to do some unit conversion. According to https://zenodo.org/record/2525553#.Y9Ida-zMKb0 the soil organic carbon is given in 5 g/kg so two conversions were done:
1. Divide by 2 (to convert to %) then divide by 100 (to get a fraction)
2. Multiply by the conventional factor "Van Bemmelen factor" 1.72 (soil organic matter = 1.72 soil organic carbon)

To clarify this, we added this paragraph {lines 287: 290} to subsection 2.4.2 (**Static attributes**):

"*For $\beta_t$ calculations, we used data from Hengl & Wheeler (2018) for the soil organic carbon content at different soil depths, where the conventional Van Bemmelen factor of 1.72 was used to convert to soil organic matter ($F_{om}$). Data for sand and clay percentages (%sand, %clay) were obtained from Hengl (2018). Both are global datasets available at 250 m resolution at 6 different soil depths (0, 10, 30, 60, 100, and 200 cm) which describe five soil layers.*"

16. Line 410-411: the authors claim that the predicted Vcmax25 'were well correlated with' literature values. However, the correlation coefficient or determination coefficient was never stated in the paper. Too few points are displayed in figure 6b, and the distribution pattern of only four PFT types (crop R, C3 grass, NET Boreal and BDS temperate) is similar to CLM.

First, the point here is that the values we estimated make physical sense, are on the same order of magnitude, and are partially correlated with the literature values. We expected there to be some

correlation but not that high. Higher correlation does not necessarily mean it's better. Imagine the extreme case --- if the correlation was 1.0 and every value is the same as literature values, then it would mean the previous values were perfect, which would mean there it does not need to learn from data, but this is not the case. Hence, the precise correlation value here is not that important. We can calculate the correlation, which is 0.843 with CLM4.5 $V_{c,max25}$.

We attached below figure 8 (added in the revised manuscript) showing the correlation between the $V_{c,max25}$ learnt by V+B model versus TRY database or other default models. As the figure shows, there is high correlation between the estimated $V_{c,max25}$ by V+B model versus CLM4.5 (0.843), BETHY (0.897), and TRY (0.698). However, low correlation exists between the estimated $V_{c,max25}$ by V+B model and AVIM model where the V+B has lower values for BET Temperate, BET Tropical, and BDT Temperate while it shows higher values for BDS Temperate, C3 grass, and Crop R. It is difficult to comment which set is better without all models being run on the same dataset.

[Figure]

**Figure 8. shows the correlation between the $V_{c,max25}$ values estimated by V+B model on** the y-axis versus $V_{c,max25}$ values from **CLM4.5 (black markers), AVIM (cyan markers), BETHY (magenta markers), and TRY database (orange markers). Different marker shapes represent different PFTs, while different colours represent different reference sources for $V_{c,max25}$ per PFT. For** the TRY database, we don't have values for C3 grass and C4 grass due to the lack of overlap in species between the **TRY database and our dataset for those two PFTs.**

For Figure 6b (7b in the revised manuscript), since one point is for a PFT for CLM4.5, and $V_{c,max25}$ is defined on a PFT level, there should be exactly the same number of points as there are PFTs. As a result, the number of data points seemed correct.

17. Line 431-432: I cannot identify the C3 grass at the lower left corner of figure 5b. Maybe a violin plot per PFT can be helpful to show the difference between optimizing B or not for a specific plant type. The figures in the paper only show the net photosynthesis rate across all sites. However, the site-level comparison might be more meaningful to assess the four parameterization strategies: Vdef+B, Vdef+Bdef, V+Bdef, and V+B.

Measurements in Lin15 dataset were taken on sub-hourly scale but not necessarily on a continuous daily interval. For almost all the sites, data were available on some random days (not necessarily continuous) in one or a few years. This means that the data distribution across sites is not balanced some sites have very low amounts of data compared to other sites. For this reason, we didn't assess the models using the site-level comparison but instead computed the metrics for all sites combined. Site-level comparison makes more sense when each site has a large amount of data and the dataset amount is uniformly across sites, which was not the case here.

For better clarification of different PFTs, we split figure 5 (6 in the revised manuscript) into 3 rows representing 3 PFTs (each with 2 subplots) as shown in question 14 above (in the detailed comments) by the first reviewer. We used the cross-validation dataset for making the plot to avoid confusion concerning which dataset was used for the plot (train or test). Splitting figure 5 (6 in the revised manuscript) this way helps present the same information as the previous violin plot (which we exclude in the new version to avoid redundancy).

18. Line 445-450: I didn't see any significant correlation between the estimated Vcmax25 and the PFT-mean from TRY database or other model default values. The authors should provide the scatter plots and the correlation coefficients between the Vcmax values to conclude that the dPL can get parameters correlated with literature values (line 490).

Previously responded to in comment No.16 in the major comments.

Minor comments:

1. Line 123: the right part looks very similar to the middle part in equation 3, but the subscript 'W' beside 'argmin' is not explained. As I understand, the 'argmin' in the right part is the same as the 'argmin' in the middle part.

The equation was modified to have w below argmin and not as a side subscript.

$$W = \underset{W}{\operatorname{argmin}}(L(\delta_{psn}(g^W(R), \theta_c, F), y^*)) \tag{3}$$

By this way, we mean to express that our target is to find the weights of the neural network that minimize the loss function between the observed and the target variable which is the net photosynthesis rate here. So, W here refer to the neural network weights ($NN_V$ and $NN_{Bi}$) in our problem. This was explained clearly in the manuscript paragraph preceding this equation.

2. Line 142: The short name for CO2 partial pressure at the leaf surface is 'Ca', but is 'Cs' in the appendix. Please use a uniform short name across the paper.

$C_s$ and $C_a$ refer to different variables, however, there definitions are close to each other.
$C_s$: is the $CO_2$ partial pressure **at** the leaf surface.
$C_a$: is the $CO_2$ partial pressure **near** the leaf surface.
They are correct in the way the equations are written inside the manuscript body and the appendix. We modified the definition of $C_a$ {lines 171: 176} in section 2.2 (**The Farquhar photosynthesis model**) to describe $CO_2$ partial pressure **near** the leaf surface as shown in the following:

*"Equation 4 is a single-variable nonlinear equation, with the intercellular leaf $CO_2$ pressure ($C_i$) as the unknown term to be solved (serving as the x term in Equation 1). $C_i$ is influenced by the $CO_2$ partial pressure near the leaf surface ($C_a$), the net photosynthetic rate ($A_n$), the atmospheric pressure ($P_{atm}$), the leaf stomatal conductance ($g_s$), and the leaf boundary layer conductance ($g_b$). Upon solving for $C_i$, we can further calculate $A_n$, which is the y term in equation 1. In the original implementations of FATES and CLM, the system of nonlinear equations was solved iteratively using fixed-point iteration (Oleson et al., 2013)."*

3. Line 187: equation 11 is cited at line 187 for the first time, but the full equation is placed at line 218. The equation should appear close to the first citation.

The equation is written in a more general way as $B = NN_B(R)$ at line 187 (in the previous manuscript version), where R refers to the underlying attributes or the raw inputs. In line 218 (in the previous manuscript version), we show the actual equation that we used for the parameterization in our study. To avoid confusion, we removed this equation "$B = NN_B(R)$" in the revised manuscript and kept only equation 11.

4. Line 193: does 'i' represent the soil layer number? I didn't see the explanation around the equation.

Yes, the subscript $i$ refers to the soil layer number. We clarified this in the revised manuscript {lines 227: 229} in section 2.3 **(The parameterization pipeline and model changes)** as the following:

*"B is purely a function of soil properties and is defined for each soil layer as $B_i$ where i refers to the soil layer (see Appendix B). $B_i$ equations will later be replaced by our NN-based parameterization scheme as explained in section 2.3.1, because they were originally empirical and may not be optimal at the global scale."*

5. Line 197: 'across different soil different layers' should be 'across different soil layers'.

Modified it in the revised manuscript {lines 239: 240}.

6. Line 203/equation 9: the second line should be $T_i \leq T_f - 2$ 'or' $\theta_{liq} \leq 0$.

Modified it in the revised manuscript {see Equation 8}.

7. Line 205: the short name for the physical parameter at the second blue area should be $\theta$ but not $\theta_c$.

Modified it in the revised manuscript {see Figure 1}.

8. Line 218/equation 11: B and Fom should have a subscript, i.

$B_i = NN_{Bi}(\%sand, \%clay, PFT, F_{om}, i)$.

Modified it in the revised manuscript {see Equation 11}.

9. Line 222: the 'one-hot embedding' was already stated at line 183. The definition should be explained where it is mentioned for the first time.

We better clarified that in the revised manuscript by differentiating between two terminologies: one hot encoding {lines 220: 221} and embedding layer {lines 257: 260}:

In section 2.3 (**The parameterization pipeline and model changes**), we defined one hot encoding as the following:
*"where PFT is the plant functional type category (in one-hot encoding format, which translates each category to a binary vector) and the neural network used for parameterization of $V_{c,max25}$ is referred to as $NN_V$ hereafter."*

We later defined embedding layer as the following:
*"Since in $NN_{Bi}$, we use quantitative inputs (%$sand_i$, %$clay_i$, $F_{om,i}$) along with categorical inputs (PFT), we used the embedding layer in PyTorch, which translates each category to a vector of quantitative variables. This categorical data can then easily be combined with other quantitative inputs we provide to our neural network."*

10. Line 228: the short name for 'differentiable learning framework' is defined but not used.

In line 228 (in the previous manuscript version), 'differentiable learning framework' refers to the dPL "differentiable parameter learning framework". To unify, we changed it in the revised manuscript to be "differentiable parameter learning framework".

11. Line 310/Figure 2: the full names of the land cover types (e.g., BET tropical) are not explained before or around the figure.

Our dataset included 9 different PFTs categories, a paragraph (see comment No.3 in the major comments) with more details about Lin15 dataset was added to section 2.4.1 (**Forcing and Photosynthesis rates**) stating the number of PFTs considered plus the full name of each PFT.

12. Line 349: table 2 is mentioned for the first time, but the full table is placed after two pages.
We have better organized the revised manuscript.

13. Line 384: the CO2 should be CO2(subscript).
Modified it in the revised manuscript.

14. Line 390/figure 5: I cannot understand the titles of the subplots. What is the meaning of 'learning B' and 'learning Vcmax25'? The B is not optimized in figure 5a.
"Learning" describes which parameter is being obtained. Figure 5a (6a in the revised manuscript) subplot shows two models, Vdef+Bdef (red color) and Vdef + B (blue color). So both models agree in using the default $V_{c,max25}$ values corresponding to each PFT that's why subplot (a) title includes **"with default $V_{c,max25}$"**. **"Learning B"** is added to title "a" since B is learnt in Vdef + B model.

Figure 5b (6b in the revised manuscript) subplot shows two models, V+Bdef (yellow color) and V+B (green color). So both models agree in learning $V_{c,max25}$ values corresponding to each PFT that's why subplot (b) title includes **"Learning $V_{c,max25}$"**. **"Varying B"** is added to title "b" since

the parameter B is computed from the default equations in CLM4.5 for V+Bdef model, while it is learnt simultaneously with $V_{c,max25}$ for V+B model.

We have made some adjustments to the figure caption which should hopefully be more clear. The revised caption is shown in question 14 above (in the detailed comments) by the first reviewer.

15. Line 514/equation A7: the Cs is not used.

Cs which refers to the $CO_2$ partial pressure at the leaf surface is used in the model block of equations corresponding to the stomatal conductance computations. Figure 2 (added in the revised manuscript and copied below) shows equations corresponding to f and h. The box marked with red color shows the usage of Cs

[Figure]

**Figure 2. Model block of equations corresponding to f and h in equation 1. Blue boxes indicate equations corresponding to f. Yellow boxes indicate equations corresponding to h. First, we obtain a solution for Ci (intercellular leaf CO₂ pressure) by solving the nonlinear system (f equations) as illustrated in the last blue box. Then, we forward h equations to compute Aₙ (net photosynthesis rate) using Aᴄ, Aⱼ, and Aₚ as discussed in section 2.2. Details about different variables and parameters included in f and h equations are provided in Appendix A.**

16. Line 520-530: the three functions, Φ1, Φ2, and Φ3, need to be clarified.

$\Phi_1$, $\Phi_2$, and $\Phi_3$ refer to the equations or the subroutines that we used to prepare the inputs required to run the FATES photosynthesis module. To run the photosynthesis module, we had to run other correlated subroutines in FATES that provide some crucial inputs required to simulate the net photosynthetic rate.

$\Phi_1$ corresponds to the set of equations in which we used factors from literature or from the Community Land Model (CLM) to map the maximum electron transport rate at 25 °C ($J_{max25}$), the plant respiration rate at 25 °C ($R_{d25}$), the initial slope of $CO_2$ response curve at 25 °C ($K_{p25}$) from $V_{c,max25}$ as shown below:

| | |
|---|---|
| $J_{max25} = 1.67\, V_{c,max25}$ | (Medlyn et al., 2011) |
| $R_{d25} = \begin{cases} 0.015\, V_{c,max25}, & \text{for C3 plants} \\ 0.025\, V_{c,max25}, & \text{for C4 plants} \end{cases}$ | (Lawrence et al., 2019) |
| $K_{p25} = \begin{cases} 20000\, V_{c,max25}, & \text{for C4 plants} \end{cases}$ | |

$\Phi_2$ corresponds to the equations responsible for rescaling and adjusting the parameters $J_{max25}$, $K_{p25}$, and $V_{c,max25}$ for the leaf temperature to output $J_{max}$, $K_p$, and $V_{c,max}$

$\Phi_3$ corresponds to the equations responsible for rescaling and adjusting $R_{d25}$ for the leaf temperature to output $R_d$.

All these equations are well documented in FATES code and in CLM5.0 (Lawrence et al., 2019) in chapter 9 section 9.4.We added this paragraph {lines 664: 670} in appendix A to better clarify as the following:

*"The three biophysical rates $V_{c,max}$, $J_{max}$, and $K_p$ along with the plant respiration ($R_d$), adjusted for $T_v$ are calculated using their standardized values at 25°C multiplied by temperature response functions defined in chapter 9.0 in CLM5.0 (Lawrence et al., 2019). $V_{c,max}$ is also adjusted for the soil water availability by multiplying it with the soil water stress function($\beta_t$).*

*In our case, $V_{c,max25}$ is either the default value provided in CLM4.5 or is learned by a neural network, which then is used to calculate other standardized biophysical rates as:*

$$J_{max25} = 1.67\, V_{c,max25} \tag{A6}$$

$$R_{d25} = \begin{cases} 0.015\, V_{c,max25} & for\ C3\ plants \\ 0.025\, V_{c,max25} & for\ C4\ plants \end{cases} \tag{A7}$$

$$K_{p25} = \{20000\, V_{c,max25}\ for\ C4\ plants\} \tag{A8}$$

*"*

17. Appendix: the citations of equations are wrong (e.g, lines 503-504, 512, 520, 534…). The equations should be cited using A1-A23.

The citations for all equations in Appendix A were modified to A[no.] in the revised manuscript.

The authors' reply is timely and clarifies most of the questions and confusion. I'm looking forwards to reading the new version of this manuscript! According to the new information provided by the authors, the manuscript presented how well the net photosynthesis can be simulated using two parameters (Vcmax25 and Bi) predicted via a simple MLP neural network (one hidden layer) with a few attributes (PFT, %sand, %clay and Fom). After reading the authors' reply, I still have the following concerns and comments:

1. There are limited site-level temporal data, thus the seasonality of net photosynthesis cannot be assessed.

We did not claim the model can simulate seasonality very well at a site. Currently our differentiable model follows the same structure of the photosynthesis module in the process-based model "FATES". We didn't make significant changes to the model. Because the backbone of the model is Farquhar, its seasonal behavior should be comparable to what we expect out of the other Farquhar models, because in this paper we only estimated static parameters. On the other hand, the nature of our dataset doesn't enable us to test the seasonality and we didn't mention this in the manuscript. This might be within our scope in future work, but is not here.

To avoid confusion, we added this paragraph to the discussion section in the revised manuscript {lines 608: 615} as the following:
*"Although applying the dPL framework improved the parameters to an extent, the model still has similar structural limitations as other Farquhar-type models. We didn't test the model's ability to capture the seasonality of the net photosynthetic rate due the limited site level temporal data. The seasonal behavior of the model is expected to be similar to other Farquhar models as here we only learned static parameters. Further improvement likely will need to consider vegetation growth. Also, this study doesn't cover the spatial generalization of the dPL model since we don't present results for spatial tests or based on site-level comparison. To improve spatial generalization may require further changes in the model, dynamical parameters, or using other error mitigation approaches. This is not our scope for this study; however, it will be considered for future work."*

2. The violin plots showed the net photosynthesis per PFT, but I think readers would be more interested in how different is the simulated net photosynthesis from the measured net photosynthesis. Maybe the fourth violin plot (measured values) can be added on the right side and the NSE can be displayed at the top. Moreover, I think only the test dataset (or better cross-validated dataset) should be compared with the measured values (e.g., Fig. 5) and used to make the violin plots.

For better clarification of different PFTs, we split figure 5 (6 in the revised manuscript) into 3 rows representing 3 PFTs (each of 2 subplots) as shown in question 14 above (in the detailed comments) by the first reviewer. We used the cross-validation dataset for making the plot to avoid confusion concerning which dataset was used for the plot (train or test). Splitting figure 5 (6 in the revised manuscript) this way helps present the same information as the violin plot (which we now exclude in the new version to avoid redundancy).

3. The authors clarified that the Vcmax25 is predicted per PFT but did not mention Bi. Is Bi predicted per PFT or per site? How is the predicted Bi compared with values from CLM?

$B_i$ differs between different sites and for one site it differs for different PFTs; $B_i = NN_{Bi}(\%sand_i,$ $\%clay_i, F_{om,i}, PFT)$. Contrary to $V_{c,max25}$, there are no default values for B because of two reasons:

   a. $B_i$ in the default CLM4.5 equations come from empirical equations based on %clay and $F_{om}$

   b. We changed equation 7 to equation 10 (as shown below). Thus, parameter $B_i$ in equation 7 has a completely different range from the one in equation 10 which ranges between 0 and 1

**Equations 7 and 10**

$$\Psi_i = \Psi_{sat,i} \times S_i^{-B_i} \geq \Psi_c \qquad\qquad\qquad\qquad (7)\text{ default}$$

$$\Psi_i(PFT) = \Psi_o \times S_i^{-B_i(soil,PFT)} \geq \Psi_c \qquad\qquad\qquad (10)\text{ New}$$

**Default B equations in CLM4.5**

$$B_i \quad = (1 - F_{om,i}) \times B_{min,i} + F_{om,i} \times B_{om} \qquad\qquad (B9)$$

$$B_{min,i} \quad = 2.91 + 0.159 \times (\%clay)_i \qquad\qquad\qquad (B10)$$

To avoid confusion, we modified and added new text and equations {lines 241: 252} in section 2.3.1 (Model changes) as the following:

*"In the original water limitation function in CLM4.5, the stomata response to soil water potential is based on a linear function between the water potential for stomata openness and closeness (see Equation 8). In light of the possibility that plants could respond differently to soil water potential dependent on plant hydraulic traits (Christoffersen et al., 2016), in this study, we modified the soil water limitation for PFTs so that they could have different shapes. Specifically, we defined PFT-dependent soil water stress, $\psi_i(PFT)$ ranging from $\psi_c$ and $\psi_o$, depending on the soil water content, which is calculated as follows:*

$$\Psi_i(PFT) = \Psi_o \times S_i^{-B_i(soil,PFT)} \geq \Psi_c \qquad\qquad\qquad (10)$$

*$B_i$ is a PFT- and soil-texture-dependent shape parameter (between 0 and 1) estimated as:*
$$B_i = NN_{Bi}(\%sand_i, \%clay_i, F_{om,i}, PFT) \qquad\qquad\qquad (11)$$

*where $\%sand_i$, $\%clay_i$, and $F_{Im,i}$ respectively represent the percentage of sand, the percentage of clay, and the fraction of organic matter in soil layer i. The PFT-dependent soil water stress, $\psi_i$ (PFT), is then fed into the plant wilting equation (9) as the following:*

$$w_i = \frac{\Psi_c - \Psi_i(PFT)}{\Psi_c - \Psi_o} = \frac{\Psi_c - max(\Psi_c, \Psi_o \times S_i^{-B_i(soil,PFT)})}{\Psi_c - \Psi_o} \leq 1 \qquad (12)$$

*The new shape parameter B$_i$ in equation 11 has a different range (between 0 and 1) from the original one defined by Clapp & Hornberger (1978) in equation 7 and it varies spatially for different static attributes and for different PFTs as well."*

As shown in the above, we clearly stated that $B_i(soil, PFT)$ is different from the old parameter since it is now a function in soil attributes with PFT, and it also ranges between 0 and 1. We restated that in section 2.5 (Synthetic data and real data experiments) by adding the following text {lines 390: 393}:

*"A complete disagreement or a different order of magnitude would suggest that our values may be not physical. Partial discrepancies would highlight any knowledge gaps. We didn't perform a similar comparison between learned and computed B$_i$ values from default equations since the new shape parameter B$_i$(soil, PFT) (see Equation 11) is different from the original one and has a different range (between 0 and 1)."*

4. Since the site-level comparison and the site-average An comparison are not possible, the generalizability cannot be evaluated. However, the model performance across sites can be compared to other papers using the Farquhar model (e.g., Fig 1B of Chen at al., PNAS, https://doi.org/10.1073/pnas.2115627119, 2022).

Concerning the spatial generalization or the site-level comparison, as mentioned in the previous response (No.12 in Reviewer 2's major comments), a spatial test is not within the scope of this paper. We are working on further improving the spatial generalization with some error mitigation approaches. This will add lots of content and should be for the scope of another paper. So, we added the following paragraph to the discussion section {lines 611: 615}:

*"Also, this study doesn't cover the spatial generalization since we don't present results for spatial tests or based on site-level comparison. To improve spatial generalization may require further changes in the model, dynamical parameters, or using other error mitigation approaches (Feng et al., 2021, 2022b; Ma et al., 2021a). This is not our scope for this study; however, it will be considered for future work."*

Things being asked for and added or modified in the new manuscript version:

1. Explanation for temporal and cross validation test >> (clarified in the revised manuscript through paragraphs added)

2. f1 (renamed to f) and f2 (renamed to h) equations clear explanation in the manuscript body >> (proposed figure 2 added with explanation for the terms in the equations in Appendix A)

3. Details on NNs hyperparameters and hyperparameters tuning >> (clarified in the revised manuscript through paragraphs added)

4. Inquiries about Lin15 dataset >> (Number and full name of PFTs, forcing variables, atmospheric $CO_2$ ($C_a$), leaf layer boundary conductance ($g_b$)) >> (clarified in the revised manuscript through paragraphs added)

5. Reasons of replacing $\Psi_{sat}$ by $\Psi_o$ in equations 7 and 10 >> (clarified in the revised manuscript through paragraphs added)

6. CLM4.5 contribution to the study >> (clarified in the revised manuscript through paragraphs added)

7. Inquiries about B calculations across soil layers >> (clarified in the revised manuscript through paragraphs added for synthetic and real case experiments)

8. Cross validation tests >> (were performed and results were added in the revised manuscript)

9. Model performance is impacted by certain set of model equations and forcings >> (clarified in the revised manuscript through paragraphs added to the discussion section)

10. Modify typos in model equations >> (modified in the revised manuscript)

11. $NN_B$ and $NN_V$ constraints on outputs and output range >> (clarified in the revised manuscript through paragraphs added)

12. More complex NN for the real case than synthetic case >> (already done for $NN_{Bi}$ but not applicable for $NN_V$)

13. Loss function clarification >> (better clarified in the revised manuscript + Figure 1 modified)

14. Timeseries of observations >> (can't be provided due to the site limited temporal data)

15. Spatial variability of the parameters not fully captured by dPL >> (spatial test is not the scope of this paper, we are working on further improving the spatial generalization with

some error mitigation approaches. This will add lots of content and should be for the scope of another paper.)

16. Soil organic carbon content unit conversion >> (clarified in the revised manuscript)

17. $V_{c,max25}$ correlation literature values >> (figure 8 was added showing the correlation between learnt and the reference $V_{c,max25}$ values)

18. Split plots per PFT >> (Figure 6 was split into 3 rows each with only 3 PFTs)

19. Figure 5b (6b in the revised manuscript) plotted using both training and test datasets >> (was replotted using the cross- validation test results in the revised manuscript)

References:

Feng, D., Beck, H., Lawson, K., and Shen, C.: The suitability of differentiable, learnable hydrologic models for ungauged regions and climate change impact assessment, Hydrology and Earth System Sciences Discussions, 1–28, https://doi.org/10.5194/hess-2022-245, 2022.

Lawrence, D. M., Fisher, R. A., Koven, C. D., Oleson, K. W., Swenson, S. C., Bonan, G., Collier, N., Ghimire, B., van Kampenhout, L., Kennedy, D., Kluzek, E., Lawrence, P. J., Li, F., Li, H., Lombardozzi, D., Riley, W. J., Sacks, W. J., Shi, M., Vertenstein, M., Wieder, W. R., Xu, C., Ali, A. A., Badger, A. M., Bisht, G., van den Broeke, M., Brunke, M. A., Burns, S. P., Buzan, J., Clark, M., Craig, A., Dahlin, K., Drewniak, B., Fisher, J. B., Flanner, M., Fox, A. M., Gentine, P., Hoffman, F., Keppel-Aleks, G., Knox, R., Kumar, S., Lenaerts, J., Leung, L. R., Lipscomb, W. H., Lu, Y., Pandey, A., Pelletier, J. D., Perket, J., Randerson, J. T., Ricciuto, D. M., Sanderson, B. M., Slater, A., Subin, Z. M., Tang, J., Thomas, R. Q., Val Martin, M., and Zeng, X.: The Community Land Model Version 5: Description of New Features, Benchmarking, and Impact of Forcing Uncertainty, Journal of Advances in Modeling Earth Systems, 11, 4245–4287, https://doi.org/10.1029/2018ms001583, 2019.

Medlyn, B. E., Duursma, R. A., Eamus, D., Ellsworth, D. S., Prentice, I. C., Barton, C. V. M., Crous, K. Y., de Angelis, P., Freeman, M., and Wingate, L.: Reconciling the optimal and empirical approaches to modelling stomatal conductance, Global Change Biology, 17, 2134–2144, https://doi.org/10.1111/j.1365-2486.2010.02375.x, 2011.

Tsai, W.-P., Feng, D., Pan, M., Beck, H., Lawson, K., Yang, Y., Liu, J., and Shen, C.: From calibration to parameter learning: Harnessing the scaling effects of big data in geoscientific modeling, Nat Commun, 12, 5988, https://doi.org/10.1038/s41467-021-26107-z, 2021.

---

## Author Response (AR2)

Andreas Ibrom

Thank you for your evaluation!

We have implemented the changes. We have also made some revisions so that the proof stage is easier:
1, We revised Figure 1 to remove some arrows that went out of the NN. It now illustrates the approach concept more correctly.
2. As promised, we have made the code repository publicly available and has announced it in the Code Availability section.
3. The numbers are figures were updated slightly, but no statements or conclusions need to be changed. This mainly resulted from our rigorous self-check, in which we updated a formula for more scientific rigor. As you can see from a track change, the changes to the figures are hardly discernible. The update to the numbers in the text are also minor.

Please consider the following technical corrections:

line 190: add a 'the' before 'adjoined'
Done

lines 459-461: remove the repetitive sentence and add 'see methods section, ' before 'Table 2b'.
Done

line 490: replace 'have' by 'has' (the subject 'group' is singular)
Done

line 638: replace 'can' by 'does'
Modified to "Not only is differentiable modeling able to improve simulation quality and provide model parameterization, **it also allows us** to modify model structure and ask questions regarding unclear parts of the model in the future."

line 679: check the unit "s^-1/2"
Equation removed and the sentences was updated as the following since we no longer use this equation:
"... $g_b$ is the leaf boundary layer conductance, which was available in Lin15 for some sites and the missing values were filled using the mean $g_b$ of the whole dataset"